Letter

# Genome-wide association study reveals mechanisms underlying dilated cardiomyopathy and myocardial resilience

Dilated cardiomyopathy (DCM) is a heart muscle disease that represents an important cause of morbidity and mortality, yet causal mechanisms remain largely elusive. Here, we perform a large-scale genome-wide association study and multitrait analysis for DCM using 9,365 cases and 946,368 controls. We identify 70 genome-wide significant loci, which show broad replication in independent samples and map to 63 prioritized genes. Tissue, cell type and pathway enrichment analyses highlight the central role of the cardiomyocyte and contractile apparatus in DCM pathogenesis. Polygenic risk scores constructed from our genome-wide association study predict DCM across different ancestry groups, show differing contributions to DCM depending on rare pathogenic variant status and associate with systolic heart failure across various clinical settings. Mendelian randomization analyses reveal actionable potential causes of DCM, including higher bodyweight and higher systolic blood pressure. Our findings provide insights into the genetic architecture and mechanisms underlying DCM and myocardial function more broadly.

DCM is a disease of the cardiac muscle characterized by increased left ventricular (LV) dimensions and decreased contractile function, which is not explained by abnormal loading conditions or coronary artery disease (CAD)[1–5]. DCM represents a main cause of morbidity and mortality, as it predisposes to heart failure (HF) and lethal arrhythmias[3,4]. While causal rare genetic variants are found in up to 25% of probands, most cases do not harbor a known monogenic cause of disease[6,7]. Furthermore, actionable disease mechanisms remain elusive, with few preventative therapeutics[4]. Genome-wide association studies (GWAS) have recently demonstrated a polygenic contribution to DCM[8–11], opening an avenue for new mechanistic discovery, although these smaller studies were limited in power and identified only a handful of significant loci.

Here, we set out to assemble a large-scale GWAS meta-analysis using six datasets, comprising clinical DCM case–control and biobank sets. We included a total of 4,343 clinically ascertained DCM cases from three datasets (Fig. 1 and Supplementary Tables 1 and 2), including two published DCM datasets[8,10] (one reanalyzed; Supplementary Note)

and a new clinical dataset from Amsterdam UMC (with one significant locus at *BAG3*; Supplementary Note, Supplementary Table 3 and Supplementary Figs. 1 and 2). We also performed harmonized GWAS of a strict, billing-code based phenotype of nonischemic DCM (NI-DCM) in three biobank datasets. Substantial yield was afforded by the FinnGen study[12] (*n* = 3,350 cases; 14 loci; most significantly at *BAG3* and *HSPB7*), with additional contributions from the United Kingdom (UK) Biobank (UKB; one locus at *BAG3*)[13] and Mass General Brigham Biobank (MGB)[13,14] (Supplementary Tables 1 and 2 and Supplementary Figs. 1 and 2). We found strong genetic support for the strict biobank-based DCM construct (Supplementary Tables 4 and 5 and Supplementary Note). In comparison, we explored a broader definition of nonischemic cardiomyopathy (NICM)[15,16], which yielded diminished discovery yield despite substantially larger case numbers (Extended Data Fig. 1, Supplementary Note and Supplementary Figs. 2 and 3). Therefore, we proceeded with the strict NI-DCM phenotype and performed a GWAS meta-analysis across all biobank and clinical DCM datasets, hereafter 'GWAS-DCM.'

✉e-mail: ellinor@mgh.harvard.edu; mark.daly@helsinki.fi; karagam@broadinstitute.org; c.r.bezzina@amsterdamumc.nl

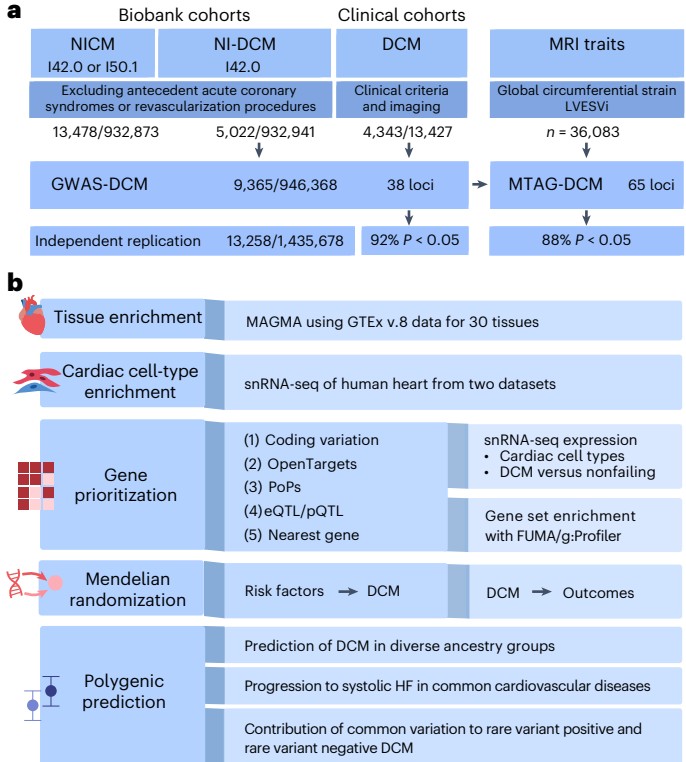

**Fig. 1 | Study design and flowchart. a**, Design of genetic discovery analyses. GWAS were conducted in biobank cohorts for NICM and NI-DCM, and in clinical cohorts that ascertained DCM cases. GWAS results for NI-DCM and clinical DCM were aggregated in a meta-analysis (GWAS-DCM). GWAS-DCM was further combined with GWAS data for cardiac MRI traits (global circumferential strain and left ventricular end systolic volume) in an MTAG. Case and control numbers are represented as no. of cases/no. of controls. **b**, Various downstream analyses conducted using GWAS-DCM and MTAG-DCM results. We used tissue and cardiac-cell-type-specific enrichment analyses to identify tissues and cell types of relevance to DCM. To identify potentially causal genes from the analyses, five complementary methods were used to prioritize genes from associated loci. Prioritized genes were further evaluated in gene set enrichment and cell-type-specific DE analyses. To identify potential causes and consequences of DCM, we used Mendelian randomization analyses, modeling DCM both as exposure and outcome, across a range of common diseases and traits. PRS for DCM were constructed and their utility in predicting NI-DCM was assessed across different ancestries; we assessed the prediction of systolic heart failure across a range of clinical settings. Within the Amsterdam cohort, we assessed the predictive capacity of PRS for DCM, and assessed whether PRS distributions and contributions differed depending on rare pathogenic variant status.

GWAS-DCM included 9,365 cases and 946,368 controls and included 12,600,235 common variants (minor allele frequency (MAF) > 0.5%) after quality control (Fig. 1). The meta-analysis showed some genomic inflation ($\lambda_{GC,LDSC} = 1.19$; $\lambda_{GC}$, genomic inflation factor; LDSC, linkage disequilibrium score regression), which could largely be resolved as polygenic signal (LDSC intercept = 1.06; Supplementary Table 4 and Extended Data Fig. 2). At conventional genome-wide significance ($P < 5 \times 10^{-8}$) we uncovered 38 distinct loci, 27 of which had not been previously described for DCM (Fig. 2a, Supplementary Tables 6–8 and Supplementary Note).

Most of the previously published DCM loci were recapitulated in GWAS-DCM[8,11] (Supplementary Table 6 and Extended Data Fig. 3). Furthermore, most loci overlapped with DCM loci from a recent preprint by Zheng et al.[17] (Supplementary Note). GWAS-DCM signals showed strong pleiotropic effects on relevant cardiovascular traits, including cardiac magnetic resonance imaging (MRI) traits, electrocardiographic traits, blood pressure, HF and arrhythmia (Supplementary Note).

Previously published GWAS for DCM used multitrait analyses GWAS (MTAG)[18] to boost discovery power for new loci[11]. We similarly aimed to maximize discovery using an MTAG approach, using GWAS of eight LV traits from 36,083 UKB participants[19]. We identified two clusters of genetically correlated traits that included endophenotypes with strong genetic correlation to GWAS-DCM (Supplementary Fig. 4 and Supplementary Table 9). Using the most strongly correlated trait from each cluster—global circumferential strain (Ecc; $r_g = 0.75$ with DCM) and LV end systolic volume (LVESVi; $r_g = 0.7$ with DCM)—we performed an MTAG for DCM ('MTAG-DCM'). MTAG-DCM identified 65 significant loci, 50 of which had not been published previously for DCM (Supplementary Tables 10–12; Extended Data Fig. 3 and Supplementary Note).

We then performed a replication analysis using independent samples from HERMES (Heart Failure Molecular Epidemiology for Therapeutic Targets), MVP (Million Veteran's Program) and the 'All of Us'[20] datasets, totaling up to 13,258 cases of NICM/DCM and 1,435,287 controls (Extended Data Fig. 4 and Supplementary Tables 13 and 14). Of 36 testable GWAS-DCM loci, all were concordant in effect direction and 92% replicated at $P < 0.05$. Of 64 testable MTAG-DCM loci, 88% replicated at $P < 0.05$ (81% for 'MTAG-only' loci; Supplementary Note). No loci showed meaningful heterogeneity in discovery (Supplementary Tables 15 and 16). These results confirm the robustness of our GWAS and MTAG approaches.

To identify cell types of relevance to DCM biology, we performed enrichment analyses using two published LV single nucleus RNA sequencing (snRNA-seq) datasets[21,22]. Only cardiomyocyte-specific genes were significantly and robustly enriched for DCM heritability across datasets ($P < 3 \times 10^{-7}$ for enrichment coefficient; Supplementary Table 17, Extended Data Fig. 5 and Supplementary Fig. 5). Of note, Zheng et al. described enrichments for DCM heritability in other cardiac cell types[17]; this discrepancy is most probably due to technical differences, including use of a different enrichment statistic[23] (Supplementary Note). Taken together, our results highlight the central role of cardiomyocyte dysfunction in DCM pathogenesis.

We applied various approaches for variant-to-gene mapping[24–26] (Methods). In ten GWAS-DCM loci, a lead variant was linked to a protein-altering coding variant affecting a single gene (for example, *BAG3*, *TTN*, *FHOD3*, *ADAMTS7*, *CAND2*; Supplementary Tables 6 and 10). Among these, *BAG3*, *TTN* and *FHOD3* represent known Mendelian cardiomyopathy genes[7,27,28]. A well-imputed (INFO = 0.997) *TUBA8* missense variant (22:18609493:G:A) was a lead variant in GWAS-DCM (Supplementary Fig. 6). TUBA8 is an α-tubulin predicted to be a component of myocyte cytoskeletons[29]. The variant was testable only in FinnGen, reflecting an 18-fold enrichment in Finnish over non-Finnish Europeans[30].

Colocalization analyses with molecular traits—using expression quantitative trait loci (eQTLs) for LV from the genotype-tissue expression project (GTEx)[31], eQTLs for blood from eQTLGen[32] and protein quantitative trait loci (pQTLs) in blood from the UKB Pharma Proteomics Project (PPP)[33]—helped prioritize genes and informed direction of effect in certain loci (Supplementary Table 18). We found 24 distinct transcripts/proteins associated with DCM at high posterior probability (PP4 > 70%). For instance, genetically predicted lower LV expression of *TMEM182* (encoding a regulator of myoblast differentiation[34]) and lower genetically predicted blood expression of *FBXO32* (a recessive DCM gene[35,36]) were associated with increased DCM risk. Higher predicted expressions of several genes, including *MLF1, MMP1* and *MAPT*, were associated with increased DCM risk.

We found that the polygenic priority score method (PoPS) was a powerful tool to identify cardiomyopathy genes, as the top 100 genes from GWAS-DCM were enriched 119-fold (95% confidence interval (CI) (47–285), two-sided $P < 2.6 \times 10^{-16}$; Fisher exact test) for known Mendelian DCM and hypertrophic cardiomypathy (HCM) genes (ClinGen genes at ≥moderate evidence; Supplementary Table 19). Therefore, PoPS was assigned high weight in our final prioritization score.

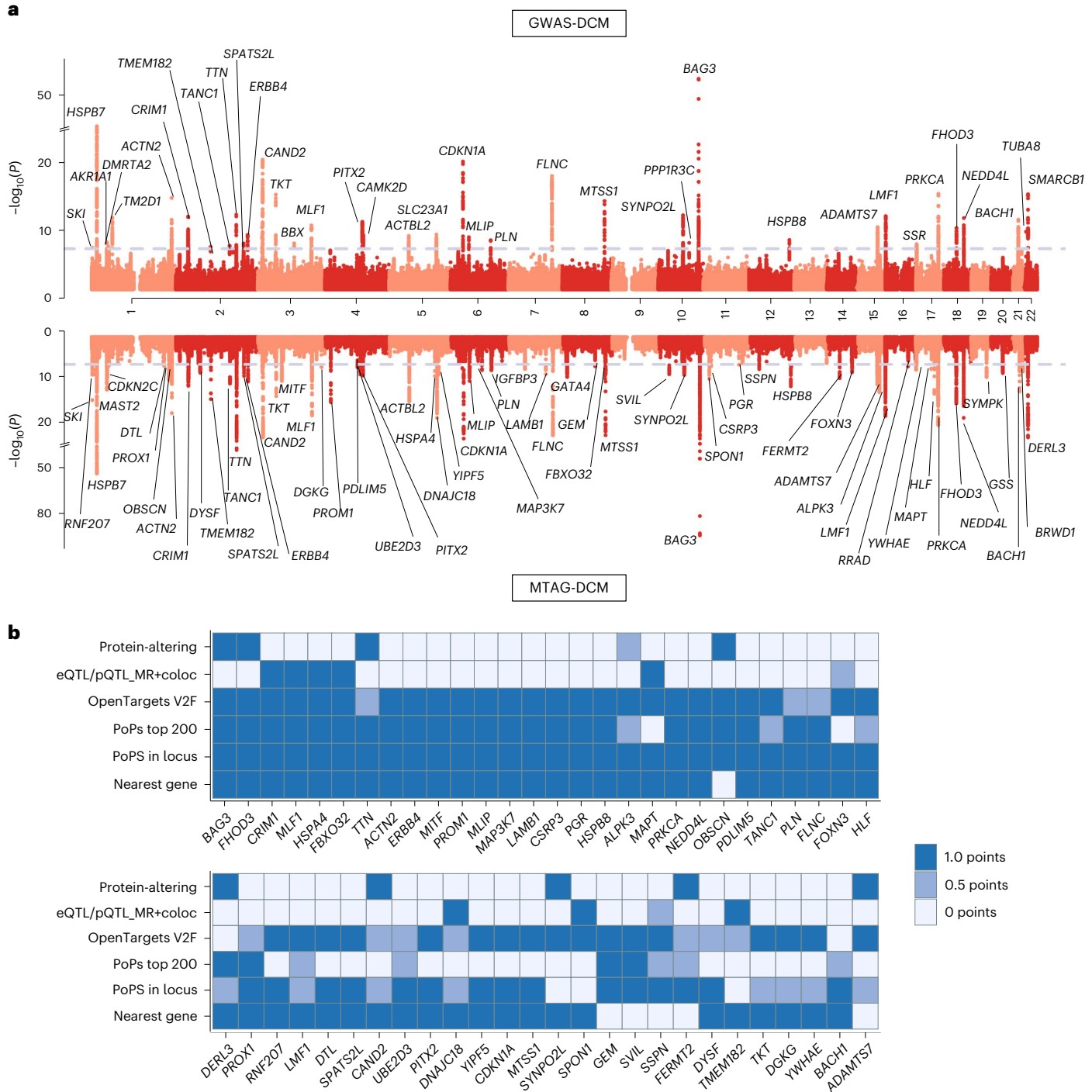

**Fig. 2 | Locus and gene discovery for DCM. a**, Miami plot for the GWAS and MTAG for DCM. Top, results from the GWAS meta-analysis for DCM (GWAS-DCM) that included 9,365 cases and 946,368 controls; bottom, results from the MTAG integrating the GWAS-DCM with cardiac MRI traits (MTAG-DCM). In both plots, the *y* axis represents the −log₁₀ of the *P* value, and the *x* axis represents genomic positions (chromosome, and chromosomal positions) of variants, where each dot represents a single test statistic for a single variant. *P* values are derived from inverse-variance-weighted meta-analysis of logistic regression models (GWAS-DCM) or from MTAG analysis of such statistics (MTAG-DCM); reported *P* values are two-sided and unadjusted for multiple testing. The significance threshold is determined by the dotted lines at the conventional genome-wide level ($\alpha = 5 \times 10^{-8}$). Significant loci are annotated with their most highly prioritized

gene (Methods); loci not overlapping with previous genome-wide significant loci (from published DCM-GWAS or published multitrait studies) are highlighted in bold. **b**, Gene prioritization overview for the top prioritized genes from MTAG-DCM. The heatmaps show the different gene prioritization methods on the *y* axis and prioritized genes on the *x* axis. Genes are ordered from left to right based on their priority score (high to low); the top part of the heatmap shows the genes with the highest scores. A color mar indicates assignation of points based on the given prioritization method; prioritized genes were defined as genes with 2.5 or higher points, which were also the most highly prioritized in their respective loci. For a similar plot for GWAS-DCM, see Extended Data Fig. 6. coloc, colocalization analyses.

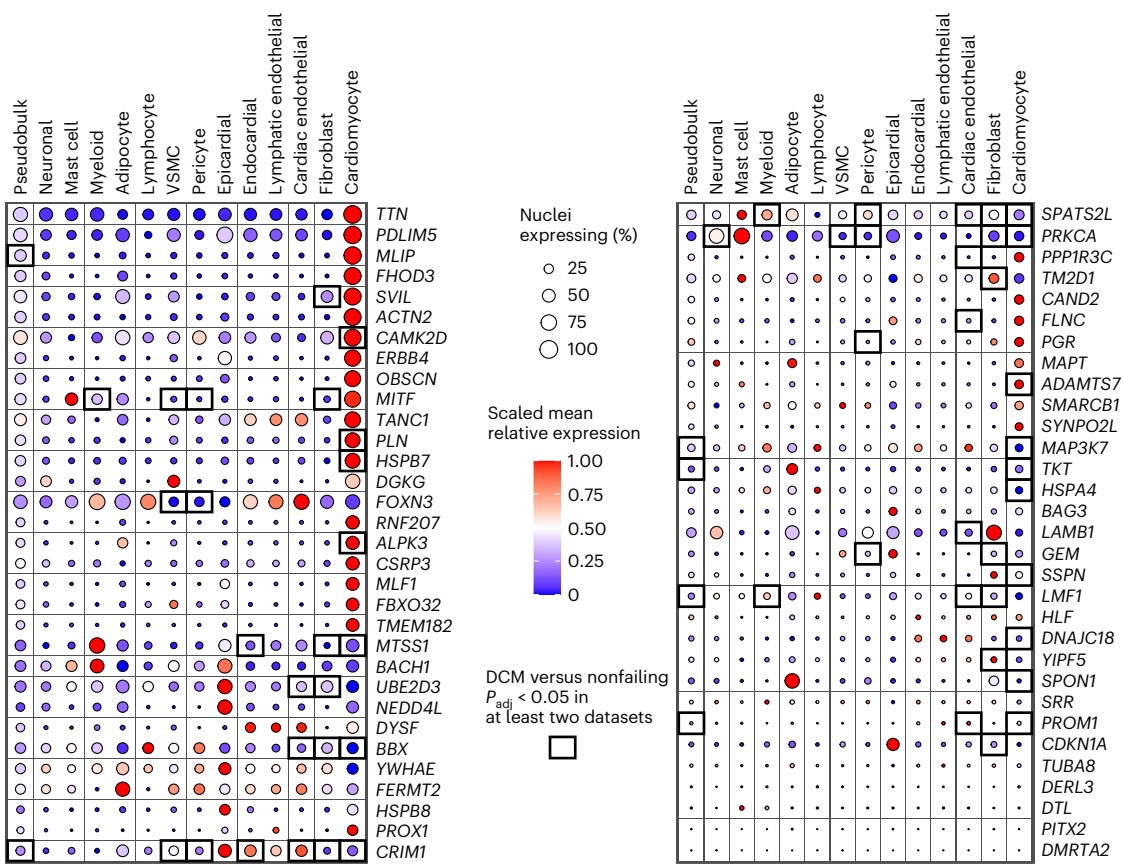

**Fig. 3 | Cell-type-specific expression and DE of the top prioritized genes for DCM from three single-cell LV datasets.** Bubble-heatmap showing data collected from three published sn/scRNA-seq datasets of DCM and control LVs[21,22,41]. The y axis represents a shortlist of highly prioritized genes from GWAS-DCM and MTAG-DCM (63 genes), while the x axis shows different LV cell types harmonized across the three expression datasets. Cell type expression data were computed by combining reformatted data from the three datasets, after restricting to LV samples from nonfailing donors ($n_{max}$ = 61 donors; Supplementary Note and Supplementary Table 24). The size of the dots represent the percentage of nuclei/cells expressing a given gene in a given cell type at nonzero values, while the color of the dot represents the scaled relative normalized expression of the given gene in the given cell type (as compared with all other cell types). A black border indicates that the given gene is significantly differentially expressed in the given cell type in DCM LVs ($n_{max}$ = 82 patients) as compared with the nonfailing LVs ($n_{max}$ = 61 donors); significant DE was declared if the gene reached $P_{adj}$ < 0.05 with concordant direction of effect in at least two of the sn/scRNA-seq datasets within similar cell types (Supplementary Table 25). P values were derived from DEseq2 DE frameworks; P values are two-sided. Of note, not all cell types were assessed in DE testing in all three datasets, and therefore the approach is conservative for less-abundant cell types (for example, epicardial, adipocyte, lymphatic endothelial), although useful for more abundant cell types (for example, cardiomyocyte, fibroblast, cardiac endothelial). $P_{adj}$, transcriptome-wide multiple-testing-adjusted two-sided P value.

We synthesized the various prioritization approaches into one score to identify a list of prioritized genes (Fig. 2b and Supplementary Tables 20 and 21; Methods). Across prioritized GWAS-DCM genes (n = 35 genes with ≥2.5 points) and MTAG-DCM genes (n = 60 genes), we narrowed down to 63 unique prioritized genes (defined as ≥2.5 points and highest score within a locus in either GWAS-DCM or MTAG-DCM; Fig. 2b and Extended Data Fig. 6). Among these prioritized genes were—as expected—several Mendelian cardiomyopathy genes, but also several genes with unknown or lesser-known roles in the heart (for example, *CRIM1*, *MLF1*, *HSPA4*, *ERBB4*, *MITF*, *MLIP*, *MAP3K7*, *NEDD4L*, *DNAJC18* and *HSPB8*). *HSPB8*, *HSPA4* and *DNAJC18* encode proteins from the heatshock family, along with *HSPB7*, a gene functionally validated in DCM biology after being identified initially through GWAS[37].

Accordingly, gene set enrichment analyses, using the 63 prioritized genes, identified several significant gene sets including 'Cellular response to heat stress' (Supplementary Tables 22 and 23 and Supplementary Fig. 7). Most remaining gene sets were related to (cardiac) muscle development and function. Other distinct pathways emerged including ERBB signaling[22] and cytoskeletal organization[38,39], as well as 'Apoptosis by doxorubicin' and 'Aberrant mitosis by docetaxel.'

Doxorubicin and docetaxel are chemotherapeutics that may induce DCM-like phenotypes[40].

To scrutinize the prioritized genes further, we queried published single-cell data of the human LV from three datasets[21,22,41]—including data from 61 nonfailing donors and 81 DCM patients. We found that many of the prioritized genes showed high and/or preferential expression in cardiomyocytes (Fig. 3 and Supplementary Table 24). These genes underscore the role of the contractile apparatus in DCM pathogenesis[42], through known cardiac sarcomeric genes (for example, *TTN*, *OBSCN* and *ACTN2*), but also lesser-described structural genes including *SVIL* (encoding an actin-binding protein recently implicated in HCM[19]) and *PDLIM5* (encoding a cytoskeletal linker[43]). Other genes with cardiomyocyte-specific expression included *MITF* (encoding a transcription factor implicated in cardiac hypertrophy in vitro[44]) and *MLIP* (encoding a lamin-interacting protein associated with myocardial adaptation in mice[45]). Several genes showed significant differential expression (DE) between DCM and nonfailing hearts (Fig. 3 and Supplementary Table 25). Notably, within cardiomyocytes, such genes included *MAP3K7* (encoding a mitogen-activated protein implicated in cardiospondylofacial syndrome[46]), *ADAMTS7* (encoding a thrombospondin-regulating metalloprotease[47]) and both *PRKCA* and

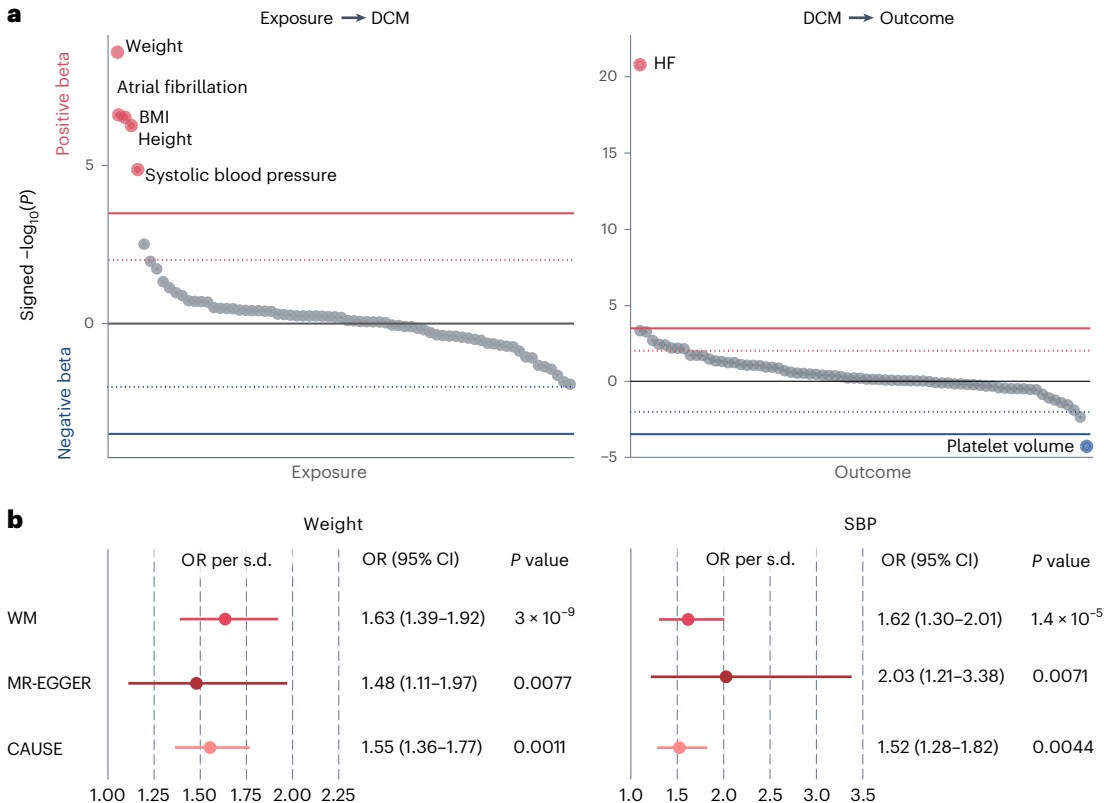

**Fig. 4 | Bidirectional MR screen for DCM and 73 common diseases and quantitative traits. a,** Bubble plot showing results from the MR screen using the WM method. Left panel, results from analysis modeling DCM as outcome; right panel, results modeling DCM as exposure. In both panels, the y axis represents the signed $-\log_{10}$ of the P value from the MR analysis where each bubble represents a different disease/trait (exposure or outcome) and $-\log_{10}$ (P values) are signed by the direction of the MR effect estimate. In both panels, the diseases/traits are ordered by their signed $-\log_{10}$ (P values) from high (left) to low (right). The full red and blue lines represent the Bonferroni-corrected significance level (P < 0.05; 146 tests), while the dotted lines represent P < 0.01. Traits/diseases reaching Bonferroni significance in the screen are annotated with their names. Reported P values are two-sided and unadjusted for multiple testing. **b,** Forest plots showing more detailed results and sensitivity analyses performed for two traits associated with increased DCM that passed all MR filters. Left panel, results for MR of bodyweight; right panel, results for SBP. The different MR effect estimates represent results from different methods: WM (discovery analysis), MR-Egger and CAUSE. The MR-Egger and WM P values are two-sided. For CAUSE, the P value is not based on the CI of the estimate, but rather represents a one-sided P value for the comparison with a pleiotropy model (Methods). For weight, $n_{instruments}$ = 733 in the WM and MR-Egger analyses, while $n_{instruments}$ = 2,286 in the CAUSE analysis. For SBP, $n_{instruments}$ = 376 in the WM and MR-Egger analyses, while $n_{instruments}$ = 1,846 in the CAUSE analysis. Error bars, 95% CIs for the estimated effect. All reported P values are unadjusted for multiple testing.

*CAMK2D* (involved in calcium handling[48,49]). Of note, several genes highlighted from both GWAS and single-cell data are being investigated as targets for other conditions (Supplementary Table 26). These results show how integration of GWAS and single-cell data—paired with appropriate cell type priors—may identify plausible gene candidates for cardiomyopathy and LV function.

We next used genetic data to identify potential causes and consequences of DCM through Mendelian randomization (MR)[50]. We performed a bidirectional MR screen using the weighted median (WM) method, based on genetic instruments constructed from GWAS for 73 common diseases and quantitative traits (Methods). At Bonferroni significance, we identified five potential causal risk factors for DCM (weight, body mass index (BMI), atrial fibrillation (AF), systolic blood pressure (SBP) and height), and two potential consequences of DCM liability (HF and mean platelet thrombocyte volume; Fig. 4a, Supplementary Table 27 and Supplementary Note). Weight, systolic blood pressure and AF remained as independent risk factors for DCM in multivariable MR (Supplementary Table 28). While these results partially recapitulate previous descriptions of causal factors for general HF[51], we did not observe evidence for a causal role of coronary disease (g = −0.09, P = 0.13) or diabetes (g = −0.05, P = 0.18) on DCM.

To scrutinize the potential causal associations further, we employed two additional methods. First, we used MR-Egger regression[50] and found that most of the signals survived filtering using this method ($P_{slope}$ < 0.05 and $P_{intercept}$ > 0.1; Fig. 4 and Supplementary Table 27). Second, we used CAUSE—an approach that models a pleiotropic pathway and tests whether a causal model is a better fit for the data than a sharing model[52] (Supplementary Table 29 and Supplementary Figs. 8–10). CAUSE estimated that BMI, weight and SBP all conferred increased risk of DCM (Fig. 4b). The causal role of blood pressure is consistent with the main pharmacotherapeutic approach to DCM, which consists partly of blood-pressure-lowering medications[4]. Similarly, there is a growing body of observational evidence linking obesity to risk of HF, DCM and other cardiomyopathies[53–55]. In summary, our data support that SBP and weight are reasonable parameters for action in (premorbid) DCM management.

We then constructed polygenic risk scores (PRS) from our GWAS-DCM and MTAG-DCM summary statistics[56], and tested these in three datasets. PRS constructed from GWAS-DCM and MTAG-DCM were associated significantly and strongly with DCM (Fig. 5 and Supplementary Tables 30–32), with MTAG-DCM scores yielding the best predictive performance across all tested strata (Extended Data Figs. 7 and 8, Supplementary Fig. 12 and Supplementary Note). For instance,

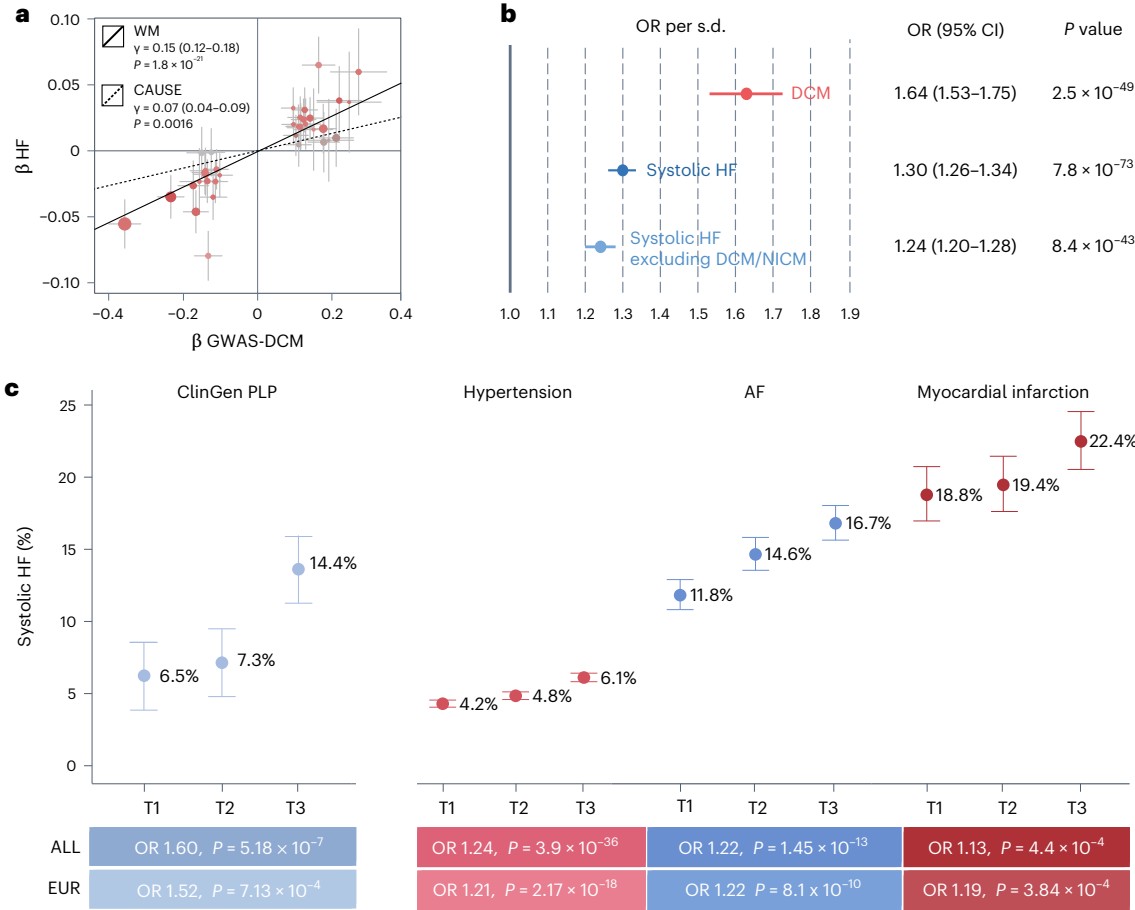

**Fig. 5 | DCM genetic liability as a predictor of systolic HF across a range of settings in All of Us. a**, MR scatter plot for DCM liability on risk of HF. The *x* axis shows beta coefficients (±s.e.) for 37 genetic instruments identified from GWAS-DCM, while the *y* axis shows the corresponding beta coefficients on general HF[50]. The estimated causal association lines for two methods are added, including the WM method (black line) and CAUSE (dotted line). **b**, Forest plot for associations of DCM PRS in the All of Us dataset: NI-DCM (*n* = 928/181,773 cases/controls); systolic HF (*n* = 5,123/190,410 cases/controls) and systolic HF after removing DCM/NICM (*n* = 4,273/189,976 cases/controls). Statistics are derived from logistic regression models (two-sided, unadjusted for multiple testing). **c**, Prevalence of systolic HF in the All of Us dataset across a range of settings, stratified by DCM PRS. Left, results for individuals who carry rare disease-causing variants for DCM (*n* = 1,429), where the *y* axis represents the percentage with systolic HF at any time and the *x* axis stratifies those individuals into low, middle and high PRS tertiles. Right, a similar plot restricting to three different clinical settings: after hypertension diagnosis (*n* = 76,985), after AF diagnosis (*n* = 11,369) and after myocardial infarction (*n* = 5,098). Cases with systolic HF coded before or concurrently to the index event were removed, leaving *n* = 3,877; 1,634 and 1,028 cases, respectively. Data are presented as percentages with 95% CIs. Beneath each setting, the OR and *P* value for PRS are added, from logistic regression with the quantitative PRS used as predictor (two-sided, unadjusted for multiple testing). ClinGen PLP, carriers of disease-causing rare variants for DCM; T1, tertile 1 of PRS; T2, tertile 2 of PRS; T3, tertile 3 of PRS.

in the All of Us dataset, PRS was associated strongly with DCM among European (OR per s.d. = 1.73; *P* = 9.0 × 10^{-37}) and African ancestries (OR per s.d. = 1.61; *P* = 2.5 × 10^{-10}), with a weaker but significant signal among Admixed-American ancestry (OR per s.d. = 1.34; *P* = 2.4 × 10^{-3}).

In the Amsterdam UMC dataset, clinical DCM cases carrying rare disease-causing variants ('genotype-positive') had significantly lower PRS than genotype-negative DCM cases (*P* = 0.0015), and genotype-negative cases were enriched more strongly for higher PRS (Fig. 6). Nevertheless, DCM PRS was enriched significantly in both groups compared with controls. These results highlight that polygenic burden contributes to disease risk in carriers and in noncarriers of rare pathogenic alleles, although carriers might need less polygenic burden to reach disease state[57,58].

Finally, we assessed whether DCM PRS may have value for prediction of systolic HF—a condition associated with substantial morbidity and healthcare costs[59,60]. In All of Us, we found significant associations for DCM PRS with systolic HF (OR per s.d. = 1.30; *P* = 7.8 × 10^{-73}), which persisted after removal of NI-DCM and NICM cases (OR per s.d. = 1.24; *P* = 8.4 × 10^{-43}; Supplementary Table 33). Furthermore, the

PRS was a predictor of systolic HF across a range of settings, including after AF diagnosis (*P* = 1.4 × 10^{-13}), after hypertension diagnosis (*P* = 2.4 × 10^{-39}), after myocardial infarction (*P* = 4.4 × 10^{-4}) and among carriers of pathogenic rare variants for DCM (*P* = 5.2 × 10^{-7}; Fig. 5 and Extended Data Fig. 9). These findings support the notion that the DCM PRS captures liability to intrinsic myocardial dysfunction or structural weakness, which may determine the resilience of the LV upon experiencing adverse events or prolonged stress.

In summary, we performed a large-scale GWAS and MTAG for DCM—including 9,365 strict DCM cases—and identified 70 loci at genome-wide significance. Several main conclusions arise from our work. First, on a cell-type level, we found that the heritability of DCM is enriched predominantly for cardiomyocyte expression, highlighting the central role of cardiomyocyte dysfunction in DCM pathogenesis. Second, mapping of loci to genes using various methods identified 63 potential effector genes, which may inform on-target and off-target effects in therapeutics development. Third, MR analyses support a causal role of bodyweight and SBP in DCM risk, indicating that early blood pressure regulation and weight reduction may be considerations

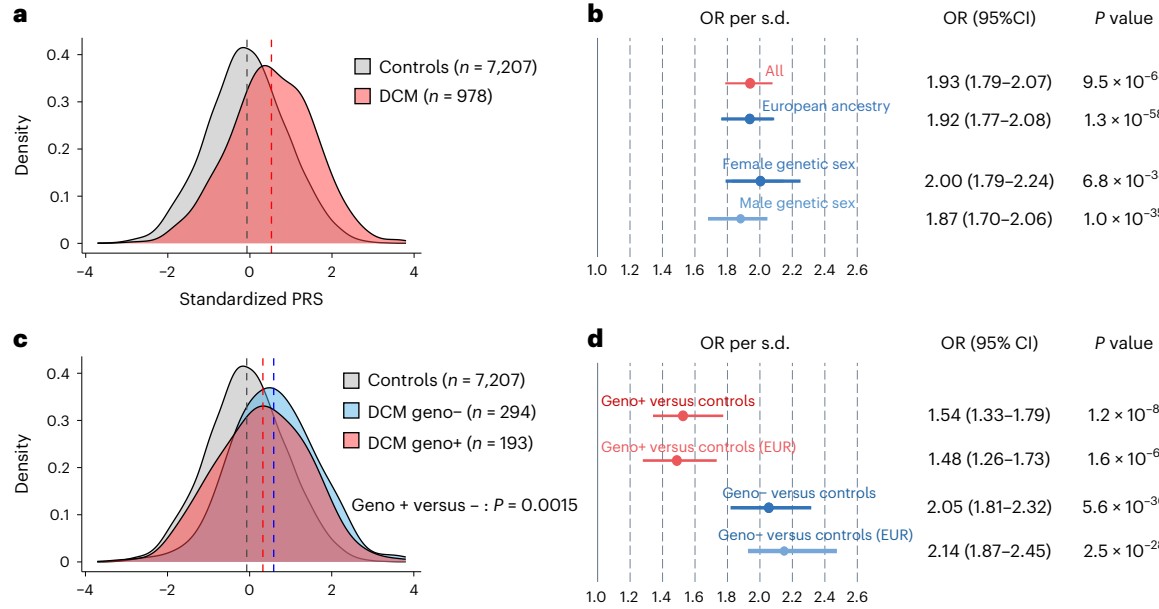

**Fig. 6 | Distribution and association of PRS among DCM patients by rare variant genotype status in the Amsterdam cohort. a**, Density plot with standardized PRS values for individuals from the Amsterdam UMC cohort on the *x* axis, and density representing the frequency of those PRS values in the cohort on the *y* axis. Dotted lines, means of the distributions. **b**, Forest plot showing effect sizes for the PRS in various subsets of the Amsterdam UMC dataset; *x* axis, ORs computed per s.d. of the standardized PRS distribution from logistic regression, adjusting for sex and ancestral PCs. Data are presented as estimated ORs with 95% CIs. **c**, Density plot with standardized PRS values on the x axis and density representing the frequency of those PRS values on the *y* axis. Dotted lines, means of the distributions. A two-sided *P* value from a linear regression model, testing the difference between genotype-positive and negative, is added. **d**, Forest plot showing effect sizes for the PRS where only genotype-positive or only genotype-negative cases are tested against the control cohort using logistic regression, adjusting for sex and ancestral PCs. Data are presented as estimated ORs with 95% CIs. Other performance metrics are presented in Supplementary Table 32. geno−, DCM case without a rare pathogenic or likely pathogenic rare variant; geno+, DCM case with a rare pathogenic or likely pathogenic rare variant.

in DCM patients or at-risk people. Fourth, a PRS derived from our GWAS predicts DCM, with impactful—albeit potentially differing—contributions in carriers and noncarriers of rare pathogenic variants. Fifth, the genetic liability to DCM underlies systolic HF, and may modulate risk of systolic failure across a range of settings. Our results have implications for our understanding of the mechanisms underlying DCM and myocardial resilience.

## Online content

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

## FinnGen

**Joel T. Rämö**[2,3,4,55]**, Amanda L. Elliott**[4,18,23,24,25]**, Juha Sinisalo**[37,38]**, Teemu Niiranen**[39,40,41]**, Jari Laukkanen**[51,52,56]**, Aarno Palotie**[4,53,54,56] **& Mark Daly**[4,53,54,56]

Full lists of members and their affiliations appear in the Supplementary Information.

## VA Million Veteran Program

**Jennifer E. Huffman**[16,17,18]**, Krishna G. Aragam**[2,3,56]**, Kyong-Mi Chang**[46,47] **& Philip S. Tsao**[48,49]

Full lists of members and their affiliations appear in the Supplementary Information.

## HERMES Consortium

**Sean L. Zheng**[11,12,13]**, Albert Henry**[14,15]**, Kiran Biddinger**[2]**, Patrick T. Ellinor**[2,3,56]**, Krishna G. Aragam**[2,3,56]**, James S. Ware**[11,12,13,24] **& R. Thomas Lumbers**[15,50]

Full lists of members and their affiliations appear in the Supplementary Information.

## Methods

### GWAS for dilated cardiomyopathy

We collected data from three case–control datasets that ascertained clinical DCM patients, and data from three large biobank studies. The clinical DCM datasets included (1) a published GWAS by Garnier et al. that enrolled 2,651 DCM cases from France, Germany, Italy, the UK and the United States[8]; (2) a reanalyzed dataset of 909 DCM cases from Heidelberg, Germany[10] (Supplementary Note) and (3) a new dataset of Dutch DCM cases from Amsterdam UMC. The Amsterdam cohort comprised DCM patients referred for genetic testing at Amsterdam UMC, who underwent chart review for DCM diagnosis and had evidence of hypocontractility on imaging; 978 DCM cases passed all genotype quality-control criteria, of which 783 homogeneous cases of patients of European ancestry were included in GWAS (Supplementary Note and Supplementary Table 3). Further details for the various cohorts are described in the Supplementary Note and are summarized in Supplementary Tables 1 and 2. All clinical DCM cohorts applied imaging criteria as part of case definition.

We further performed GWAS in three biobanks, namely FinnGen (freeze 11)[12], UKB[13] and MGB[14] (Supplementary Tables 1 and 2 and Supplementary Note). In these datasets, we defined two phenotypes using International Classification of Disease (ICD) coding. First, we defined an NICM phenotype as described previously[15], using ICD10 code I42.0 'dilated cardiomyopathy' and ICD codes for 'left heart failure,' with exclusion of—at minimum—antecedent acute coronary syndromes and revascularization procedures (Supplementary Table 1 and Supplementary Note). Across biobanks, the NICM definition totaled 13,478 cases. We also defined a strict NI-DCM phenotype using only I42.0 (again with minimum exclusion of antecedent acute coronary syndromes and/or revascularization procedures), totaling 5,022 cases (Supplementary Tables 1 and 2). In all biobank datasets, individuals with other HF codes—but not fulfilling the case criteria—were removed from the controls. In all biobanks, REGENIE[61] was used for GWAS. Further details are presented in Supplementary Table 2 and Supplementary Note.

All study cohorts either collected informed consent from research participants, or received appropriate approval from ethical/review committees to waive the requirement of informed consent. All study protocols were approved by appropriate ethical/review committees; approval was granted as described in the original publications for published cohorts[8,10–14]; the Amsterdam UMC study protocol—focused on GWAS for heritable cardiovascular diseases—was approved by the Amsterdam UMC Medical Ethical Review Committee.

### GWAS meta-analyses

Stringent variant quality control was applied in each dataset. Variants were filtered to high imputation quality (INFO $\geq$ 0.5 or $R^2 \geq$ 0.5; MAF $\geq$ 0.5%; INFO $\geq$ 0.8 if MAF < 1%; INFO $\times$ MAF $\times$ Ncases $\times$ 2 $\geq$ 5; and variants with nonambiguous alleles (Supplementary Table 2). Before meta-analysis, variants were aligned to genome build GRCh38, using the liftOver command line tool if not already on the correct genome build[62]. GWAS meta-analyses were then performed using an inverse-variance weighted fixed-effects approach implemented in METAL[63] (March 25, 2011 release). GWAS meta-analyses were performed combining the three clinical DCM datasets, combining the three NICM GWAS from the biobank datasets and combining the NI-DCM-GWAS from the biobanks. After meta-analyses, results were filtered to common variants (MAF > 0.5%). Variants were considered significant if reaching the conventional genome-wide significance level ($P < 5 \times 10^{-8}$). In all GWAS, hypothesis tests were two-sided.

### Heritability and genetic correlations

We used LDSC[64] (v.1.0.1) to estimate the heritability attributable to common single nucleotide polymorphism (SNP) variants ($h^2_{SNP}$) for different meta-analyses. The European subset of the 1000Genomes[65] (v.3.5) dataset was used as a linkage disequilibrium (LD) reference panel,

and analyses were subsetted to nonambiguous HapMap3 variants. Heritability values were transformed to the liability scale, assuming a population prevalence of 0.4% for DCM[4] and 1.2% for NICM (based on UKB prevalence). We further used bivariate LDSC to estimate the genetic correlations ($r_g$) between the various meta-analyses[4,66]. Hypothesis tests were performed using a null hypothesis of 0, using two-sided tests.

The biobank NI-DCM meta-analysis showed a comparable $h^2_{SNP}$ and high $r_g$ with the clinical DCM meta-analysis (see above), and therefore we proceeded with an overall meta-analysis combining the clinical DCM-GWAS with the biobank NI-DCM-GWAS, from here referred to as GWAS-DCM.

### Multitrait analyses

MTAG leverages the genetic correlation between a target GWAS (for example, for DCM) and GWAS for related traits (for example, LV parameters) to increase the discovery power, while accounting for potential sample overlap. We used MTAG (v.1.0.8) to first estimate a genetic correlation matrix between GWAS-DCM, NICM, HCM[19], and eight LV MRI traits from a previous GWAS ($n$ = 36,083 participants from UKB)[19]. Per SNP effective sample sizes ($n_{\text{snp-eff}}$) were computed from the s.e., using the formula

$$n_{\text{snp-eff}} = 1/(2 \times \text{MAF} \times (1 - MAF) \times (s.e.^2))$$

MTAG developers recommend utilizing GWAS of traits that are strongly genetically correlated with the target GWAS ($r_g > 0.7$). We additionally aimed to reduce the number of included traits to limit potential false-positive findings. After computing an initial genetic correlation matrix (Supplementary Table 9), we identified two large clusters of MRI traits correlated strongly with GWAS-DCM. From the clusters of genetically correlated traits (a 'contractility' cluster and a 'volumetric' cluster; Supplementary Fig. 4), we identified two index traits with $r_g > 0.7$ (Ecc and LVESVi). We then ran MTAG—including GWAS-DCM, Ecc GWAS and LVESVi GWAS—using default parameters. MTAG estimated that the boosted summary statistics for DCM equated to an increase in effective sample size of approximately 73% (ref. 18). The maximum false-discovery rate computed by MTAG was 0.03, meaning that, under the most unfavorable distribution of trait-specific effect sizes, 3% of signals may represent false positives[18]. Imaging-based contractility and LV dimensions represent direct (diagnostic) endophenotypes of DCM[3–5,67]. Therefore, the true false-discovery rate is probably even lower. The results from this analysis are referred to as 'MTAG-DCM.' Significance was determined at the conventional genome-wide level ($P < 5 \times 10^{-8}$). In all MTAG, hypothesis tests were two-sided.

### Locus definitions, variant annotation and gene prioritization

**Functional mapping and annotation processing and annotation.** GWAS-DCM and MTAG-DCM were processed in Functional Mapping and Annotation (FUMA)[68] v.1.6.1. Lead variants were defined as variants at genome-wide significance ($P < 5 \times 10^{-8}$) and $r^2 < 0.05$ (using '1KG/Phase3 EUR' as LD reference). Genomic loci were subsequently defined by merging over 1 Mb distances. FUMA utilizes Multi-marker Analysis of GenoMic Annotation (MAGMA) v.1.08 to perform gene-based testing[69]; FUMA then uses the MAGMA genes for tissue enrichment analysis based on GTEx v.8 expression (GTEx/v8/gtex_v8_ts_general_avg_log2TPM)[31]. Variants, and their LD partners, were further annotated using ANNOVAR[70] (v.2017-07-17). Loci were considered new if none of the lead variants overlapped (at 1 Mb windows) known lead variants from previous DCM-GWAS and DCM MTAG[8,11], or were found associated with DCM according to GWAS Catalog[71] or OpenTargets[24,72] (queried in October 2023).

**Protein-altering variation and closest protein-coding gene.** For gene prioritization, we first assessed whether lead variants were in LD ($r^2 > 0.65$) with protein-coding protein-altering variants based

on ANNOVAR annotations in FUMA. Second, we identified the closest protein-coding gene for lead variants, based on OpenTargets (22.10 update).

**OpenTargets Variant2Function.** Third, we used Variant2Function (V2F) from the OpenTargets platform[24] (22.10 update) to map variants to genes. V2F is a phenotype-agnostic machine-learning algorithm that identifies potential genes affected by genomic variants; we extracted the top three genes identified by V2F as being potentially affected by lead variants from GWAS-DCM and MTAG-DCM.

**Polygenic priority score.** Fourth, we used the PoPS method[25]. PoPS uses gene-level associations—computed from GWAS summary statistics—to learn gene features associated with the trait in a joint model by polygenic enrichment; features consist of cell-type-specific gene expression, biological pathways and protein–protein interactions (PPIs). We first performed gene region based analysis with MAGMA[69] v.1.10 using the European subset of the 1000Genomes Phase 3 as a reference dataset. Based on gene-level results from MAGMA, we computed polygenic priority scores for 18,383 genes using the full set of features provided with PoPS v.0.2.

**MR and colocalization for eQTLs and pQTLs.** Fifth, we used MR of quantitative trait loci for expression (eQTLs) and protein abundance (pQTLs), followed by colocalization[26]. As instruments for expression in the heart, we used *cis*-eQTLs for LV from GTEx[30] v.8 ($n$ = 386 left ventricular samples). As instruments for expression in whole blood, we used *cis*-eQTLs from the eQTLGen consortium[32] ($n$ = 31,684 samples; we used the 2019 dataset, downloaded from https://www.eqtlgen.org/cis-eqtls.html). As instruments for protein abundance, we used pQTLs derived from the UKB PPP, which used the Olink platform for proteomic profiling[33]; we downloaded summary statistics for the 'combined' set (from https://www.synapse.org/#!Synapse:syn51364943/files/; $n$ = ~34,000 samples) and defined *cis*-pQTLs as variants present within 1 Mb of the associated protein. All three datasets were subsequently processed the same way and harmonized with GWAS-DCM or MTAG-DCM summary statistics (Supplementary Note). We defined our instruments by clumping the *cis*-eQTL/*cis*-pQTL variants, using two-sided $P < 5 \times 10^{-8}$, $r^2 < 0.0005$ and window size of 10 Mb in PLINK2 (refs. [32,73]). The R-package TwoSampleMR (v.0.5.6) was used to perform two-sample MR, using Wald ratio tests for single-instruments exposures and using the inverse-variance weighted approach for exposures with multiple instruments[74]. $P$ values from MR were all two-sided. Analyses were performed for both GWAS-DCM and MTAG-DCM; separate Bonferroni corrections were applied to both, and separate corrections were applied for eQTL and pQTL datasets. Significant hits were subsequently subjected to colocalization[75] using the R-package coloc (v.4.0.4) using strict priors (p1 = $1 \times 10^{-4}$, p2 = $1 \times 10^{-4}$, p12 = $1 \times 10^{-6}$). A posterior probability for a shared causal variant (PP4) of >0.5 was considered some evidence of colocalization, while PP4 > 0.7 was considered strong colocalization.

**Omnibus gene prioritization score.** We then assembled the information from the five prioritization methods into one score. Given that PoPS showed a marked enrichment of known Mendelian DCM and HCM genes genome-wide, this method was strongly weighted in the score. In summary:

- We assigned 1 point to a gene if it was the top gene prioritized by PoPS within a locus (defined as within ±500 kb from the lead variant, or ±1 Mb if fewer than two genes within 500 kb) or 0.5 point if within the top three genes.
- We assigned an additional point to genes if they were also among the top 100 PoPS genes genome-wide, or 0.5 points if within place 101–200 genome-wide.

- We assigned 1 point to a gene if it was the nearest protein-coding gene to the lead variant.
- We assigned 1 point to a gene if it was affected by protein-altering variation (in LD with) a lead variant, or 0.5 points if several genes in the locus were implicated by protein-altering variation.
- We assigned 1 point to the highest OpenTargets V2F gene for a lead variant, or 0.5 points for second and third genes.
- We assigned 1 point to a gene within a locus if there was strong evidence from eQTL/pQTL colocalization (PP4 > 0.7), or 0.5 points if there was moderate evidence (PP4 > 0.5) and/or several genes were implicated in the locus by this approach.

In total, therefore, any given gene could attain between 0 and 6 points. For downstream analyses, we assigned the gene with the highest score across lead variants in the locus as the most highly prioritized gene for that locus. In case of ties, we first assessed whether the gene was convincingly prioritized in the locus based on the other discovery approach (that is, GWAS or MTAG); if not, then one was picked at random. From these genes, we further defined a final list of prioritized targets, using a prioritization score cutoff of ≥2.5 points.

## Gene set enrichment analyses

We used two platforms for gene set enrichment analyses. First, we used the FUMA Gene2Func function[68] (v.1.6.1), to perform enrichment analyses restricting to FUMA-curated pathways. As input we used the curated set of prioritized genes across GWAS-DCM and MTAG-DCM ($n$ = 63 genes), and used all Ensembl (v.102) protein-coding genes as background. We required at least two overlapping genes to identify a potential gene set, and we determined significance using a false-discovery-rate adjusted one-sided $P < 0.05$ (by two-step Benjamini–Krieger–Yekutieli method).

We additionally used the g:Profiler platform[76] (v. September 2023) to test for enrichment of gene sets from several predefined sources. The g:Profiler algorithm uses one-sided Fisher's exact tests to test for enrichment of a prespecified list of genes across many gene sets, and subsequently adjusts one-sided $P$ values for multiple testing while taking into account the correlation between gene sets (g:SCS method[76]). Again the 63 prioritized genes were put forward for enrichment testing; g:Profiler used Ensembl v.110 as the background of protein-coding genes.

Since our prioritized genes may have been preselected towards genes with high cardiac expression (that is, through gene features learnt by PoPS), we performed a sensitivity analysis using genes nominated by MAGMA[69]—a method based only on association signals near gene regions.

## Cardiac-cell-type enrichment

To identify causal cell types for GWAS-DCM and MTAG-DCM, we used stratified LDSC, as described in Finucane et al.[23]. To this end, we utilized two published single-nucleus RNA sequencing (snRNA-seq) datasets, one from Chaffin et al.[21] and another from Reichart et al.[22] The Chaffin et al. dataset included LV expression data on 11 DCM hearts, 16 nonfailing hearts and 15 HCM hearts. The cardiomyopathy samples came from explanted hearts with end-stage disease. Chaffin et al. identified 17 main cell types, which were used to define cell-type-specific gene programs for enrichment testing (see Supplementary Note for detailed methods). The Reichart et al. dataset included data on 61 end-stage cardiomyopathy hearts (52 with DCM) and 18 nonfailing controls. Reichart et al. identified nine main cell types in the LV, which were used to define cell-type-specific gene programs for enrichment testing (see Supplementary Note for detailed methods). Finally, in addition to the 'cell-type-specific' expression annotations described above, we also explored 'disease-dependent' cell-type annotations. Disease-dependent programs were based on genes with significant

DE between DCM samples and nonfailing samples, irrespective of their cell-type-specificity. The detailed methods for this analysis are described in the Supplementary Note. Of note, cell-type-enrichment analyses were not informed in any way by our GWAS/MTAG gene prioritization scheme.

**Single-cell expression and DE**

We then aimed to identify cell-type-expression patterns and cellular functions for the prioritized genes from our GWAS and MTAG. To this end, we used available snRNA-seq or scRNA-seq data from three published datasets, including Chaffin et al.[21], Reichart et al.[22] and Koenig et al.[41]. Koenig et al. performed snRNA-seq/scRNA-seq on 18 LVs from DCM patients and 27 LVs from control donors.

Using the processed AnnData/Seurat objects from each study, we first restricted to control/nonfailing samples from the LV, and then log-normalized the expression data with scale 10,000 (if not already normalized). To harmonize cell-type data across datasets, we then used the available cell-type and/or cell-state annotations to collapse or split cell types into 'harmonized' cell types (Supplementary Note). For genes with at least 0.5 points from our prioritization scheme in GWAS-DCM or MTAG-DCM, we then exported several expression measures from each dataset. These included (1) the mean normalized expression within harmonized cell types and pseudobulk data and (2) the percentage of nuclei/cells with nonzero expression for each harmonized cell type and in pseudobulk. We then combined data by taking the weighted average of expression values (weighted by the number of nuclei per cells contributing in each dataset). For plotting purposes, we then focused on the list of 63 prioritized genes and computed the scaled relative normalized expression of a given gene in a given cell type, as compared with all other cell types.

We further aimed to identify genes differentially expressed between DCM and nonfailing hearts. To this end, we utilized results from cell-type-specific DE analysis for DCM versus nonfailing hearts, as described in Chaffin et al.[21] and Koenig et al.[41] For the published Chaffin et al. DE analysis, we consider results suggestive if reaching transcriptome-wide multiple-testing-adjusted two-sided $P < 0.05$ using CellBender-adjusted counts, without failing the 'background contamination' flag. For the published Koenig et al. DE analysis, we considered results suggestive if reaching transcriptome-wide multiple-testing-adjusted two-sided $P < 0.05$. Finally, we used the Reichart et al. dataset[22], to perform a new DE analysis, comparing the 52 DCM LVs with 18 control LVs, using the same cell types that could be included for DE testing in their original publication (Supplementary Note). Again, a transcriptome-wide multiple-testing-adjusted two-sided $P < 0.05$ was considered suggestive. While we acknowledge that the cell types included in DE testing were not perfectly aligned across datasets, we approximately matched cell types to identify signals that were consistent across datasets (Supplementary Table 25). Finally, we declared significance for a gene, if at least two of three datasets showed a suggestive result with concordant direction of effect within comparable cell types.

**MR for DCM**

We used two-sample MR to identify potential causes and consequences of DCM using genetic data[50]. To this end, we utilized the GWAS-DCM summary statistics and additionally collected published GWAS summary statistics for various common diseases and potential risk factors, including AF[77], CAD[78], type 2 diabetes[79], chronic kidney disease[80], HF[50], thyroid disease[81], BMI[82], alcohol use (drinks per day)[83], smoking (cigarettes per day)[83] and an additional 65 commonly measured quantitative traits (including blood pressure, anthropometry and laboratory values)[84]. The GWAS summary statistics were chosen such that they were largely of European ancestry (and if European-only summary statistics were available, those were used; this was chosen to make the LD structure most comparable with the DCM-GWAS) and such that

FinnGen was not included in the GWAS (to keep sample overlap to a reasonable minimum for two-sample MR).

We performed a bidirectional MR screen, where the above-mentioned traits were modeled as exposure and DCM modeled as outcome, and vice versa (DCM modeled as exposure). Harmonization of summary statistics is described in the Supplementary Note. For our discovery analysis, we used the WM method implemented in the R-package TwoSampleMR (v.0.5.6); the WM method may give more robust results than the inverse-variance-weighted approach in case of outliers[50]. Results at a Bonferroni correction (two-sided $P < 0.05$; 146 comparisons) were considered significant. As a secondary filter for significant results, we then used the MR-Egger method. MR-Egger has lower power but may better account for directional pleiotropy, and further provides an estimate of the regression intercept (which may flag implausible relationships between outcome and exposure effects due to correlated directional pleiotropy)[50]. We required that signals persisted with Egger-slope two-sided $P < 0.05$ without a substantial Egger-intercept (two-sided $P > 0.1$).

For any 'exposure to DCM' or 'DCM to outcome' pairs that remained after discovery and MR-Egger filtering, we then assessed the potential causal effect using CAUSE[52] (v.1.2.0)—a recently proposed mixture approach that accounts for correlated and uncorrelated pleiotropy. In short, CAUSE assesses whether GWAS data for two traits are consistent with a causal effect, by fitting and comparing two nested models. These include a 'sharing' model that allows only a pleiotropic pathway, and a 'causal' model that additionally estimates a causal pathway. These models are compared using the expected log pointwise posterior density, and a one-sided $P$ value is computed from a $Z$-test comparing the 'causal' model with the 'sharing' model[52]. For step 1 of CAUSE (estimating nuisance parameters), we used default parameters that include using 1 M random genome-wide markers for parameter estimation. For step 2 of CAUSE (estimating causal effects) we used filtered and pruned variants (two-sided $P < 0.001$ and $r^2 < 0.0005$ over 10 Mb windows) and otherwise default parameters.

**PRS analyses**

We then aimed to assess the performance of DCM PRS for prediction of DCM and systolic HF across ancestries and different clinical settings. To this end, we used the Amsterdam DCM cohort and the All of Us Research Program, as described below. In addition, we assessed the predictive capacity of the PRS in a third dataset, the UKB, as described in detail in the Supplementary Note.

**Association with DCM and systolic HF in All of Us.** All of Us is a cohort study enrolling participants from across the United States, with an emphasis on participants classically underrepresented in genetics research[20,85]. Whole genome sequencing data were available for over 245,000 participants, of which 84% had complete electronic health record linkage. After quality control (Supplementary Note), we were left with 195,533 unrelated samples, of which 102,886 (52.6%) were of genetically defined European ancestry, and of which 928 had NI-DCM. Characteristics can be found in Supplementary Table 30.

From the GWAS-DCM and MTAG-DCM summary statistics, we created various PRS. Since MGB and All of Us have some overlapping samples[86], we reran our GWAS meta-analyses and MTAG omitting MGB for all PRS analyses described in All of Us. Using these updated summary statistics (DCM-GWAS (excluding MGB) and MTAG-DCM (excluding MGB)) we created genome-wide PRS using PRScs (v.2022-11 (ref. 56)). We used the 'auto' function that learns the optimal shrinkage parameter directly from the GWAS summary statistics. Considering our discovery GWAS was of largely European ancestry, we used the ldblk_ukbb_eur files as LD reference. Participants in the All of Us dataset were subsequently scored using the '--score' function in PLINK2 (ref. 73). To account for ancestral differences in PRS distribution in this multi-ancestry dataset, we first regressed the first ten ancestral

principal components (PCs) of ancestry out of the PRS values, and then standardized them to mean 0 and unit variance.

We first tested the association of both PRS with NI-DCM, using logistic regression models adjusting for age, age$^2$, sex and PCs 1–10. We assessed the association of PRS in the entire multi-ancestry cohort, as well as within the three largest ancestral subgroups, namely European ($n = 102,886$), African ($n = 40,496$), and Admixed-American ($n = 30,358$) ancestry. Correcting for the number of tests, we considered results with $P < 0.05$ (($2 \times 4$)) = 0.00625 significant. In all PRS analyses, hypothesis tests were two-sided.

Using the best performing PRS for DCM prediction (MTAG-DCM (excluding MGB)), we then assessed whether PRS could predict systolic HF. We used logistic regression models to predict systolic HF—defined using ICD10-CM code I50.2 (and subcodes; Supplementary Note)—using PRS, adjusting for age, age$^2$, sex and PCs 1–10. Additionally, we assessed whether the PRS could predict these outcomes across a range of clinical settings as a 'second hit,' namely after AF diagnosis, after hypertension diagnosis and after myocardial infarction. In these analyses, individuals with systolic HF coded before or concurrently with the initial event (for example, AF, hypertension, myocardial infarction) were removed from the respective analyses. Furthermore, we also assessed whether the PRS could predict systolic HF in carriers of likely pathogenic or pathogenic variants in high-confidence DCM genes (ClinGen strong/definitive; Supplementary Note). The significance cutoff was set to two-sided $P < 0.05$ (6) = 0.0083. In all these models, we performed sensitivity analyses removing participants with NI-DCM and NICM to assess whether potential signals were driven by these hard phenotypes; we also performed analyses restricting to European ancestry participants to assess whether results were driven solely by continental ancestry.

**Cumulative contribution of rare and common variation to DCM in the Amsterdam cohort.** We next assessed the distribution and discriminatory capacity of DCM PRS within the Amsterdam DCM cohort. The same general methodological framework from the All of Us cohort was applied to construct PRScs scores[56] in the Amsterdam (AUMC) dataset. Notably, however, we included MGB and omitted the Amsterdam cohort from GWAS-DCM and MTAG-DCM to prevent overfitting. As such, PRScs scores were created for GWAS-DCM (excluding AUMC) and MTAG-DCM (excluding AUMC). After scoring all individuals, the first ten PCs of ancestry were regressed out of the PRS values, and were scaled to mean 0 and variance 1 within the dataset.

We then tested whether the PRS based on GWAS-DCM (excluding AUMC) and MTAG-DCM (excluding AUMC) could discriminate between cases and controls, using logistic regression models adjusting for the first ten PCs of ancestry and sex. To assess performance in various subgroups, we assessed (1) all individuals, (2) individuals of European ancestry, (3) individuals of non-European ancestry, (4) male participants only and (5) female participants only. To determine significance, we used Bonferroni correction at two-sided $P < 0.05$ (2 scores × 5) = 0.005. We focused further analyses on the MTAG-DCM (excluding AUMC) PRS, which performed the best across groups (see above).

We then aimed to assess the cumulative contribution of common and rare genetic variation to clinical DCM, as described previously for rare arrhythmia syndromes[57,58]. We grouped DCM cases into 'rare genotype-positive,' 'rare genotype-negative' and 'uncertain rare genotype,' based on clinical genetic testing findings (Supplementary Note). We performed logistic regression analyses restricting to either 'genotype-positive' cases or 'genotype-negative' cases, comparing either with the general control group. We also assessed distributions of PRS using density plots across (1) controls, (2) all cases ($n = 978$), (3) genotype-positive cases ($n = 193$) and (4) genotype-negative cases ($n = 294$). To identify statistical difference between PRS distribution among genotype-positive and genotype-negative cases, we used linear

regression analyses with PRS as outcome and rare variant status as predictor (adjusting for sex and PC 1–10; Supplementary Note). In sensitivity analyses, all above approaches were repeated, restricting to individuals of genetically determined European ancestry, to assess whether results were driven by continental ancestry.

**Reporting summary**

Further information on research design is available in the Nature Portfolio Reporting Summary linked to this article.

## Data availability

Summary statistics for our GWAS meta-analyses have been made available for download through the Cardiovascular Disease Knowledge Portal (https://cvd.hugeamp.org/downloads.html); summary statistics for various meta-analyses, including clinical dataset-only and biobank dataset-only, are available (https://api.kpndataregistry.org/api/d/CQyqth). Our PRS scoring weights—for both GWAS and MTAG scores—have been deposited into the PGS Catalog (publication ID: PGP000672; score IDs: PGS004946–PGS004951) and into the Cardiovascular Disease Knowledge Portal (https://api.kpndataregistry.org/api/d/9jevLe). Access to individual-level data for the Meder et al. cohort, the Garnier et al. cohort, the Amsterdam UMC cohort and MGB will not be made publicly available at this time, due to the restrictive/sensitive nature of the genomic and/or phenotypic data in question. Access to individual-level UK Biobank data, both phenotypic and genetic, is available to bona fide researchers through application on the UK Biobank website (https://www.ukbiobank.ac.uk). Access to individual-level phenotypic and genetic data from All of Us Research Program is currently available to bona fide researchers within the United States through the All of Us Researcher Workbench—a cloud-based computing platform (https://www.researchallofus.org/register/). The Finnish biobank data can be accessed through the Fingenious services (https://site.fingenious.fi/en/) managed by FINBB. Finnish Health register data can be applied for from Findata (https://findata.fi/en/data/). All processed snRNA-seq/scRNA-seq datasets used in the present study are publicly available: the Chaffin et al. dataset is available for download from the Broad Single Cell Portal (https://singlecell.broadinstitute.org/single_cell/study/SCP1303/single-nuclei-profiling-of-human-dilated-and-hypertrophic-cardiomyopathy); the Reichart et al. dataset was downloaded from GEO (https://www.ncbi.nlm.nih.gov/geo/download/?acc=GSE183852&format=file&file=GSE183852%5FDCM%5FIntegrated%2ERobj%2Egz); the Koenig et al. dataset was downloaded from CellxGene (https://datasets.cellxgene.cziscience.com/3716fb19-cedd-4fe5-abc4-5dbeb007fb65.rds). Other datasets include *cis*-eQTLs from the eQTLGen consortium (https://www.eqtlgen.org/cis-eqtls.html); *cis*-eQTLs from GTEx v.8 (https://www.gtexportal.org/home/downloads/adult-gtex#qtl) and tissue expression levels from GTEx v.8 (https://www.gtexportal.org/home/downloads/adult-gtex#bulk_tissue_expression); pQTLs derived from the UK Biobank PPP (summary statistics for the 'combined' set from https://www.synapse.org/#!Synapse:syn51364943/files/); the 22.10 update of the OpenTargets platform (https://genetics.opentargets.org/); GWAS Catalog queried in October 2023 (https://www.ebi.ac.uk/gwas/); ANNOVAR v.2017-07-17 (https://annovar.openbioinformatics.org/en/latest/); 1000 Genomes project Phase 3 (https://www.internationalgenome.org/data/); gnomAD exomes v.2.1 (https://gnomad.broadinstitute.org/downloads); the ClinVar database (https://www.ncbi.nlm.nih.gov/clinvar/) was accessed in April 2023.

## Code availability

Processing of genotype data, quality control, imputation and genome-wide association analyses were performed with various software tools as described in Supplementary Table 2. Notably, in most of the datasets, various versions of PLINK were used for quality control (https://www.cog-genomics.org/plink/ and https://www.cog-genomics.org/plink/2.0/) and various versions of REGENIE were used for GWAS

(https://github.com/rgcgithub/regenie). Meta-analysis of GWAS was performed using the 2011-03-25 release of METAL (https://github.com/statgen/METAL). Heritability and genetic correlation parameters were computed using LDSC v.1.0.1 (https://github.com/bulik/ldsc). Multitrait analysis of GWAS was performed using MTAG v.1.0.8 (https://github.com/JonJala/mtag). For Mendelian randomization analyses, we used R-packages TwoSampleMR v.0.5.6 (https://mrcieu.github.io/TwoSampleMR/), coloc v.4.0.4 (https://github.com/chr1swallace/coloc) and CAUSE v.1.2.0 (https://github.com/jean997/cause/tree/master), implemented in custom MR pipelines (https://github.com/seanjosephjurgens/MR_pipeline_sjj). Annotation of GWAS was performed using FUMA v.1.6.1 (https://fuma.ctglab.nl/) as well as MAGMA v.1.10 (https://ctg.cncr.nl/software/MAGMA/prog/magma_v1.10.zip) and PoPS v.0.2 (https://github.com/FinucaneLab/pops). Gene set enrichment analyses were performed using FUMA v.1.6.1 (https://fuma.ctglab.nl/) and g:Profiler v.September 20 2023 (https://biit.cs.ut.ee/gprofiler/). For cell-type-specific heritability analyses, we used R-packages edgeR v.3.22.3 (https://github.com/OliverVoogd/edgeR), DESeq2 v.1.20.0 (https://github.com/thelovelab/DESeq2) and limma v.3.36.2 (https://bioconductor.org/packages/release/bioc/html/limma.html) as well as stratified LDSC v.1.0.1 (https://github.com/bulik/ldsc). For wrangling of single-cell/nucleus data, we used R-package Seurat v.5.0 (https://github.com/satijalab/seurat). For polygenic scoring analyses, we used PRScs v.2022-11 (https://github.com/getian107/PRScs) and PLINK2 (https://www.cog-genomics.org/plink/2.0/; various versions from May 2020 release onwards). All analyses that were run in R, were run in R v.4.0.0.

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

## Acknowledgements

We gratefully thank all research participants, as this work would not have been possible without their contributions. Relevant funding sources for the FinnGen, UKB and All of Us datasets are presented in the Supplementary Note. S.J.J. received research support through the Junior Clinical Scientist Fellowship from the Dutch Heart Foundation (03-007-2022-0035), as well as a doctoral fellowship from the Amsterdam UMC. J.T.R. was supported by a research grant from the Aarne Koskelo Foundation. S.K. was supported by the Walter Benjamin Fellowship from the Deutsche Forschungsgemeinschaft (521832260). L.F.J.M.W. was supported by the Amsterdam UMC YTF, Dutch Heart Foundation Student Grant and the AFIP foundation. P.T.E. was supported by funding from the National Institutes of Health (1RO1HL092577, 1R01HL157635, 5R01HL139731), by a grant from the American Heart Association (18SFRN34110082) and from the European Union (MAESTRIA 965286). This work was further supported by a grant from the National Institutes of Health (1K08HL153937) and a grant from the American Heart Association (862032) to K.G.A. C.R.B. was supported by funding from the Dutch Heart Foundation (CVON 2018-30 PREDICT2) and the Pathfinder Cardiogenomics programme of the European Innovation Council of the European Union (DCM-NEXT). A.A.M., D.R.K. and Y.M.P. were supported by funding from the PSIDER programme of the Netherlands Organisation for Health Research

and Development (ZonMW; project 40-46800-98-018). T.N. was supported by grants from the Sigrid Jusélius Foundation, the Finnish Foundation for Cardiovascular Research and the Finnish Research Council (grants 321351 and 354447). This work was supported by a grant from the GENMED Laboratory of Excellence on Medical Genomics (ANR-10-LABX-0013)—a research program managed by the National Research Agency (ANR) as part of the French Investment for the Future; Aviesan-ITMO Genetique-Genomique-Bioinformatique (ResDiCard: Resolving diagnostic deadlock in cardiomyopathies) and the Société Française de Cardiologie/Fédération Française de Cardiologie; the SFB-TR19 registry was supported by the Deutsche Forschungsgemeinschaft (DFG). The Study of Health in Pomerania (SHIP) is part of the Community Medicine Research net of the University of Greifswald, Germany, funded by the Federal Ministry of Education and Research (grants 01ZZ9603, 01ZZ0103, and 01ZZ0403); the Ministry of Cultural Affairs and the Social Ministry of the Federal State of Mecklenburg-West Pomerania; and grants from the German Center for Cardiovascular Research (DZHK). The KORA study was initiated and financed by the Helmholtz Zentrum München—German Research Center for Environmental Health, funded by the German Federal Ministry of Education and Research (BMBF) and by the State of Bavaria. KORA research was supported at the Munich Center of Health Sciences (MC-Health), Ludwig-Maximilians-Universität, as part of LMUinnovativ. D.A.T. is supported by the 'EPIDEMIOM-VTE' Senior Chair from the Initiative of Excellence of the University of Bordeaux. This research is based on data from the Million Veteran Program, Office of Research and Development, Veterans Health Administration, and was supported by award I01-BX003362 to P.T. and K.-M.C. This publication does not represent the views of the Department of Veteran Affairs or the United States Government. This work was also supported by the Sir Jules Thorn Charitable Trust (21JTA), Medical Research Council (UK), British Heart Foundation (RE/18/4/34215, SP/17/11/32885), the NIHR Imperial College Biomedical Research Centre, Pathfinder Cardiogenomics programme of the European Innovation Council of the European Union (DCM-NEXT) (101115416) to J.S.W.

## Author contributions

S.J.J., K.G.A. and C.R.B. conceived the study. S.J.J., J.T.R., P.T.E., M.D., K.G.A. and C.R.B were responsible for the overall study design. S.J.J., J.T.R., D.R.K., J.H. and S.G. contributed to the main discovery GWAS analyses. S.J.J., J.T.R. and L.G. performed the main gene prioritization analyses, while J.T.R., D.R.K. and L.F.J.M. performed the main polygenic risk score analyses. M.D.C. performed the main single-cell analyses pertaining to cell-type enrichment and differential expression, while S.J.J. harmonized single-cell data for gene-level expression patterns. S.J.J. and L.F.J.M. performed data visualization, with support from D.R.K., M.D.C. and A.L. Statistical and analytical support was provided by L.-C.W., A.L., F.R., S.K., K.B., A.L.E., X.W., S.K. and D.S.Z. GWAS data for cardiac MRI phenotypes were contributed by C.F. and P.M.M. At the Amsterdam UMC site, patient inclusions and database management saw contributions from D.R.K., C.K.J., C.A.v.O., E.P., Y.M.P., S.v.d.C., A.S.A. and C.R.B., while L.B. handled and processed patient samples. D.I.B., E.J.C.d.G. and J.-J.H. contributed control sample data for the Amsterdam dataset. J.S., T.N., J.L., A.P. and M.D. were responsible for phenotyping, analysis supervision and oversight within FinnGen. J.-F.D., E.V., D.-A.T., R.I. and P.C. were responsible for oversight, inclusions and analysis of French DCM patient samples, while B.M. was responsible for oversight of DCM-GWAS data from Heidelberg. The biological relevance of prioritized genes was assessed by C.D.V. and R.R. using in-house wet-laboratory data. S.L.Z., A.H., J.S.W. and T.L. were responsible for replication data from the HERMES dataset, while J.E.H., S.C., K.-I.C. and P.S.T. were responsible for replication data from the Million Veteran Program. S.J.J. performed replication analyses using the All of Us data, and performed the meta-analysis of the various replication datasets. Important intellectual contributions were provided by R.T., Y.M.P., A.A.M.W., J.S., T.N., R.W., A.F.S., S.H.C., J.S.W., T.L. and S.v.d.C., while main supervision of the study was provided by J.L., A.P., A.S.A., P.C., B.M., P.T.E., M.D., K.G.A. and C.R.B. S.J.J., J.T.R., D.R.K., P.T.E., M.D., K.G.A. and C.R.B. wrote the manuscript. All authors reviewed and critically revised the manuscript.

## Competing interests

P.T.E. has received sponsored research support from Bayer AG, IBM Health, Bristol Myers Squibb and Pfizer; he has consulted for Bayer AG, Novartis and MyoKardia. K.G.A. has received sponsored research support from Sarepta Therapeutics and Bayer AG, and reports a research collaboration with Novartis. Y.M.P. is involved in the development of therapies for DCM as an advisor to Forbion and Medical Director at ARMGO pharma and CMO at Phlox Therapeutics. P.C. reports personal fees for consultancies, outside the present work, for Amicus, OWKIN, Pfizer and SANOFI. J.S.W. has received research support from Bristol Myers Squibb, and has acted as a consultant for MyoKardia, Pfizer, Foresite Labs, Health Lumen and Tenaya Therapeutics. The remaining authors declare no competing interests.

## Additional information

**Extended data** is available for this paper at https://doi.org/10.1038/s41588-024-01975-5.

**Correspondence and requests for materials** should be addressed to Patrick T. Ellinor, Mark Daly, Krishna G. Aragam or Connie R. Bezzina.

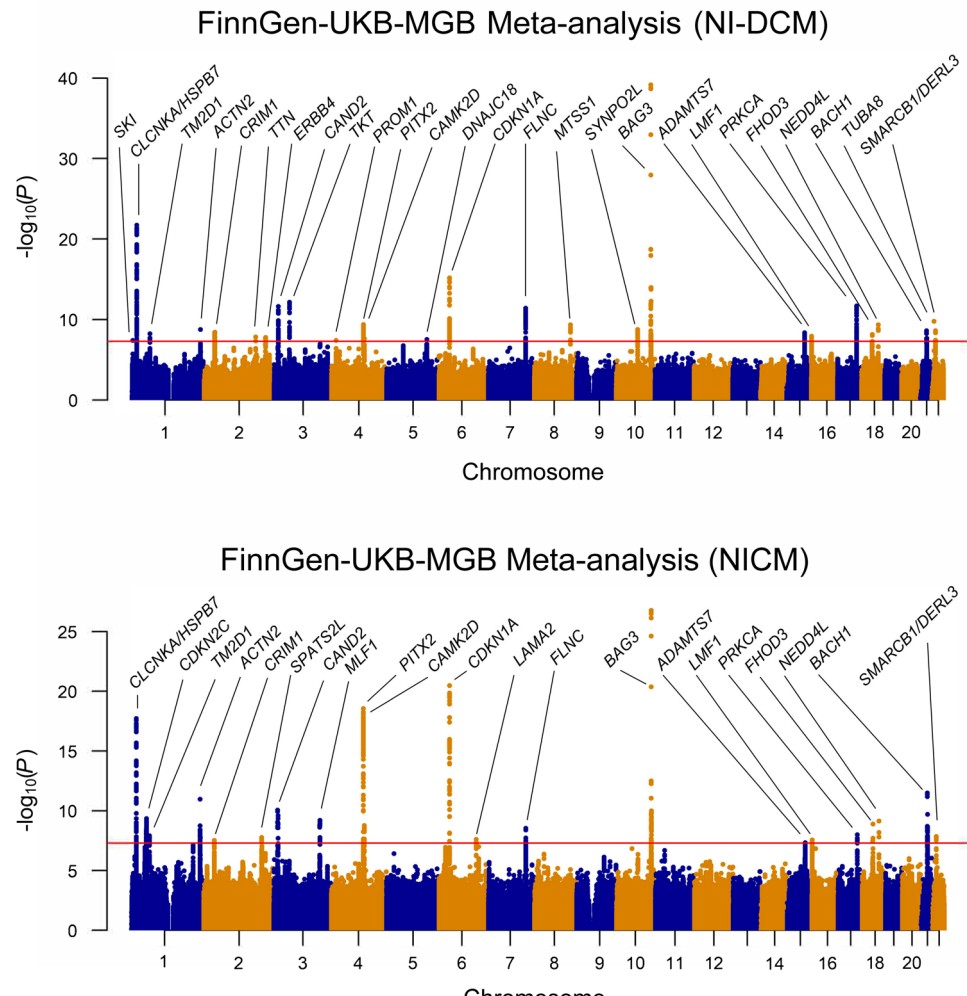

**Extended Data Fig. 1 | Manhattan plots for biobank meta-analysis for NI-DCM and NICM.** Each panel shows a Manhattan for a GWAS meta-analysis of a phenotype across biobank datasets (FinnGen+UKB+MGB), where the top plot shows the results for the strict NI-DCM phenotype (N = 5,022 cases; N = 932,941 controls), and the bottom plot shows results for the broader NICM phenotype (N = 13,478 cases; N = 932,873 controls). In both figures, each dot represents a single tested variant, the x-axis shows genomic coordinates for those variants (chromosome, and position on chromosome), while the y-axis shows the -log10 of the $P$-value from GWAS. $P$-values are derived from inverse-variance-weighted meta-analysis of logistic regression models; reported $P$-values are two-sided and unadjusted for multiple testing. The red line indicates the conventional genome-wide significance level (alpha = $5 \times 10^{-8}$). Loci reaching above the significance line are annotated with a gene name, where the annotated gene is harmonized with the locus name from our main GWAS (ie, highest prioritized gene in locus from GWAS-DCM/MTAG-DCM) for easy comparisons; sometimes an additional gene is highlighted to serve easier comparison to previously-published GWAS; if locus was not identified in GWAS-DCM/MTAG-DCM, the closest protein-coding gene is used. Note: GWAS, genome-wide association study; NICM, nonischemic cardiomyopathy; NI-DCM, nonischemic dilated cardiomyopathy.

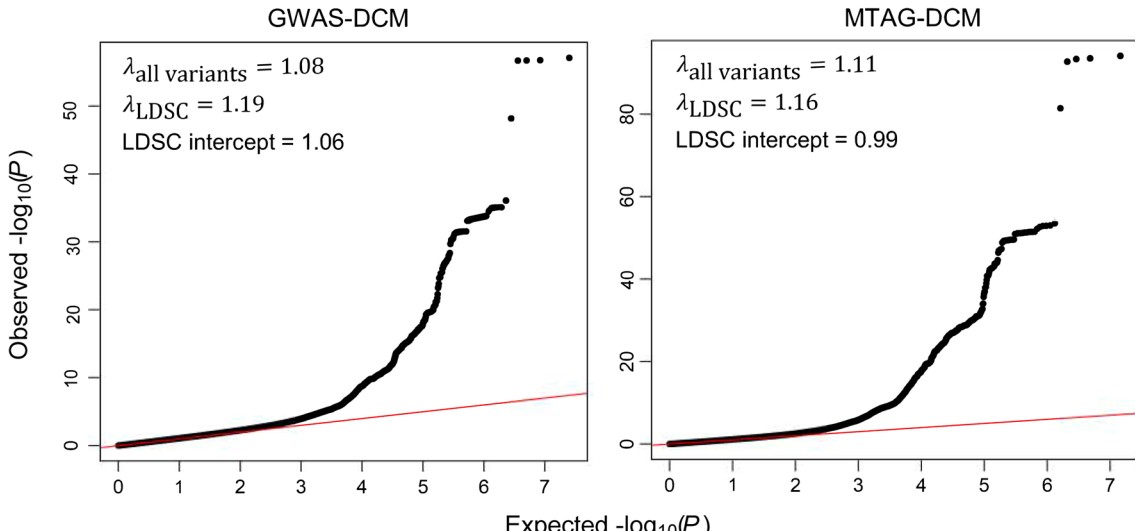

**Extended Data Fig. 2 | Quantile-quantile plots for the final meta-analysis (GWAS-DCM) and the final MTAG analysis (MTAG-DCM).** The quantile-quantile plots show results for the GWAS meta-analysis of DCM (left) and for the MTAG analysis of DCM with cardiac MRI traits (right). In each quantile-quantile plot, the x-axis represents the expected -log10 of the P-value of variants under the null hypothesis, while the y-axis represents the observed -log10 of the P-value in the analysis. The top corner shows calibration statistics, namely i) the inflation factor lambda, computed as the observed X^2 statistic at the median over the expected under the null, from all plotted variants, ii) the inflation factor computed by LDSC, which filters to high-confidence common genetic variants found in their internal reference, iii) the LDSC intercept which quantifies the residual inflation (computed as the intercept in a regression of X^2 statistics over linkage-disequilibrium scores[63]), due to biases. *P*-values are derived from inverse-variance-weighted meta-analysis of logistic regression models (GWAS-DCM) or from MTAG analysis of such results (MTAG-DCM); reported *P*-values are two-sided and unadjusted for multiple testing. Note: GWAS, genome-wide association study; MTAG, multi-trait analysis GWAS; DCM, dilated cardiomyopathy; LDSC, linkage-disequilibrium score regression.

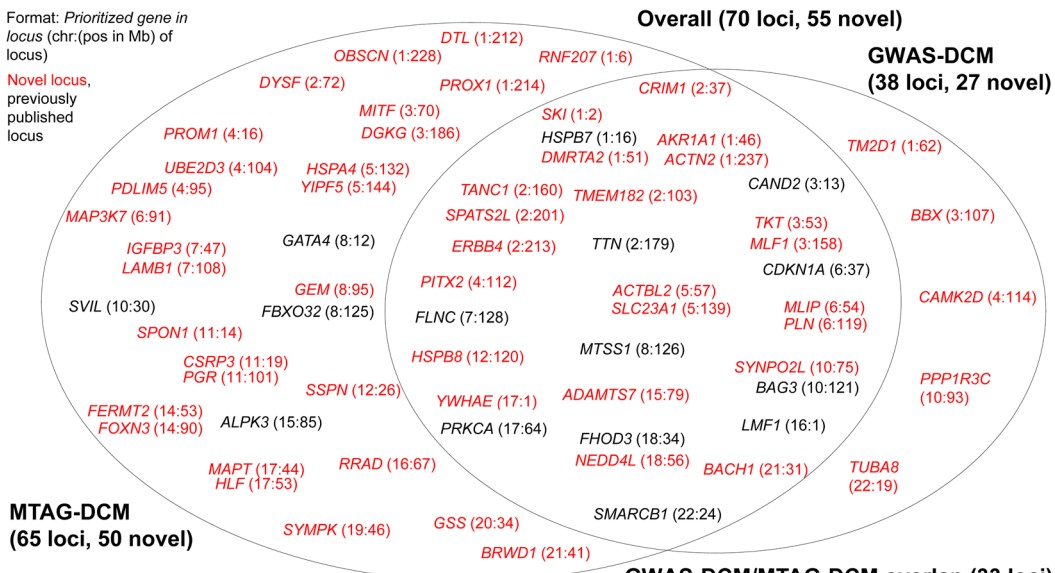

**Extended Data Fig. 3 | Venn Diagram highlighting the loci identified in GWAS-DCM and MTAG-DCM.** This Venn diagram shows loci that were significantly associated in GWAS-DCM, MTAG-DCM, or both. The right ellipse shows results from GWAS-DCM, the left ellipse shows results from MTAG-DCM, and the overlapping area shows loci found in both. A genomic locus was defined based on distance, taking the top index variant in a region, and merging with other potential index variants if within a 1 Mb window up or downstream (and merging MTAG-DCM and GWAS-DCM loci based on distance as well). Loci are annotated with the most highly-prioritized gene using our methodology

(Methods); in case of different genes prioritized by MTAG-DCM or GWAS-DCM (for overlapping loci), one was chosen at random for annotation. Loci are also annotated with the genomic coordinates (chromosome:position in megabases) for GRCh37. Loci annotated in red were 'novel', which was defined as: Not within 1 Mb distance with a previously described locus from a peer-reviewed published genome-wide association study for DCM, or MTAG for DCM, by querying GWAScatalog[85], OpenTargets[23] and two previous larger studies[8,11]. Note: GWAS, genome-wide association study; MTAG, multi-trait analysis GWAS; DCM, dilated cardiomyopathy.

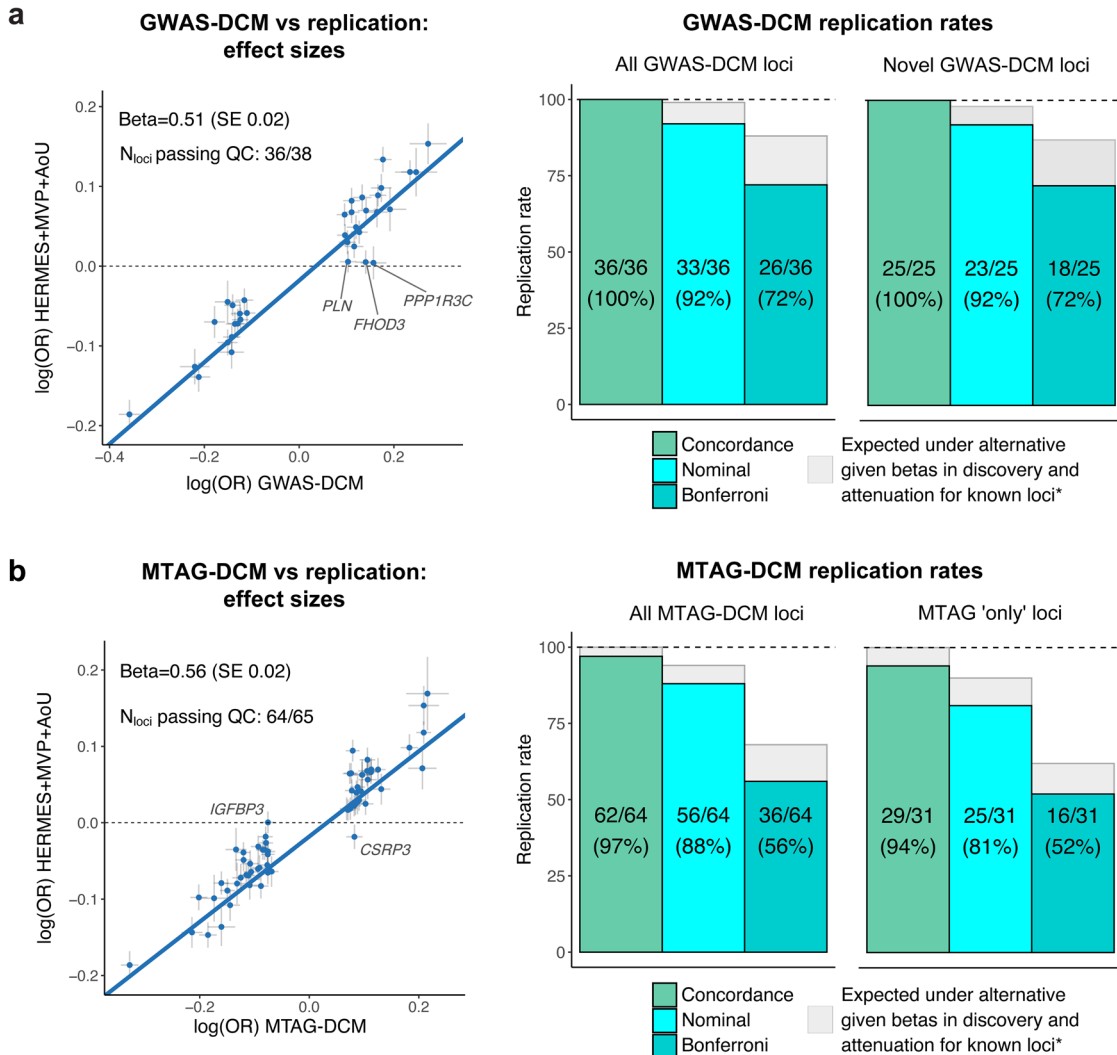

**Extended Data Fig. 4 | Independent replication of GWAS-DCM and MTAG-DCM loci.** The figure shows the summary of the replication effort performed within HERMES, the Million Veteran's Program (MVP) and All of Us (AoU) datasets. Part **a** shows the replication effort for GWAS-DCM loci, while part **b** shows results for replication of the MTAG-DCM loci. In both parts, data are restricted to loci passing quality-control for replication, and are restricted to a single lead variant per locus (the lead variant with strongest significance in discovery). The left panels show dot plots, with on the x-axis the effect sizes from discovery (ie, GWAS-DCM or MTAG-DCM) and on the y-axis the estimated effect size from the replication set (a meta-analysis of independent cohorts/samples from HERMES, MVP, and AoU), totalling up to 13,258 DCM/NICM cases and 1,435,287 controls (see Supplementary Note and Supplementary Tables 13 and 14). Data represent estimated beta coefficients ± standard errors. A trend line from linear regression is added to the plot, with the estimated beta coefficient and standard error from this regression added to the top left of the panels. Genes showing substantial deviation from the line are annotated with their gene names. The right panels represent bar charts that show the replication rate (ie, the percentage of replicating loci) using different definitions for replication; the green bars (left) represent directional concordance, the light blue bars (middle) represent replication at nominal unadjusted one-sided $P < 0.05$, while the dark blue (right) bars represent replication at Bonferroni-adjusted significance (one-sided $P < 0.05$/# loci) which leaves cutoffs of $P < 0.0014$ and $P < 0.002$ in part **a** and cutoffs of $P < 0.00078$ and $P < 0.0015$ in part **b**. Given the estimated attenuation of effect sizes for previously-established DCM loci, we computed 'expected' replication rates under the assumption that all loci are true and share the same degree of attenuation (Supplementary Note); the expected replication rates are added as light gray bars behind the colored bars. Note: OR, odds ratio.

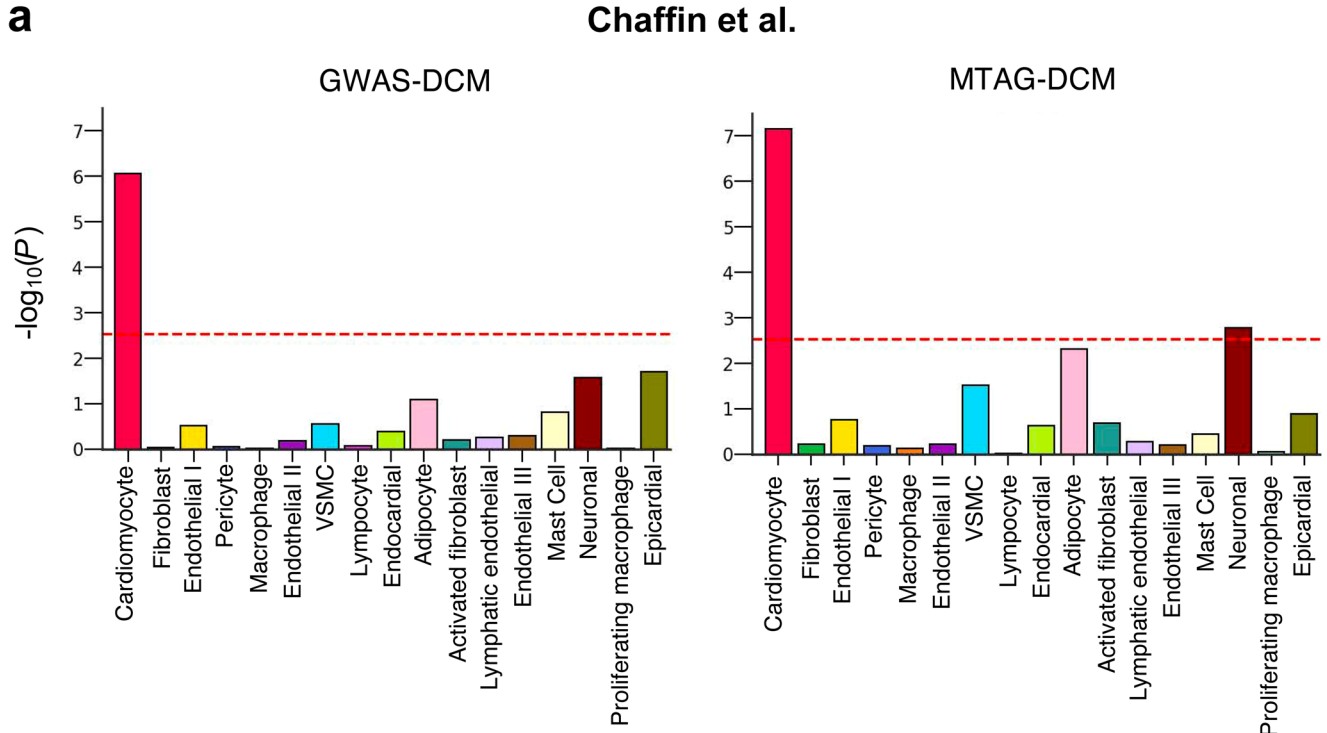

**a**

**Chaffin et al.**

GWAS-DCM

MTAG-DCM

**b**

**Reichart et al.**

GWAS-DCM

MTAG-DCM

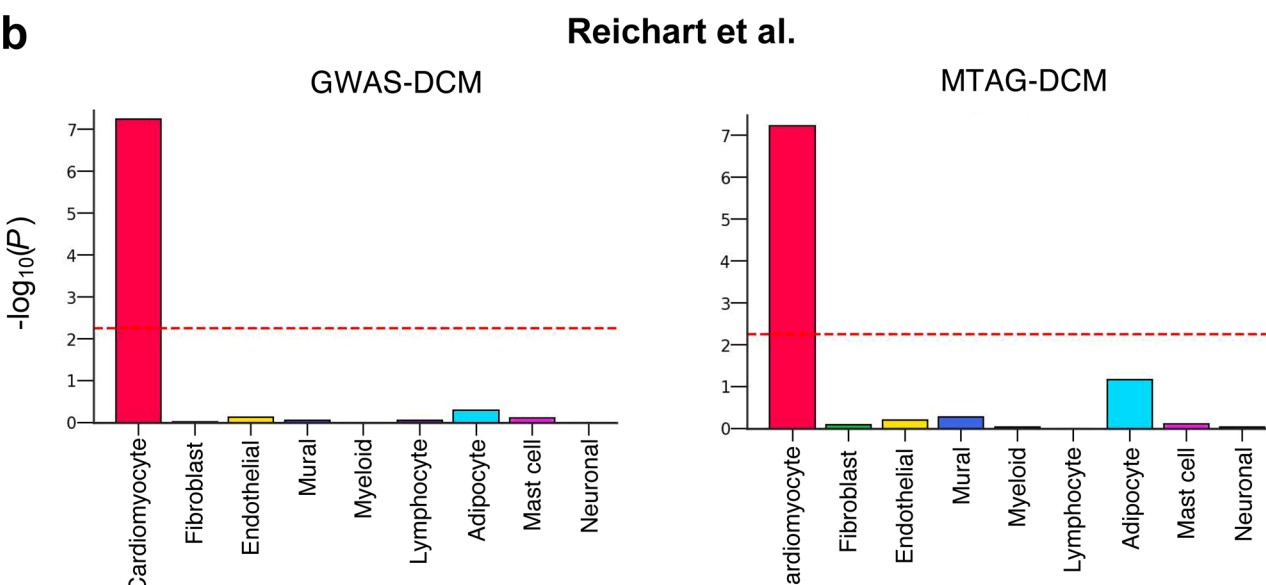

**Extended Data Fig. 5 | Cardiac cell type enrichment of DCM heritability from two snRNA sequencing datasets.** Bar charts represent the -log10 of the P-value from the analysis testing for enrichment of cell type-specific gene programs in our GWAS/MTAG results. The x-axis shows different cell types from the respective snRNAseq datasets. Part **a** shows results from enrichment analysis using the Chaffin et al.[20] snRNAseq dataset, while part **b** shows results for the Reichart et al.[21] snRNAseq dataset. The dotted lines represent the significance cutoffs within the panel, using a Bonferroni correction for the number of included cell types. The left panels show the results from testing for enrichment of GWAS-DCM heritability, while the right panels show results for testing for enrichment of MTAG-DCM heritability. P-values are derived from the Tau statistic from stratified LD score regression, and represent one-sided P-values that are unadjusted for multiple testing. Note: GWAS, genome-wide association study; MTAG, multi-trait analysis GWAS; DCM, dilated cardiomyopathy.

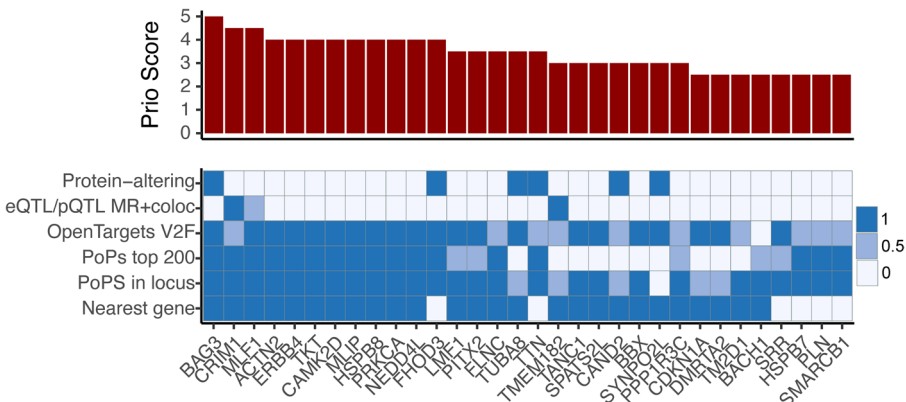

**Extended Data Fig. 6 | Gene prioritization scores for top prioritized genes from GWAS-DCM.** The bottom side of the figure shows a heatmap with different gene prioritization methods on the y-axis and highly-prioritized genes on the x-axis. The top side of the figure shows the corresponding gene prioritization scores, represented in bar charts, that show the sum of the individual components from the heatmap. Genes are ordered from left to right based on their priority score (high to low). In the heatmap, a very light blue panel indicates no points, a middle-blue panel indicates 0.5 points, while a dark blue panel indicates 1 point assigned to the given gene based on the given prioritization method. Highly-prioritized genes were defined as genes with 2.5 or higher points, which were also the most highly-prioritized genes in their respective loci. For the similar plot for MTAG-DCM, see Fig. 2b. Note: GWAS, genome-wide association study; DCM, dilated cardiomyopathy; MTAG, multi-trait analysis GWAS; PoPs, polygenic priority score method; eQTL, expression quantitative trait locus; pQTL, protein quantitative trait locus.

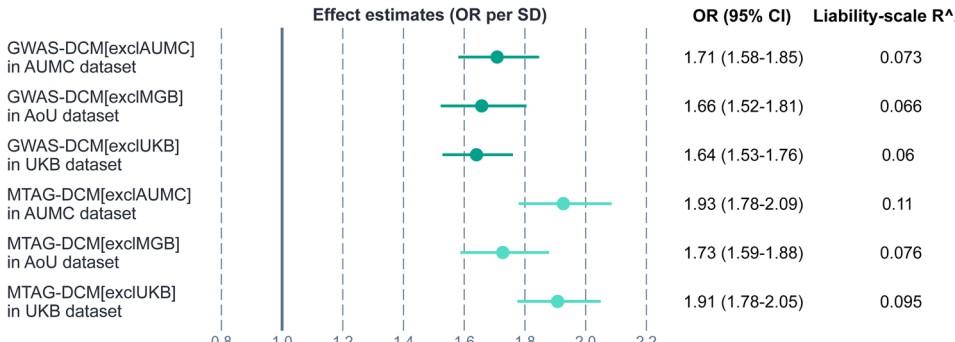

**Extended Data Fig. 7 | Associations between polygenic risk score and DCM across three European ancestry datasets.** This forest plot shows association results for the PRS constructed from GWAS-DCM and MTAG-DCM with DCM status across three different datasets. In all cases, association data are shown in a European ancestry 'testing set' (dataset in which PRS is tested) that is made as independent as possible from the 'training data' (ie, the base GWAS and MTAG data used to construct PRS). In the Amsterdam UMC (AUMC) dataset, AUMC samples were omitted from the PRS training data, and PRS was used to discriminate clinical DCM cases (N = 783) from referents (N = 6,978). In the *All of Us* (AoU) dataset, samples from Massachusetts General Hospital (MGB) were omitted from the PRS training data, and PRS was used to discriminate NI-DCM cases (N = 506) from controls (N = 95,510). In the UK Biobank (UKB) dataset, samples from UKB were omitted from the base GWAS, and participants were excluded from the testing set if they contributed to the MRI sub-study of UKB (first 45k); PRS was used to discriminate NI-DCM cases (N = 793) from controls (N = 325,313). All PRS were constructed using the PRScs algorithm

(Methods). In the plot, the *x*-axis shows odds ratios per standard deviation of the PRS distribution, estimated from logistic regression (adjusted at least for ancestral principal components in all cases). Data are presented as estimated odds ratios with 95% confidence intervals. The first three rows with dark green color show results for PRS constructed from GWAS-DCM, while the bottom three rows in light green color show results for PRS constructed from MTAG-DCM. On the right of the plot we show the R^2 for each PRS in the respective dataset, where R^2 represents the residual variance explained by the PRS (computed as the improvement of model R^2 inclusive of PRS as compared to the model without PRS, divided by the proportion of residual variance); all R^2 values were computed on the liability-scale to allow better comparisons across datasets. Note: Other performance metrics are presented in Supplementary Table 41. GWAS, genome-wide association study; DCM, dilated cardiomyopathy; NI-DCM, nonischemic dilated cardiomyopathy; MTAG, multi-trait analysis of GWAS; OR, odds ratio; 95%CI, 95% confidence interval; SD, standard error; R^2, variance explained.

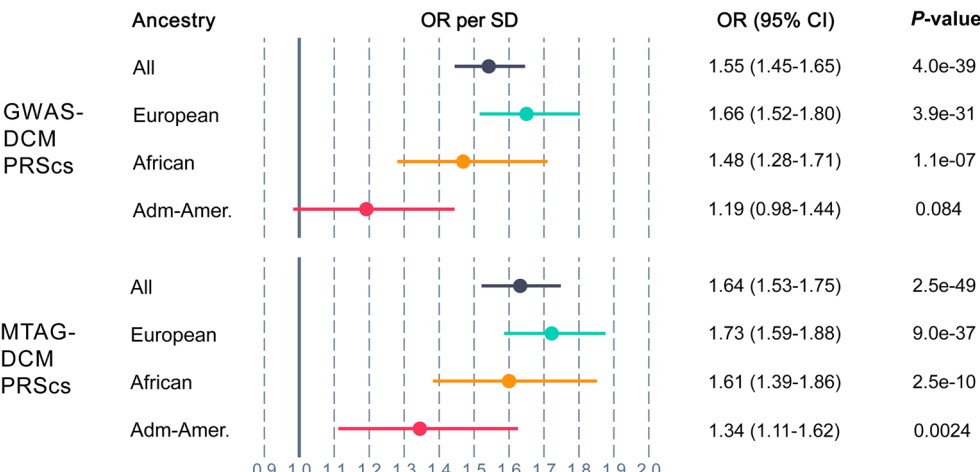

**Extended Data Fig. 8 | Associations between DCM polygenic risk score and NI-DCM across different ancestries in the *All of Us* dataset.** This forest plot shows association results for the PRS constructed from GWAS-DCM and MTAG-DCM with NI-DCM in the *All of Us* dataset. PRS were constructed using the PRScs algorithm (Methods), with x-axis showing odds ratios per standard deviation of the PRS distribution, estimated from logistic regression, adjusting for age, age^2, sex and ancestral principal components. Data are presented as estimated odds ratios with 95% confidence intervals. The figure shows results for all samples (N = 928 cases and 181,773 controls), European ancestry only (N = 506 cases and 95,510 controls), African ancestry only (N = 246 cases and 36,864 controls), and Admixed-American ancestry only (N = 107 cases and 28,784 controls). The top of the figure shows results for the PRS constructed from GWAS-DCM, while the bottom shows results for PRS constructed from MTAG-DCM. Reported *P*-values are two-sided and unadjusted for mutliple testing. Note: Other performance metrics are presented in Supplementary Table 32. GWAS, genome-wide association study; NI-DCM, nonischemic dilated cardiomyopathy; MTAG, multi-trait analysis of GWAS; OR, odds ratio; 95%CI, 95% confidence interval; SD, standard error.

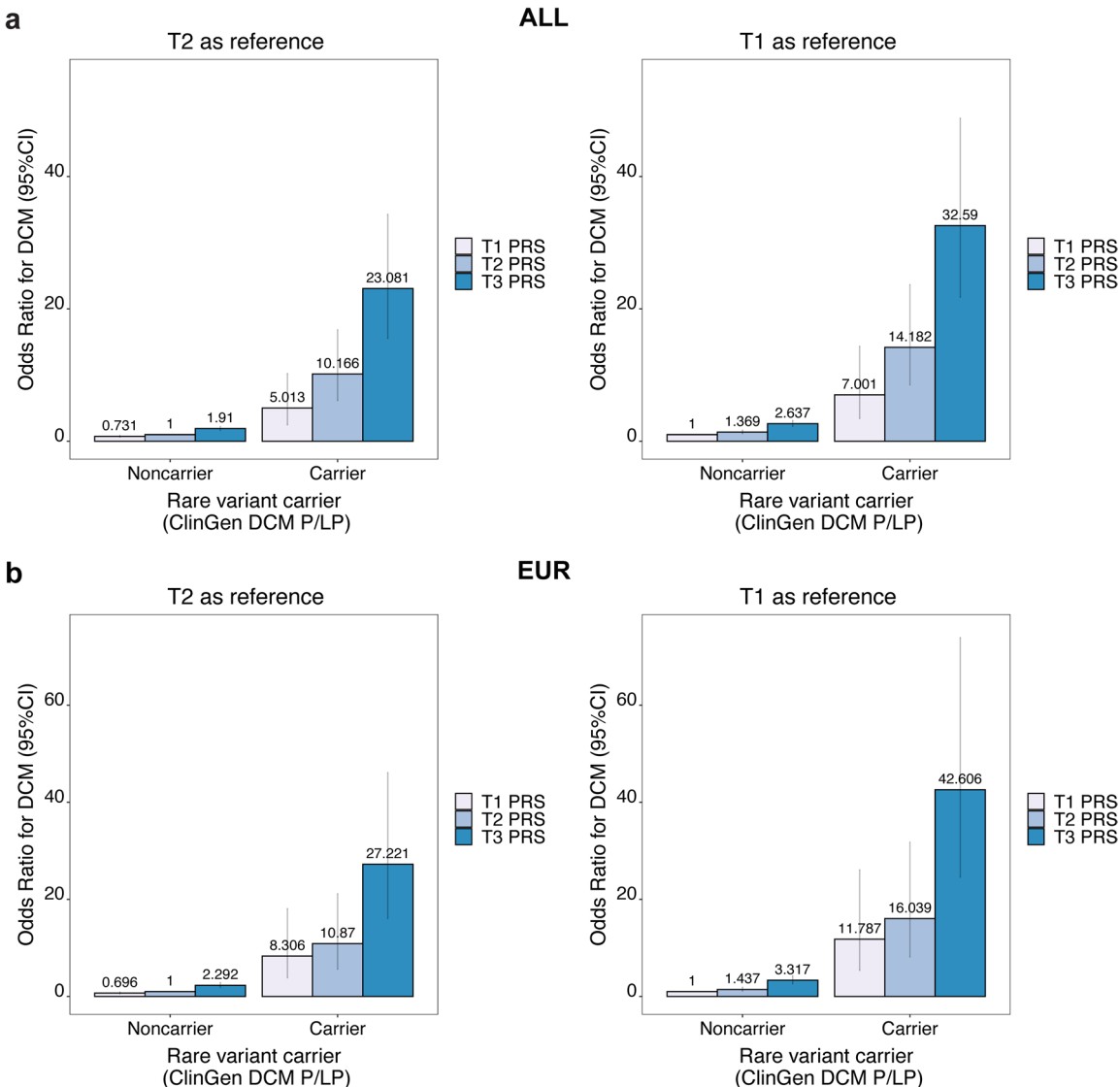

**Extended Data Fig. 9 | The additive contribution of PRS and rare pathogenic variants to NI-DCM risk in the *All of Us* dataset.** The figures show bar charts, where the x-axis shows different strata based on genetics, including three tertiles of PRS (tertile one [T1] in very-light blue, tertile two [T2] in light blue, and tertile three [T3] in dark blue) and two strata based on rare variant carrier status, that is non-carriers and carriers of rare pathogenic or likely pathogenic variants for DCM. The y-axis shows the estimated odds ratio for the given group as compared to a reference group; odds ratios were estimated using logistic regression analyses. Data are presented as estimated odds ratios with 95% confidence intervals. Part **a** shows results inclusive of all individuals passing our quality-control in *All of Us* (N = 928 cases and 181,773 controls), while part **b** is additionally restricted to samples with genetically-determined European ancestry (N = 506 cases and 95,510 controls). In both parts, the left panel shows results where the reference group is represented by individuals without rare variants in the second tertile of PRS; the right panel shows results where the reference group is represented by individuals without rare variants who are in the first tertile of PRS. Note: NI-DCM, nonischemic dilated cardiomyopathy; P/LP, likely pathogenic or pathogenic rare variants; CI, confidence interval; ALL, all individuals irrespective of ancestry; EUR, individuals of genetically-determined European ancestry.

# Reporting Summary

## Statistics

For all statistical analyses, confirm that the following items are present in the figure legend, table legend, main text, or Methods section.

| n/a | Confirmed | |
|---|---|---|
| ☐ | ☒ | The exact sample size (*n*) for each experimental group/condition, given as a discrete number and unit of measurement |
| ☐ | ☒ | A statement on whether measurements were taken from distinct samples or whether the same sample was measured repeatedly |
| ☐ | ☒ | The statistical test(s) used AND whether they are one- or two-sided<br>*Only common tests should be described solely by name; describe more complex techniques in the Methods section.* |
| ☐ | ☒ | A description of all covariates tested |
| ☐ | ☒ | A description of any assumptions or corrections, such as tests of normality and adjustment for multiple comparisons |
| ☐ | ☒ | A full description of the statistical parameters including central tendency (e.g. means) or other basic estimates (e.g. regression coefficient) AND variation (e.g. standard deviation) or associated estimates of uncertainty (e.g. confidence intervals) |
| ☐ | ☒ | For null hypothesis testing, the test statistic (e.g. $F$, $t$, $r$) with confidence intervals, effect sizes, degrees of freedom and $P$ value noted<br>*Give P values as exact values whenever suitable.* |
| ☒ | ☐ | For Bayesian analysis, information on the choice of priors and Markov chain Monte Carlo settings |
| ☒ | ☐ | For hierarchical and complex designs, identification of the appropriate level for tests and full reporting of outcomes |
| ☐ | ☒ | Estimates of effect sizes (e.g. Cohen's *d*, Pearson's *r*), indicating how they were calculated |

*Our web collection on statistics for biologists contains articles on many of the points above.*

## Software and code

Policy information about availability of computer code

**Data collection**

For the Meder et al. and Garnier et al. clinical cohorts, data collection has been described previously, and no new software was used for data collection specific to the present study.

For the Amsterdam dataset, DCM cases were collected by clinicians and researchers using standard in-house electronic health record environments (ie, EPIC). Cases and controls were genotyped on the Illumina Global Screening Array; standard pre-processing and variant calling were applied by third parties according to manufacturer instructions.

For the UK Biobank, FinnGen, MGB, All of Us, HERMES and MVP datasets, data collection and data pre-processing were performed centrally, and therefore no commercial software was needed to collect data specific to the present study.

**Data analysis**

Processing of genotype data, quality-control, imputation, and genome-wide association analyses were performed with various software tools as described in Supplementary Table 2. Notably, in most of the datasets, various versions of PLINK were used for quality control (https://www.cog-genomics.org/plink/ and https://www.cog-genomics.org/plink/2.0/) and various versions of REGENIE were used for GWAS (https://github.com/rgcgithub/regenie). Meta-analysis of GWAS was performed using the 2011-03-25 release of METAL (https://github.com/statgen/METAL). Heritability and genetic correlation parameters were computed using LDSC version version 1.0.1 (https://github.com/bulik/ldsc). Multi-trait analysis of GWAS was performed using MTAG version 1.0.8 (https://github.com/JonJala/mtag). For Mendelian randomization analyses, we used R-packages TwoSampleMR version 0.5.6 (https://mrcieu.github.io/TwoSampleMR/), coloc version 4.0.4 (https://github.com/chr1swallace/coloc), and CAUSE version 1.2.0 (https://github.com/jean997/cause/tree/master), implemented in custom MR pipelines (https://github.com/seanjosephjurgens/MR_pipeline_sjj). Annotation of GWAS was performed using FUMA version v1.6.1 (https://fuma.ctglab.nl/), as well as MAGMA version 1.10 (https://ctg.cncr.nl/software/MAGMA/prog/magma_v1.10.zip), and PoPS version 0.2 (https://github.com/FinucaneLab/pops). Gene set enrichment analyses were performed using FUMA version 1.6.1 (https://fuma.ctglab.nl/) and g:Profiler version September 20 2023 (https://biit.cs.ut.ee/gprofiler/). For cell type specific heritability analyses, we used R-packages edgeR version 3.22.3 (https://github.com/OliverVoogd/edgeR), DESeq2 version 1.20.0 (https://github.com/thelovelab/DESeq2), and limma version 3.36.2 (https://bioconductor.org/packages/release/bioc/html/limma.html), as well as stratified LDSC version 1.0.1 (https://

github.com/bulik/ldsc). For wrangling of single cell/nucleus data, we used R-package Seurat version 5.0 (https://github.com/satijalab/seurat). For polygenic scoring analyses, we used PRScs version 2022-11 (https://github.com/getian107/PRScs) and PLINK2 (https://www.cog-genomics.org/plink/2.0/; various versions from May 2020 release onwards). All analyses that were run in R, were run in R version 4.0.0.

For manuscripts utilizing custom algorithms or software that are central to the research but not yet described in published literature, software must be made available to editors and reviewers. We strongly encourage code deposition in a community repository (e.g. GitHub). See the Nature Portfolio guidelines for submitting code & software for further information.

## Data

Policy information about availability of data

All manuscripts must include a data availability statement. This statement should provide the following information, where applicable:
- Accession codes, unique identifiers, or web links for publicly available datasets
- A description of any restrictions on data availability
- For clinical datasets or third party data, please ensure that the statement adheres to our policy

Summary statistics for our GWAS meta-analyses have been made available for download through the Cardiovascular Disease Knowledge Portal (https://cvd.hugeamp.org/downloads.html); summary statistics for various meta-analyses, including clinical dataset-only and biobank dataset-only, are available (download link: https://api.kpndataregistry.org/api/d/CQyqth). Our PRS scoring weights - for both GWAS and MTAG scores – have been deposited into the PGS Catalog (publication ID: PGP000672; score IDs: PGS004946- PGS004951) and into the Cardiovascular Disease Knowledge Portal (download link: https://api.kpndataregistry.org/api/d/9jevLe). Access to individual-level data for the Meder et al. cohort, the Garnier et al. cohort, the Amsterdam UMC cohort, and MGB will not be made publicly available at this time, due to the restrictive/sensitive nature of the genomic and/or phenotypic data in question. Access to individual level UK Biobank data, both phenotypic and genetic, is available to bona fide researchers through application on the UK Biobank website (https://www.ukbiobank.ac.uk). Access to individual-level phenotypic and genetic data from All of Us Research Program is currently available to bona fide researchers within the United States through the All of Us Researcher Workbench, a cloud-based computing platform (https://www.researchallofus.org/register/). The Finnish biobank data can be accessed through the Fingenious® services (https://site.fingenious.fi/en/) managed by FINBB. Finnish Health register data can be applied for from Findata (https://findata.fi/en/data/). All processed snRNAseq/scRNAseq datasets used in the present study are publicly available: The Chaffin et al. dataset is available for download from the Broad Single Cell Portal (https://singlecell.broadinstitute.org/single_cell/study/SCP1303/single-nuclei-profiling-of-human-dilated-and-hypertrophic-cardiomyopathy); the Reichart et al. dataset was downloaded from GEO (https://www.ncbi.nlm.nih.gov/geo/download/?acc=GSE183852&format=file&file=GSE183852%5FDCM%5FIntegrated%2ERobj%2Egz); the Koenig et al. dataset was downloaded from CellxGene (https://datasets.cellxgene.cziscience.com/3716fb19-cedd-4fe5-abc4-5dbeb007fb65.rds).

Other datasets include cis-eQTLs from the eQTLGen consortium (https://www.eqtlgen.org/cis-eqtls.html); cis-eQTLs from GTEx v8 (https://www.gtexportal.org/home/downloads/adult-gtex#qtl) and tissue expression levels from GTEx v8 (https://www.gtexportal.org/home/downloads/adult-gtex#bulk_tissue_expression); pQTLs derived from the UK Biobank Pharma Proteomics Project (summary statistics for the 'combined' set from https://www.synapse.org/#!Synapse:syn51364943/files/); the 22.10 update of the OpenTargets platform (https://genetics.opentargets.org/); GWAS Catalog queried in October 2023 (https://www.ebi.ac.uk/gwas/); ANNOVAR v2017-07-17 (https://annovar.openbioinformatics.org/en/latest/); 1000 Genomes project Phase 3 (https://www.internationalgenome.org/data/); gnomAD exomes v.2.1 (https://gnomad.broadinstitute.org/downloads); the ClinVar database (https://www.ncbi.nlm.nih.gov/clinvar/) was accessed in April 2023.

# Field-specific reporting

Please select the one below that is the best fit for your research. If you are not sure, read the appropriate sections before making your selection.

☒ Life sciences   ☐ Behavioural & social sciences   ☐ Ecological, evolutionary & environmental sciences

For a reference copy of the document with all sections, see nature.com/documents/nr-reporting-summary-flat.pdf

# Life sciences study design

All studies must disclose on these points even when the disclosure is negative.

| Sample size | For GWAS discovery and replication analyses, sample sizes were based on the number of samples for which phenotypic and genetic data were available in the datasets at the respective time points, with a clear aim of maximizing effective sample size. No power calculations were performed to pre-determine the required sample size. |
|---|---|
| Data exclusions | For the discovery datasets, sample exclusions are described for each cohort in detail in Supplementary Table 2. In all discovery datasets, steps were performed to identify samples with bad genetic data (ie, removal of samples with high missingness). In the biobank datasets, samples with general heart failure codes (but not fulfilling case definitions) were removed from the controls in GWAS. In all clinical case-control datasets, exclusions were applied based on imaging data, while in some datasets additional exclusions were applied based on additional disease status. Details can be found in Supplementary Table 2 and Supplementary Note.<br><br>For the All of Us WGS dataset, most QC was performed centrally and consisted of per-sample QC, including fingerprint concordance (array vs. WGS data), sex concordance (genetically determined vs. self-reported), cross-individual contamination rate and coverage to detect major errors, such as sample swaps or contamination. Participants who failed these tests were removed from the release. We removed flagged participants (population outliers) and possible duplicates from the current study.<br><br>For the MVP replication cohort, exclusions and sample selections are described in the Supplementary Note. For the HERMES datasets, sample selections and exclusions were performed previously, as described in the work of Zheng et al. (2024). |
| Replication | We performed a replication analysis using a meta-analysis of three datasets, namely All of Us, HERMES, and MVP. This replication meta-analysis included 13,258 DCM/NICM cases and 1,435,287 controls. This replication showed good overall replication, with 92% of GWAS-DCM |

loci reaching P<0.05 and 72% reaching Bonferroni corrected significance; of MTAG-DCM loci, 88% reached P<0.05 and 56% reached Bonferroni-corrected significance. These results show a strong replication trend in our significant loci.

**Randomization**      Samples were not experimentally randomized, given that the exposure in our analysis is genetic variation.

**Blinding**      No blinding was performed during analysis of the data. We note that the main discovery analyses represented genome-wide association tests, where all variants reaching quality-control criteria were put forward for meta-analysis and were tested for association. As such, while blinding was not formally applied, the approach should not have been affected by the absence of a formal blinding procedure.

# Reporting for specific materials, systems and methods

We require information from authors about some types of materials, experimental systems and methods used in many studies. Here, indicate whether each material, system or method listed is relevant to your study. If you are not sure if a list item applies to your research, read the appropriate section before selecting a response.

## Materials & experimental systems

| n/a | Involved in the study |
|---|---|
| ☒ | ☐ Antibodies |
| ☒ | ☐ Eukaryotic cell lines |
| ☒ | ☐ Palaeontology and archaeology |
| ☒ | ☐ Animals and other organisms |
| ☐ | ☒ Human research participants |
| ☒ | ☐ Clinical data |
| ☒ | ☐ Dual use research of concern |

## Methods

| n/a | Involved in the study |
|---|---|
| ☒ | ☐ ChIP-seq |
| ☒ | ☐ Flow cytometry |
| ☒ | ☐ MRI-based neuroimaging |

# Human research participants

Policy information about studies involving human research participants

**Population characteristics**

In the Meder et al. cohort, all samples were of German-White descent. Among the 909 DCM cases, 25.2% had female genetic sex, the mean age was 56.6 years (SD=12.9), and the mean left ventricular ejection fraction (LVEF) was 28.5 (SD=10.9). Among 2120 controls, 49.7% were of female genetic sex and the mean age was 57.4 (SD=14.1).

The Garnier et al. dataset consisted of samples of European genetically-determined ancestry. Several sub-cohorts comprised the dataset:
French Cardigene cases (N=408): mean age 45.1 (SD=10.7), mean LVEF 24.0 (SD=8.2), mean LVEDD 74.2 mm (SD=9.4), N with heart transplant 212.
French PHRC cases (N=204): mean age 52.0 (SD=13.0), mean LVEF 28.2 (SD=8.9), mean LVEDD 68.5 mm (SD=9.0), N with heart transplant 0.
French Eurogene cases (N=83): mean age 46.9 (SD=13.2), mean LVEF 29.2 (SD=10.2), mean LVEDD 67.5 mm (SD=8.1), N with heart transplant 0.
Italy Eurogene cases (N=82): mean age 43.1 (SD=13.4), mean LVEF 27.2 (SD=7.9), mean LVEDD 66.5 mm (SD=8.5), N with heart transplant 0.
German Eurogene cases (N=214): mean age 45.9 (SD=11.7), mean LVEF 30.3 (SD=11.3), mean LVEDD 69.8 mm (SD=9.7), N with heart transplant 0.
Germany Berlin cases (N=987): mean age 44.0 (SD=11.6), mean LVEF 24.1 (SD=9.7), mean LVEDD 68 mm (SD=10.0).
UK Royal Brompton (N=109): mean age 54.9 (SD=13.4), mean LVEF 30 (SD=8.7), mean LVEDV 347.9 ml (SD=132.6).
US MAGNet cases (N=631): mean age 51.1 (SD=14.2), mean LVEF 20.3 (SD=9.9), mean LVEDD 58.5 (SD=22.3), N with heart transplant 328.
French controls (N=1084) were sourced from the PPS3 study, including 731 with male genetic sex and 353 with female genetic sex, with mean age 62 (SD=6.4).
German controls (N=3,264) were sourced from the population-based KORA F4 study, including 1579 with male genetic sex and 1685 with female genetic sex, with mean age 57.4 (SD=12.9).
Italian controls (N=92) were collected as part of the EHF study, including 70 with male genetic sex and 22 with female genetic sex, with mean age 49.8 (SD=12.7).

The Amsterdam dataset consisted of cases (N=978) from Amsterdam UMC, of which 560 had male genetic sex (57.3%), 783 were of genetically-determined European ancestry (80.6%), with a median LVEF of 28% (Q1,Q3: [20.00, 37.00]), and with median age of 57 years (Q1,Q3: [48.00, 65.00]). Of controls sourced from the Dutch Twin registry (N=7207), 6978 were of genetically-determined European ancestry, and 3172 (44.0%) had male genetic sex.

In the UK Biobank (N=472474), 1065 individuals were identified as cases for nonischemic DCM. 95% of individuals were of genetically-determined European ancestry (94% of cases, 95% of controls), 45% were of male genetic sex (69% of cases, 45% of controls), and mean enrollment age was 57 years (SD=8); of cases mean enrollment age was 60 (SD=7) years, and of controls mean enrollment age was 57 (SD=8) years.

In FinnGen (N=422920), 3550 individuals were identified as cases for nonischemic DCM. 100% of individuals were of genetically-determined Finnish ancestry, 43% were of male genetic sex (74% of cases, 42% of controls), and mean enrollment age was 52 years (SD=18); of cases mean enrollment age was 61 (SD=13) years, and of controls mean enrollment age was 52 (SD=18) years.

In MGB (N=42637), 407 individuals were identified as cases for nonischemic DCM. 84% of individuals were of genetically-determined European ancestry, 42% were of male genetic sex (63% of cases, 41% of controls), and mean enrollment age was 51 years (SD=17); of cases mean enrollment age was 57 (SD=15) years, and of controls mean enrollment age was 51 (SD=17) years.

The All of Us dataset (N= 195533), is an ancestrally-diverse dataset (53% European genetic ancestry, 21% African, 16% Admixed-American ancestry, 1.9% East-Asian ancestry, 0.92% South-Asian ancestry, 0.22% Middle-Eastern ancestry, 8.0% other ancestry), with 75649 (38.7%) being of male genetic sex, and with mean age of 52.4 (SD=16.8). 928 (0.47%) were cases for nonischemic DCM, while 5123 (2.6%) were cases for broad systolic heart failure.

For details on the MVP and HERMES (sub-)cohorts used in replication, please see the Supplementary Note.

| Recruitment | In the Meder et al. cohort, cases were ascertained from clinical (cardiological) medical centers in/near Heidelberg, Germany. Controls were sourced from existing population-based reference cohorts. |
|---|---|

In the Garnier et al. dataset, all cases were ascertained from clinical (cardiological) medical centers. French controls from PPS3 were ascertained during preventative medical check-ups. German controls from KORA F4 were ascertained from population-based surveys. Italian controls from EHF were healthy health-care workers who were approached to partake in the research work.

In the Amsterdam cohort, cases were ascertained from mining of medical records at the Amsterdam UMC, and by manual review of charts; all patients were referred for genetic testing for DCM within the Amsterdam UMC (a medical center). Controls were ascertained from the Dutch Twin Register; since 1987, this register has been accumulating information on twins and triplets, either when the parents of newborn twins voluntarily register or when adult twins and their family participate.

For UK Biobank, prospective participants were invited to visit an assessment centre, at which they completed an automated questionnaire and were interviewed about lifestyle, medical history and nutritional habits; basic variables such weight, height, blood pressure etc. were measured; and blood and urine samples were taken. These samples were preserved so that it was possible to later extract DNA and measure other biologically important substances. During the whole duration of the study it was intended that all disease events, drug prescriptions and deaths of the participants are recorded in a database, taking advantage of the centralized UK National Health Service.

The FinnGen study (https://www.finngen.fi/en) is an ongoing research project that utilizes samples from a nationwide network of Finnish biobanks and digital healthcare data from national health registers. FinnGen aims to produce genomic data with linkage to health register data of 500,000 biobank participants. Samples in the FinnGen study include ~200,000 legacy samples from previous research cohorts (often disease-specific) that have been transferred to the Finnish biobanks, and ~300,000 prospective samples collected by biobanks across Finland. Prospective samples from six regional hospital biobanks represent a wide variety of patients enrolled in specialized health care, samples from a private healthcare biobank enable enrichment of the FinnGen cohort with patients underrepresented in specialized health care, whereas participants recruited through the Blood Service Biobank enrich the cohort with healthier individuals. Samples have not specifically been collected for FinnGen, but the study has incorporated all that have been available in the biobanks.

For MGB (formerly known as Partners Biobank) samples were prospectively recruited - in an ongoing observational design - from a multicenter health system in Eastern Massachusetts. In MGB, participants are enrolled with broad-based consent collected by local research coordinators, either as part of a collaborative research study or electronically through a patient portal. Demographic data, blood samples and surveys are collected at baseline and linked to electronic health record data.

For All of Us, samples were enrolled in a longitudinal cohort study (with aim of including 1 million racially, ancestrally and demographically diverse participants) from across the United States. Data is prospectively collected, combining phenotypic data from various sources including patient-derived information and electronic health record linkage. One of the goals set by All of Us was to recruit individuals that have been and continue to be underrepresented in biomedical research because of limited access to health care.

For details on the MVP and HERMES (sub-)cohorts used in replication, please see the Supplementary Note.

| Ethics oversight | For the Meder et al. dataset, the study was conducted in accordance with the principles of the Declaration of Helsinki. All participants of the study have given written informed consent and the study was approved by the ethic committees of the participating study centers. |
|---|---|

For the Garnier et al. dataset, the study protocol was approved by local ethics committees, complied with the Declaration of Helsinki, and all patients signed informed consent.

For the Amsterdam cohort, study of DCM patients from Amsterdam UMC was performed under a waiver - approved by the Medical Ethical Committee of Amsterdam UMC - allowing genotyping and genome-wide association study of individuals affected by cardiovascular disease. The study protocol for GWAS of inherited cardiovascular disease was approved by the local Medical Ethical Review Committee of Amsterdam UMC. For the controls sourced from the Dutch Twin register, Ethical clearance has been granted by the Central Ethics Committee on Research Involving Human Subjects at the VU University Medical Centre in Amsterdam, which is an Institutional Review Board certified by the U.S. Office of Human Research Protections. The approval carries the IRB number IRB-2991 under Federal-wide Assurance-3703 and includes specific institute codes (94/105, 96/205, 99/068, 2003/182, 2010/359). All control participants, or their parents, have given their informed consent to be part of the register.

The UK Biobank resource was approved by the UK Biobank Research Ethics Committee and all participants provided written

informed consent to participate. Use of UKB data was performed under application number 17488 and was approved by the local Massachusetts General Hospital Institutional Review Board.

Participants in FinnGen provided informed consent for biobank research on basis of the Finnish Biobank Act. Alternatively, separate research cohorts, collected before the Finnish Biobank Act came into effect (in September 2013) and the start of FinnGen (August 2017) were collected on the basis of study-specific consent and later transferred to the Finnish biobanks after approval by Fimea, the National Supervisory Authority for Welfare and Health. Recruitment protocols followed the biobank protocols approved by Fimea. The Coordinating Ethics Committee of the Hospital District of Helsinki and Uusimaa (HUS) approved the FinnGen study protocol (number HUS/990/2017). The FinnGen study is approved by the THL (approval number THL/2031/6.02.00/2017, amendments THL/1101/5.05.00/2017, THL/341/6.02.00/2018, THL/2222/6.02.00/2018, THL/283/6.02.00/2019 and THL/1721/5.05.00/2019), the Digital and Population Data Service Agency (VRK43431/2017-3, VRK/6909/2018-3 and VRK/4415/2019-3), the Social Insurance Institution (KELA) (KELA 58/522/2017, KELA 131/522/2018, KELA 70/522/2019 and KELA 98/522/2019) and Statistics Finland (TK-53-1041-17).

Use of All of Us data was approved under a data use agreement between the Massachusetts General Hospital and the All of Us Research Program program.

For MGB, all adult patients provided informed consent to participate. A small number of children were enrolled with IRB-approved assent forms; upon reaching 18 years of age all enrolled children had to provide consent or were removed from the study. The Human Research Committee of MGB approved the Biobank protocol (2009P002312).

All MVP and HERMES (sub-)cohorts collected appropriate participant consents, and protocol approvals where appropriate.

Note that full information on the approval of the study protocol must also be provided in the manuscript.

