## [Peer Review File · Nature Genetics]

Peer Review Information

Manuscript Title: Genome-wide association study reveals mechanisms underlying dilated cardiomyopathy and myocardial resilience

Corresponding author name(s): Professor Connie (R.) Bezzina, Dr Patrick (T) Ellinor, Dr Mark (J.) Daly, Dr Krishna (G) Aragam

Reviewer Comments & Decisions:

Decision Letter, initial version:

13th Feb 2024

Dear Professor Bezzina,

Your Letter, "Genome-wide association study reveals mechanisms underlying dilated cardiomyopathy and myocardial resilience" has now been seen by 3 referees. You will see from their comments below that while they find your work of interest, some important points are raised. We are interested in the possibility of publishing your study in Nature Genetics, but would like to consider your response to these concerns in the form of a revised manuscript before we make a final decision on publication.

To guide the scope of the revisions, the editors discuss the referee reports in detail within the team with a view to identifying key priorities that should be addressed in revision. In this case, we think all three referees have provided constructive reviews aimed at strengthening the analyses and improving the presentation. We particularly ask that you perform independent replication, clarify the consistencies and novel aspects of the study compared to ref.19, and address all referees' technical comments as thoroughly as possible with appropriate revisions. In addition, it would be very helpful to provide summary association statistics excluding widely used databases. We hope that you will find the prioritized set of referee points to be useful when revising your study.

We therefore invite you to revise your manuscript taking into account all reviewer and editor comments. Please highlight all changes in the manuscript text file. At this stage we will need you to upload a copy of the manuscript in MS Word .docx or similar editable format.

*2) If you have not done so already please begin to revise your manuscript so that it conforms to our Letter format instructions, available here.
Refer also to any guidelines provided in this letter.

Please be aware of our guidelines on digital image standards.

[redacted]

We hope to receive your revised manuscript within 3 to 6 months. If you cannot send it within this time, please let us know.

Sincerely,
Wei

Wei Li, PhD
Senior Editor

Nature Genetics
New York, NY 10004, USA
www.nature.com/ng

Reviewers' Comments:

Reviewer #1:

Remarks to the Author:

Jurgens et al present a comprehensive GWAS of dilated cardiomyopathy (DCM). They identify novel loci through a meta analysis of data from ~9000 cases and close to one million controls. These results form the basis for a series of downstream analyses: gene prioritization, analysis of cell type specificity of these candidate genes, Mendelian randomization (MR) analysis of other traits and polygenic risk score (PRS) analysis. The study represents one of the largest GWAS of DCM to date. There is however a preprint (ref 19) on a similar effort with ~14000 DCM cases. Especially the MR and PRS analysis provide appealing new results on likely causal pathways that go beyond previous studies. Overall the study has been executed rigorously using state of the art methods and the presentation is very clear.

A major issue that needs to be addressed is replication in an independent cohort. The authors comment that they could replicate 20 of 26 DCM loci identified in a similarly large study (ref 19). What about the novel loci, can they be replicated using results from ref 19? Could other data sets such as that of the all of us program be used for replication (it is not yet part of the GWAS but it has phenotype information used in the PRS analysis)? It seems that the same imaging based traits that were analyzed in the current study were also analyzed in ref 19. What is the replication rate there? In addition, please assess whether there is heterogeneity of the effect sizes between the different study cohorts or if they show large agreement.

A second important question is how the results in the current study more generally compare to those of ref 19, given that very similar approaches were applied. How do gene prioritizations agree or differ? Do both studies highlight the same or partly different cell types as major contributors? How do the PRS compare to each other?

Minor comments:

The eQTL and pQTL analysis is very interesting. However the pQTL data that was used is from blood. The authors show that cardiac tissue and specifically cardiomyocytes are key players. Please use heart specific pqtL data (e.g. Assum et al. Nat Comms 2022) for this analysis. It would also be interesting to characterize the overlap of eQTL and pQTL results. What is the agreement between heart and blood?

The comparison of the genetic associations of the strict and more lenient DCM phenotype definition is interesting. What is the genetic correlation of the two traits? Is there more variance in the lenient definition?

How do the single cell results compare to those of Reichart et al Science 2022?

An interaction analysis of rare variant status (possibly aggregated as genotype +, as currently done) and PRS would be highly interesting.

To what extent could the finding of cardiomyocytes as main contributors be a consequence of the gene prioritization scheme? Specifically, the pops method is highly weighted and relies strongly on expression. As many important loci are cardiomyocyte specific this might lead to a self fulfilling prophecy.

Reviewer #2:

Remarks to the Author:

The report by Ellinor, Daly, Aragam and Bezzina describes results of a large DCM GWAS meta-analysis. A large number of novel loci are described and the authors include a broad set of secondary analyses. The analyses are well-done and the paper is a pleasure to read. I do have some comments:

- 1) Can the authors provide a pheWAS analysis of their PRS and lead variants? The specificity of the associations is not currently explored.
- 2) How does the PRS compares to carrying a bona fide DCM Mendelian mutation in terms of DCM risk?

Reviewer #3:

Remarks to the Author:

The authors performed a large-scale genome-wide association study (GWAS) and multi-trait analysis (MTAG) for dilated cardiomyopathy (DCM). Using 9,365 DCM cases and 946,368 controls the study is about twice as large as previous GWAS. Expectedly, the number of loci showing genome-wide significance increased. Further analyses highlight the role of the contractile apparatus in the pathogenesis of DCM and mendelian randomization analyses showing that DCM liability is associated with an increased risk of systolic heart failure in context of other cardiovascular conditions.

The analyses were conducted with great care, the methodology is sound (as far as I can tell not being a bioinformatician) and the paper is written very well. The conclusions appear to be justified.

Major comment

The combination of the DCM GWAS with the multi-trait analysis is not entirely transparent. Specifically, MRI-based measurements of end-systolic volume (LVESV) may reflect body size rather than pathological dilatation of the LV.

This is of relevance, since the previous studies (with overlapping data sets) already provided evidence for 42 DCM loci (line 157). This number is similar to the 38 GWAS loci for DCM reported here. Thus, it appears that the major novelty comes from the 65 loci identified by MTAG, i.e. by GWAS for global circumferential strain (Ecc) and LVESV. However, GWAS on structural and functional evaluations of MRI data have been reported before and is unclear whether there is (substantial) overlap with previous reports.

- The authors should clarify by which extent the MTAG loci provide at least a Bonferroni-corrected

significant signals for DCM, e.g. <0.0007 . If some loci offer no signal for DCM, it is questionable whether they reflect LV dysfunction and can be meta-analysed with DCM in a meaningful way.

- The authors should be more distinct on the number of novel loci, since GWAS for structural and functional evaluations of MRI data have been published before. I.e., loci reported for genome-wide association with functional and structural cardiac phenotypes (PMID: 32382064) or heart failure (e.g. PMID: 36376295) before should not be declared as novel here.

The reported association between mean platelet thrombocyte volume and DCM is somewhat unexpected and not further discussed by the authors. It would be interesting to have some background information (or a statement that it may be a false positive association).

Author Rebuttal to Initial comments

Reviewer 1

Jurgens et al present a comprehensive GWAS of dilated cardiomyopath (DCM). They identify novel loci through a meta analysis of data from ~9000 cases and close to one million controls. These results form the basis for a series of downstream analyses: gene prioritization, analysis of cell type specificity of these candidate genes, Mendelian randomization (MR) analysis of other traits and polygenic risk score (PRS) analysis. The study represents one of the largest GWAS of DCM to date. There is however a preprint (ref 19) on a similar effort with ~14000 DCM cases. Especially the MR and PRS analysis provide appealing new results on likely causal pathways that go beyond previous studies. Overall the study has been executed rigorously using state of the art methods and the presentation is very clear.

We thank the Reviewer for their thorough and positive assessment of our work.

A major issue that needs to be addressed is replication in an independent cohort. The authors comment that they could replicate 20 of 26 DCM loci identified in a similarly large study (ref 19). What about the novel loci, can they be replicated using results from ref 19? Could other data sets such as that of the all of us program be used for replication (it is not yet part of the GWAS but it has phenotype information used in the PRS analysis)? It seems that the same imaging based traits that were analyzed in the current study were also analyzed in ref 19. What is the replication rate there?

The Reviewer makes a series of valid suggestions to improve our work using (cross)replication with other datasets. We agree with the Reviewer that true

independent replication of our findings would represent a substantial improvement to our study. As suggested, we therefore sought independent replication using other GWAS datasets. We used data from independent cohorts from Zheng et al. (HERMES; N=~8.4k cases; (Zheng et al. 2023)), data from the All of Us dataset (AoU; N=817 cases), and samples from the Million Veteran's Program (MVP; N=3.9k cases).

In the meta-analysis of these replication datasets, we retained up to 13,258 cases and 1,435,287 controls that were independent of our discovery. While large in number, we note that a substantial number of samples from HERMES represent a more broad case definition (similar to NICM). Consistent with our heritability analyses, effect sizes were somewhat attenuated in this broader case set (attenuated to ~0.50-0.56 of the discovery effect sizes, when looking only at previously-established DCM loci).

For our replication, we then focused only on the lead variant in each locus that was most significant in discovery, and then retained lead variants with MAF \geq 1% and with >1000 cases contributing to the replication analysis.

For our GWAS-DCM loci, we could perform replication analyses for 36 of 38 loci. We found that all loci were concordant in direction of effect (100%), 33 loci were significant at a nominal level (92%), and 26 loci were significant at a Bonferroni-corrected level of significance (72%). When focusing only on novel loci, replication rates were very similar. These data have been added to **Supplementary Table 13**.

Of the loci showing no replication significance ($P>0.05$; *PLN*, *FHOD3*, and *PPP1R3C*), two are near known Mendelian cardiomyopathy genes. These two loci notably show strong associations with relevant traits (including atrial fibrillation, heart rate, LV size, and ECG traits; see new PheWAS results later on). We posit that differences in genetic architecture (eg. tagging of causal variants) might play a role for these two loci, although this cannot be confirmed at this time.

For our MTAG-DCM loci, we could perform replication analyses for 64 of 65 loci. We found that 62 loci were concordant in direction of effect (97%), 56 loci were significant at a nominal level (88%), and 36 loci were significant at a Bonferroni-corrected level of significance (56%). When focusing only on loci that were not identified in GWAS-DCM, we found that 81% reached the nominal significance level and 52% reached the Bonferroni-corrected significance level. The replication rates were only marginally less than the expectation based on the effect sizes in discovery and the attenuation we computed based on established loci. These replication data have been added to **Supplementary Table 14**. Of the discordant loci, one was near *CSRP3* (a known Mendelian cardiomyopathy gene) and another near *IGFBP3*.

Overall, these results confirm a high replication rate of our initial GWAS, and secondly provide reassurance of our MTAG approach. The summary of our replication is shown in **Figure R1**, which has also been added to the manuscript as **Extended Data Figure 4**.

Figure R1: Summary of replication.

Manuscript change:

Page 4:

“To assess the robustness of our loci, we performed a replication analysis using independent samples from HERMES, MVP and All of Us, totalling up to 13,258 cases of NICM/DCM and 1,435,287 controls (Extended Data Figure 4; Supplementary Tables 13-14). Of 36 testable GWAS-DCM loci, all were concordant in effect direction and 92% replicated at $P < 0.05$. Of 64 testable MTAG-DCM loci, 88% replicated at $P < 0.05$ (81% for ‘MTAG-only’ loci; Supplementary Note). Furthermore, none of the significant loci showed meaningful heterogeneity in discovery (Supplementary Tables 15-16). These results confirm the robustness of our GWAS and MTAG approaches.”

Supplementary Materials change:

Page 30:

“We aimed to assemble a large replication cohort to validate the findings from our discovery analyses. To this end, we combined data from a parallel GWAS effort for DCM from the Heart Failure Molecular Epidemiology for Therapeutic Targets (HERMES) consortium³¹, data from the Million Veteran Program (MVP), and data from the All of Us Research Program. In our approach, we were careful to include only samples that were not already included in our discovery datasets (as outlined in more detail for each dataset below), which yielded a replication meta-analysis of up to 13,258 cases and 1,435,678 controls.

HERMES

In a parallel effort, the HERMES consortium recently released a manuscript describing a European-ancestry GWAS meta-analysis for DCM. This effort included both ‘hard DCM’ cases and ‘broad’ DCM cases (defined as LV systolic dysfunction in absence of a number of secondary causes), totalling 14,255 cases and 1,199,156 controls. We refer to the associated preprint for details on genotyping, phenotyping and GWAS analyses³¹. We note that a substantial number of ‘hard DCM’ datasets from HERMES also contributed to the present GWAS-DCM. Therefore, to remove the possibility of overlapping samples, the HERMES meta-analyses were rerun restricting to non-overlapping datasets. These included BioVU, CHB, deCODE, DiscovEHR-GSA, DiscovEHR-Omni, EstBioBank, GoDARTS-ILLUMINA, PIVUS, ULSAM, DCM-UCL, and GEL. The datasets were combined using an inverse-variance-weighted fixed-effects meta-analysis, totalling up to 8,480 cases and 756,404 controls. The lead variants from GWAS-DCM and MTAG-DCM were extracted from this meta-analysis.

MVP

Cohort description

The Veterans Affairs Million Veteran Program (MVP) started recruiting US military veterans from 63 Veterans Affairs (VA) facilities across the United States in 2011 (ref.³²). Veterans aged 18 years and older are recruited into MVP where participants are linked to VA electronic health records (EHR), complete a questionnaire, and submit a blood sample at enrollment. The EHR includes information on inpatient International Classification of Disease (ICD) diagnosis codes, Current Procedural Terminology (CPT) procedure codes, and clinical laboratory measurements. Genotyping and quality control in MVP has been reported previously^{33,34} and are summarized in detail below

Genotyping and quality control

Specimen collection and genotype quality control have been described in detail before^{33,34}. In brief, blood specimens were collected at recruitment sites across the country then shipped within 24 hours to

the VA Central Biorepository in Boston, MA for processing and storage. Study participants were genotyped using a customized Affymetrix Axiom biobank array (the MVP 1.0 Genotyping Array), containing 723,305 variants. Duplicate samples were excluded from the genetic analysis. Additional exclusion criteria included: samples with observed heterozygosity greater than the expected heterozygosity, missing genotype call rate greater than 2.5%, and incongruence between sex inferred from genetic information and gender extracted from phenotype data. Probes with high missingness (>20%), those that were monomorphic, or those with a Hardy Weinberg Equilibrium $p < 1 \times 10^{-6}$ in both the overall cohort and within one of the 3 major harmonized race/ethnicity and genetic ancestry (HARE)³⁵ race or ethnicity groups (non-Hispanic White, non-Hispanic Black, or Hispanic/Latino). See below for HARE methods.

KING13 was used to measure relatedness between individuals in the sample. ADMIXTURE³⁶ was used to calculate loadings on five 1000Genomes reference populations³ representing the majority of ancestry within the United States - GBR (British), PEL (Peruvian), YRI (Yoruba/Nigerian), CHB (Han Chinese), and LWK (Luhya/Kenyan). Pre-analysis QC was performed to remove SNPs that were rare (MAF < 1%), had high missingness (> 5%), or had excess heterozygosity ($F_{st} < -0.1$). SNPs that passed filters were then merged with the 1000 Genomes phase 3 reference panel³, removing SNPs that were not shared in both filesets. LD pruning was performed using the 'indep-pairwise' command in PLINK version 1.9, with window = 1000, shift = 50, and $r^2 = 0.05$, and excluding loci with complicated LD structure (i.e. MHC and KIR). Principal components (PCs) were computed using plink2 (ref.11).

The HARE approach, developed by MVP, was used to assign individuals to populations or groups³⁵. This machine learning algorithm leverages information from both the self-identified race/ethnicity data from the survey and data from the genome-wide array to create respective variables for downstream analyses. HARE categorized Veterans into four mutually exclusive groups: (1) non-Hispanic White, (2) non-Hispanic Black, (3) Hispanic or Latino, or (4) Asian. High concordance was observed between HARE-defined non-Hispanic White and non-Hispanic Black populations, and genetically inferred European and African ancestry populations, respectively.

Imputation to TOPMed Imputation Panel

Genetic imputation was performed to the TOPMed reference panel²⁴. Pre-phasing was performed using SHAPEIT4 (v 4.1.3; ref.37) using 20MB chunks and 5MB overlap, and Minimac4 (ref.38) software was used for imputation using 20MB chunks with 3MB overlap between chunks.

Genetically Inferred Ancestry (GIA) definition

To estimate ancestry, we obtained a reference dataset from the 1000 Genomes Project and used the smartpca module in the EIGENSOFT package (<https://github.com/DReichLab/EIG>) to project the PC loadings from a group of unrelated individuals in the reference dataset. We merged this dataset with the MVP dataset and ran smartpca to project the PCA loadings from the reference dataset. We trained a random forest classifier using continental ancestry meta-data based on the top 10 principal components from the reference training data to define genetically inferred ancestry. We then applied this random forest to the predicted MVP PCA data and assigned ancestries to individuals with a probability greater than 50%. Those with a probability less than 50% for any particular ancestry group were excluded from the study. The final GIA population classifications were (1) African (AFR), (2) Admixed American (AMR), (3) East Asian (EAS), (4) European (EUR), or (5) South Asian (SAS).

Cardiomyopathy Phenotyping

NI-DCM cases and controls were defined using International Classification of Diseases, 9th or 10th Revision (ICD-9; ICD-10) billing codes. In MVP, the version 21.1 clinical data freeze was used, which

contains EHR data up to September 30, 2021. Cases were defined by the presence of 'dilated cardiomyopathy' code (I42.0) excluding individuals with prior ischemic cardiomyopathy (I25.5) or coronary artery disease (CAD; I21-I24, I25.2, 410-412), or presence of a CAD code with 30 days after their first DCM code. Controls were defined by a lack of DCM code then individuals were excluded if they ever had codes for heart failure, hypertrophic cardiomyopathy (I42.1, I42.2, 425.1), alcoholic cardiomyopathy (I42.6, 425.5), peripartum cardiomyopathy (O90.3, 674.5), secondary cardiomyopathy (425.9), or drug induced cardiomyopathy (I42.7). Date of first event was defined as the date of the occurrence of the first code. This left a total of 3,964 cases (1,239 AFR, 223 AMR, 2,502 EUR) and 522,610 controls (99,878 AFR, 53,475 AMR, 369,257 EUR) for GWAS analysis.

GWAS

A case-control genome-wide association analysis (GWAS) for DCM was performed within each GIA group using REGENIE, then combined in an inverse variance weighted meta-analysis using GWAMA. Only AFR, AMR, and EUR had enough cases for analysis. A mixed model approach was implemented with adjustment for age at study enrollment, biological sex, and the first 10 genetic PCs. The lead variants from GWAS-DCM and MTAG-DCM were extracted from this meta-analysis.

All of Us

Details on sequencing and DCM phenotyping in the All of Us Research Program are described earlier in this document. For purposes of replication, we ran a GWAS analysis for our NI-DCM phenotype. Since the MGB health system contributed some samples to All of Us, we took a restrictive approach to minimize the potential for sample overlap between discovery and replication. In particular, we removed any sample in All of Us with a ZIP code from Massachusetts. This procedure left 815 NI-DCM cases, and 156,209 controls. We then used REGENIE v3.2.2 to perform a GWAS for the NI-DCM phenotype, using an approximate Firth's regression model. The lead variants from GWAS-DCM and MTAG-DCM were extracted from this multi-ancestry analysis.

Meta-analysis and quality-control

To combine data from the several replication cohorts, we performed an inverse-variance-weighted fixed-effects meta-analysis. This meta-analysis included up to 13,258 cases and 1,435,678 controls. We then filtered these results based on several criteria. First, we retained variants with $MAF > 1\%$ in the replication meta-analysis and with at least 1000 cases contributing to the replication meta-analysis. Second, per locus, we restricted to the single strongest lead variant in discovery. This procedure left qualifying replication results for 36/38 GWAS-DCM loci and for 64/65 MTAG-DCM loci. *P*-values were computed as one-sided *P*-values taking into account the direction of effect in discovery.

We first assessed the calibration of effect sizes between replication and discovery. When restricting to previously-established DCM loci, we found that effect sizes in replication were attenuated to ~0.5 of the GWAS-DCM discovery effect sizes. For MTAG-DCM, previously-established loci were attenuated to ~0.56 of discovery. Similar calibration was seen also when assessing all loci (Extended Data Figure 4). The attenuation of effect sizes is likely a reflection of i) the broader case definition used in most of the HERMES cohorts - for which we established a substantially lower heritability estimate - and ii) the older age of DCM cases included in MVP. Other contributory factors may be the inclusion of several non-European ancestry samples from MVP and AoU, and Winner's curse inflating effect sizes in discovery. These last points do not seem substantial, however, as restriction to European ancestry samples did not meaningfully alter effect sizes, and effect size calibration was highly similar between known and novel loci on average.

Power calculations

We then computed the expected power in replication. To this end, we computed the effective sample size for each variant in each contributing dataset, computed using the formula $4/(1/cases + 1/controls)$, and then computed the meta-analysis effective sample size as the sum of these values. We then used the function `genpwr.calc()` in R package `genpwr` to compute power for each variant. We used the effective sample size in replication, the minor allele frequency in replication, and the 'attenuated' effect sizes based on discovery as input; we computed power assuming a logistic additive model. The attenuated effect sizes were computed based on the effect size attenuation based on previously-established DCM loci only. Power was computed at the 'nominal' level (one-sided $\alpha=0.05$) and at the Bonferroni-corrected level (one-sided $\alpha=0.05/\text{number of testable loci}$). To then calculate the total number of expected replicating loci, we took the sum of the power values across loci. Assuming all discovery loci are true and assuming homogeneous effect size attenuation across loci, we estimated that we had power to replicate $\sim 35.6 / 36$ GWAS-DCM loci at the nominal level, and $\sim 31.8 / 36$ loci at the Bonferroni-corrected level. When considering only novel loci, we had power to detect $\sim 24.6 / 25$ GWAS-DCM loci at the nominal level and at $\sim 21.7 / 25$ loci at the Bonferroni-corrected level. For MTAG-DCM, we calculated that we had power to replicate $\sim 60.4 / 64$ loci at the nominal level and $\sim 43.2 / 64$ loci at the Bonferroni-corrected level. When considering only MTAG loci that were not identified in GWAS-DCM, we calculated that we had power to replicate $\sim 28 / 31$ loci at the nominal level and $\sim 19.1 / 31$ loci at the Bonferroni-corrected level.

Replication rates and results

For GWAS-DCM, we found that 36/36 (100%) of loci were concordant in direction of effect, 33/36 loci reached the nominal significance level (92%), and 26/36 loci (72%) were replicated at the Bonferroni-corrected level (Extended Data Figure 4). When considering only novel loci, 23/25 reached the nominal level (92%) and 18/25 reached the Bonferroni-corrected level (72%). Of non-replicating loci ($P>0.05$) two were near Mendelian cardiomyopathy genes (PLN and FHOD3). We posit that differences in genetic architecture (eg, tagging of causal variants) might underlie the difference, although this can not be proven at this time. The third non-replicating locus was near PPP1R3C.

For MTAG-DCM, we found that 62/64 loci (97%) were concordant in direction of effect, 56/64 (88%) reached the nominal level, and 36/64 (56%) reached the Bonferroni-corrected level (Extended Data Figure 4). When considering only loci not already identified in GWAS-DCM, we found that 25/31 (81%) reached the nominal level, and 16/31 (52%) reached the Bonferroni-corrected significance level. Of note, the observed replication rates for MTAG-DCM were only slightly lower than what could be expected based on our power calculations. Of discordant loci, one was near CSRP3 (a Mendelian cardiomyopathy gene) and one near IGFBP3.

Overall, the replication analyses demonstrate a substantial replicability of our initial GWAS-DCM findings. Secondly, the replication analyses provide reassurance of our MTAG approach to identify genetic signals for DCM. ”

In addition, please assess whether there is heterogeneity of the effect sizes between the different study cohorts or if they show large agreement.

As suggested, we have added **Supplementary Table 15** and **Supplementary Table 16**, which highlight heterogeneity test results for all lead variants in GWAS-DCM and MTAG-DCM, respectively. For GWAS-DCM, only 2 lead variants showed a nominal level of significance ($P < 0.05$; 2.3 expected by chance), both of which were secondary signals in their respective loci. For MTAG-DCM, 4 lead variants showed a nominal level of significance ($P < 0.05$; 4.3 expected by chance), of which two represented secondary signals in their respective loci. Of the remaining 2 loci, one was near *HSPA4* ($P = 0.045$) and one near *MAP3K7* ($P = 0.042$); in both of these loci, the association with DCM was convincingly confirmed in our replication set. Taken together, we found no evidence of substantial heterogeneity across datasets in our GWAS meta-analysis.

Manuscript change:

Page 4:

“To assess the robustness of our loci, we performed a replication analysis using independent samples from HERMES, MVP and All of Us, totalling up to 13,258 cases of NICM/DCM and 1,435,287 controls (Extended Data Figure 4; Supplementary Tables 13-14). Of 36 testable GWAS-DCM loci, all were concordant in effect direction and 92% replicated at $P < 0.05$. Of 64 testable MTAG-DCM loci, 88% replicated at $P < 0.05$ (81% for ‘MTAG-only’ loci; Supplementary Note). Furthermore, none of the significant loci showed meaningful heterogeneity in discovery (Supplementary Tables 15-16). These results confirm the robustness of our GWAS and MTAG approaches.”

A second important question is how the results in the current study more generally compare to those of ref 19, given that very similar approaches were applied.

We thank the Reviewer for this point. We have added a **Supplementary Note** discussing the similarities and differences between our study and the HERMES study (Zheng et al. 2023).

How do gene prioritizations agree or differ?

First, we discuss the overlap of loci and genes prioritized within the overlapping loci. Of GWAS-DCM loci, 27 overlapped loci from any of the analyses in Zheng et al., while 46 of 65 MTAG-DCM loci overlapped any of the loci from Zheng et al. While both studies nominated the same gene in only about 70% of overlapping loci, we found that this was mainly due to discordance in loci with no clear ‘winner’ in one or both of the studies. Among loci with strongly prioritized genes in both studies, we strikingly find ~94% concordance (only one discordant locus across GWAS and MTAG, near *CRIM1*).

Do both studies highlight the same or partly different cell types as major contributors?

Second, we discuss the divergent cell type enrichment results between our study and the results of Zheng et al. While both studies find strong enrichments within cardiomyocytes, Zheng et al. additionally report significant enrichments in other cell types, including mural/vascular cells and fibroblasts. As discussed in the extensive **Supplementary Note**, the additional enrichments in Zheng et al. came from 'disease-dependent' gene programs - ie, using genes with differential expression between DCM and non-failing hearts. We therefore performed additional analyses for cell type-specific and disease-dependent gene programs using two independent single cell datasets (see **Figures R2-3** and the new **Supplementary Figure 13**).

In both datasets, we found no meaningful enrichments for other cell types, using our predefined enrichment statistic. Of note, Zheng et al. used a more liberal enrichment statistic for hypothesis testing, and this likely contributed partly to the divergent cell type results. In addition, other technical differences likely contributed to some extent - including a slightly different underlying phenotype in GWAS and a different approach to gene mapping. These technical differences are explained in greater detail in the **Supplementary Note**. Taken together, we recommend that - outside of cardiomyocytes - enrichments in other cell types should be treated as interesting, but preliminary, at this stage.

How do the PRS compare to each other?

Third, we compared the predictive capacity of PRS constructed from both studies. Within three different datasets, our PRSs achieved better prediction accuracy than the respective PRSs from Zheng et al., when assessed by several performance metrics (effect sizes, AUC, variance explained). For instance, in UK Biobank, our GWAS-DCM and MTAG-DCM scores achieved effect sizes of OR~1.64 and OR~1.91 per standard deviation, while the Zheng et al. scores achieved effect sizes of OR~1.61 and OR~1.83 per standard deviation, respectively. Our MTAG-DCM score explained ~9.5% of variance on the liability scale, while the Zheng et al. MTAG score explained ~7.9% of variance on the liability scale. The detailed comparison has been added to new **Supplementary Table 41**. The better prediction accuracy is consistent with the larger number of significant loci identified in our study, but also potentially consistent with the larger number of variants in our PRS. Nevertheless, both PRS show strong associations with DCM. We have added these comparisons to the **Supplementary Note**.

Manuscript change:

Page 5:

“Of note, Zheng et al. described enrichments for DCM heritability in other cardiac cell types¹⁹; this discrepancy is most likely due to technical differences, including use of a different enrichment statistic²⁴ (Supplementary Note). Taken together, our results highlight the central role of cardiomyocyte dysfunction in DCM pathogenesis.”

Supplementary Materials change:

Pages 56:

“Comparison with results from Zheng et al.

Genes prioritized in overlapping loci

Similar to our study, Zheng et al. performed a GWAS and MTAG for DCM, followed by gene prioritization through integration of several lines of evidence³¹. Of our 38 significant loci in GWAS-DCM, 20 overlapped genome-wide significant loci from GWAS for NICM/DCM reported by Zheng et al., while a total of 27 overlapped loci reported by the authors at more inclusive discovery thresholds (ie, DCM-Broad analyses at FDR1%, DCM-Strict analyses at genome-wide significance, or MTAG analyses at genome-wide significance³¹). Details on locus overlap is described in Supplementary Table 39. Across the 27 overlapping loci, gene prioritization from both studies nominated the exact same gene as the most likely causal gene in ~67% of the time, while ~19% of loci were partially concordant (ie, Zheng et al. described multiple genes with equal prio scores, one of which was concordant with our prioritized gene), and ~15% of loci were discordant. Of note, this intersection analysis considers all loci, even those with no gene highly-prioritized by our definitions. Therefore, we then restricted the comparison to loci with highly prioritized genes in both studies (ie, ≥ 2.5 points and prioritized in our study AND ≥ 3 points in Zheng et al. without ties). Strikingly, among 16 overlapping loci with ‘strong prioritization’ in both studies, the nominated gene was concordant in 94% of the time; only one locus was discordant - with CRIM1 prioritized in our study and STRN prioritized in Zheng et al. (Supplementary Table 39). When focusing on the 65 loci from our MTAG-DCM, 46 overlapped any of the significant loci from Zheng et al., with similar convergence of prioritization. Of all overlapping loci, ~72% nominated the same causal gene, ~13% showed partial concordance, and in ~15% of loci the most strongly prioritized gene differed between the two studies (Supplementary Table 40). More importantly, when restricting to loci where both studies strongly prioritized a gene, ~96% were concordant (again, only the CRIM1 locus was discordant).

Interestingly, we note that both CRIM1 and STRN are differentially expressed across several cell types in the single cell comparison of DCM LVs versus non-failing LVs. Furthermore, in our analyses (in both GWAS-DCM and MTAG-DCM) we identified two lead variants in this locus, of which one closer to CRIM1 and one closer to STRN. These findings entertain the possibility that both genes have a causal role in DCM biology, although this would require functional validation.

Overall, the locus comparison results highlight a strong consistency in gene prioritization between our study and Zheng et al., in particular for genes identified with high prioritization scores.

Cell type enrichments

In the current study, we identified significant enrichment for DCM heritability only in cardiomyocyte gene programs. Zheng et al. additionally reported significant enrichment for several other cell types (eg. fibroblasts, mural cells). To understand the source of these discrepancies, we compared the similarities and differences between the two studies in more detail. Similar to our study, Zheng et al. performed cell type enrichment analyses by integrating results from their DCM GWAS data with snRNAseq data of the heart³¹. The authors re-processes the Reichart dataset⁴⁵ to serve as their expression set, and utilized an analytical pipeline similar to our cell type enrichment pipeline. Using cell type-specific gene programs - similar to our findings - the authors report significant enrichment of DCM heritability only in cardiomyocytes³¹. In contrast, the authors additionally report significant enrichment for several other cell types (eg. fibroblasts, mural cells) when using 'disease-dependent' gene programs. In disease-dependent gene program analyses (details on methods described in a previous Supplementary Note above), we did not uncover robust enrichments for any cell type at Bonferroni significance. At nominal significance, only cardiomyocytes showed a (weak) consistent signal ($P=0.04$ in the Reichart dataset with positive coefficient in the Chaffin dataset).

Initially, we considered several potential explanations for this discrepancy. First, the cases included in the GWAS by Zheng et al. were only partially overlapping with our cases (including up to ~10k non-overlapping cases included in their effort). Perhaps more importantly, several of the included cohorts in Zheng et al. utilized a wider case definition - ie, any systolic dysfunction in absence of secondary causes. As such, the underlying GWAS data may have been inherently different between both studies. Nevertheless, we should note that the top loci from both studies show strong convergence, and the genetic correlations with cardiac endophenotypes were comparable between both efforts. For these reasons, we considered it less likely that the differences in the underlying GWAS data entirely explained the divergent cell type enrichment results, it may have contributed to an extent.

Second, we considered that differences in the construction of 'disease-dependent' gene programs from the snRNAseq data may have caused different results. Analytically, the approaches between both studies were highly similar. Zheng et al. used the Reichart dataset⁴⁵, which was also one of the two datasets used in our study. The authors used a similar pipeline to define 'disease-dependent' gene programs - including similar DCM/non-failing sample definitions, use of pseudo-bulking for DE-testing, and similar cutoffs for logFC and P-value in DE testing. One difference was that Zheng et al. re-processed the expression counts using CellBender to remove potentially remaining background noise, while we used the counts as provided by Reichart et al. We nevertheless note that CellBender was used to adjust count data in the Chaffin dataset³⁹ - where we also did not identify any significant enrichments for disease-dependent programs in our analyses.

Third, we considered that analytical differences in the statistical enrichment pipeline may have caused different results. Overall, the enrichment pipelines between both studies were reasonably comparable. Zheng et al. used parts of the sc-linker pipeline to perform their analyses⁴⁶; sc-linker uses s-LDSC for enrichment testing⁴⁴, which is also the tool used by us for enrichment testing of cell type programs. We note that sc-linker uses activity-by-contact (ABC) mapping to link genes to genomic regions⁴⁶, while we used a more simple approach based on close proximity to gene bodies⁴⁴. We note, however, that our approach yielded similar - or even stronger - enrichments for cell type-specific cardiomyocyte programs, which would indicate that this technical difference needn't be substantial. Nevertheless, as compared to cell type-specific gene programs, it is possible that ABC mapping is more important for disease-dependent programs (for which genes may be more distally regulated). Overall, the genomic mapping approach may have contributed to some extent to the different cell type enrichment results.

Critically, we found that Zheng et al. used a different statistic for hypothesis testing than used in our work. Specifically, the authors reported the 'enrichment' statistic or E_c . In contrast, we performed all hypothesis testing based on the 'enrichment coefficient' or Tau_C . When using the enrichment statistic instead of the coefficient, we recapitulate many of the significant findings reported in Zheng et al., including a pattern where a large proportion of disease-dependent gene programs reach nominal $P < 0.05$ (Supplementary Figure 13). Within the s-LDSC and sc-linker frameworks, there are 4 major output statistics that involve/describe enrichment of heritability^{44,46}. The simplest is the 'enrichment' or E_c statistic, which is the proportion of total heritability captured by the functional annotation of interest, divided by the proportion of SNPs included in the given annotation; this statistic is not conditional on other annotations/features fed into the s-LDSC model. The 'coefficient' or Tau_C is the regression coefficient from s-LDSC, which captures an 'adjusted' enrichment parameter conditional on the other annotations fed into s-LDSC. In our work, all coefficients are conditional on the baseline model (which incorporates annotations for functional regions, including coding regions, enhancer regions, UTRs, etc); additionally, for disease-dependent programs, we included an annotation for all genes that could be assessed in differential expression testing (to account for the correct background of genes in the tissue). A third s-LDSC statistic is the Tau_C^* , which is simply a re-scaled standardized Tau_C statistic to represent an effect size per standard deviation of the underlying annotation. The final enrichment statistic is the 'enrichment score' or E -score, which was newly proposed as part of the sc-linker framework⁴⁶. E -score essentially represents the difference between the E_c statistic for a given annotation and the background enrichment of all protein-coding genes with the relevant genomic-mapping in the given tissue. For cell type enrichment, the developers of s-LDSC previously recommended using Tau_C (or Tau_C^*) conditional on at least the baseline model for hypothesis testing^{40,44}, as this statistic corrects for the inherent enrichment of important genomic regions one might expect in GWAS. Since the publication of sc-linker, the developers recommend using E -score as an alternative⁴⁶, since it is corrected for the background of protein-coding regions while potentially yielding more power than Tau_C . Overall, one might conclude that E_c is the most 'liberal' statistic for enrichment testing (although prone to inflated type 1 error in cell type analyses; ref.40), while Tau_C is the most 'conservative' statistic when conditioned on the baseline model and an appropriate background of genes⁴⁰. In their study, Zheng et al. used the 'enrichment' statistic or E_c , and we could indeed recapitulate several of their findings by performing hypothesis testing on E_c (Supplementary Figure 13). Importantly, these enrichments could not be recapitulated by us when conditioning on the baseline model and the appropriate background of genes (ie, when using Tau_C).

Taken together, the differing results from cell type enrichment analyses - of disease-dependent gene programs - may be partly explained by the reporting of a different enrichment statistic. Nevertheless, other technical differences likely contributed to some extent too - including a slightly different phenotype in GWAS and the use of a more simple genomic mapping approach in our work. In all, across the GWAS studies, consistent evidence was found only for cardiomyocytes. For these reasons, we recommend that - outside of cardiomyocytes - enrichments in other cell types should be treated as interesting, but preliminary, at this stage.

Polygenic score prediction

Similar to our study, Zheng et al. report strong prediction of DCM using a PRS constructed from their GWAS. The authors tested their PRS in the UK Biobank, and reported prediction effect sizes of 1.76 OR increment per SD of PRS (95% CI 1.64 to 1.90). To more directly compare results, we then also tested our PRS within the UK Biobank, using the same dataset described in our GWAS and further restricting to

samples with i) high-quality exome sequencing and genotyping array data available, ii) European genetic ancestry⁴⁸, iii) who were not related at a third degree or closer, and iv) who were not included within the first 45k participants with cardiac MRI data (since these samples contributed to the MTAG analyses). This procedure left 793 NI-DCM cases and 325313 controls. We then reran our main GWAS-DCM excluding UK Biobank, and constructed a new PRS using PRSCs as described in our main methods⁴⁹. Using this GWAS-DCM score (which was standardized to mean 0 and unit variance, and out of which the first 12 PCs were regressed), we then assessed the association with NI-DCM, adjusting for sex, age, age², PC1-12 and the genotyping array. Similarly, we re-ran the MTAG analysis using the GWAS-DCM[exclUKB] as base GWAS, and created MTAG-DCM[exclUKB] scores. The GWAS-DCM[exclUKB] score was strongly associated with NI-DCM in this dataset, with an OR increment per SD of PRS of 1.64 (95%CI 1.53 to 1.76), as was our MTAG-DCM[exclUKB] score at an OR increment per SD of PRS of 1.91 (95%CI 1.78 to 2.05). To compare more directly with the Zheng et al. PRS, we then downloaded their scoring files from the PGS catalog (GWAS:<https://www.pgscatalog.org/score/PGS004861/> and MTAG: <https://www.pgscatalog.org/score/PGS004861/>), and scored the same samples using both scoring files. We found that the GWAS (OR per SD 1.61; 95%CI 1.53 to 1.76) and MTAG scores (OR per SD 1.83; 95%CI 1.70 to 1.98) from the authors did well in prediction of NI-DCM, although slightly less well than the scores from our GWAS and MTAG, respectively. Using other metrics for prediction accuracy - including the variance explained, the AUC, and the AUPRC - similar patterns were observed (Supplementary Table 41). We note that we observed a slightly larger effect size for the Zheng et al. scores than reported by the authors; we posit that this difference is a reflection of the more stringent phenotype definition (ie, NI-DCM as compared to 'any' DCM).

Therefore, within the UK Biobank, the Zheng et al. PRSs seem to perform somewhat less well than the PRSs constructed from our data - although the confidence intervals were still overlapping. We therefore additionally assessed the Zheng et al. PRSs in the European subset of the All of Us dataset and within the Amsterdam dataset. In these datasets, we found that our GWAS-DCM and MTAG-DCM scores consistently achieved higher effect sizes, AUCs, and variances explained than the GWAS and MTAG scores from Zheng et al., respectively (Supplementary Table 41). The only exception was for AUPRC values in the All of Us dataset, which were marginally higher for the Zheng et al. scores.

Overall, the above results show that both studies produce scores that strongly predict NI-DCM and can transfer to datasets from different countries. The slightly better prediction of our PRS - despite considerably smaller case numbers - might reflect the higher specificity of our underlying phenotype. This would be consistent with the larger number of significant loci identified in our study. Alternatively, we note that Zheng et al. used ~700k variants for their PRS (as per PGS catalog), while our PRS was built using ~1.1M variants. The higher genome coverage might have contributed somewhat to a better prediction power using our PRS. Taken together, both studies produce strongly predictive PRS for DCM, with our PRS showing slightly better prediction of DCM in European ancestry."

Minor comments:

The eQTL and pQTL analysis is very interesting. However the pQTL data that was used is from blood. The authors show that cardiac tissue and specifically cardiomyocytes are key players. Please use heart specific pqtL data (e.g. Assum et al. Nat Comms 2022) for this analysis.

We thank the Reviewer for pointing us towards the study by Assum et al., who computed cardiac-specific pQTLs. As suggested, we have applied our MR and colocalization analyses to this pQTL dataset. We note that the pQTL data were based on only ~75 samples, and therefore power to identify instrumental variables was limited. Indeed, only one significant signal was identified in MR analyses across GWAS-DCM and MTAG-DCM, namely *BAG3*. While an interesting finding, it does not change gene prioritization meaningfully. For these reasons, we opt to leave these data out at this stage.

It would also be interesting to characterize the overlap of eQTL and pQTL results. What is the agreement between heart and blood?

These are interesting questions raised by the Reviewer. The proteomic assay used by the UKB-PPP only covers ~15% of potentially protein-coding genes, which complicates a direct comparison of transcript and protein MR results. Indeed, we identified only a handful of significant proteins using the blood proteome, most of which did not show evidence of colocalization and did not overlap significant transcripts from the transcriptome-wide MR. Therefore, the transcriptome-wide MR contributed much more information to gene prioritization than did the proteome-wide MR. We note, however, that several sub-threshold proteome-MR signals were consistent with findings from our transcriptome-MR analysis, indicating that larger DCM GWAS might yield biologically-informative results from proteome-MR, in the future.

The comparison of the genetic associations of the strict and more lenient DCM phenotype definition is interesting. What is the genetic correlation of the two traits? Is there more variance in the lenient definition?

The genetic correlation between our GWAS-DCM and the broader biobank-based NICM phenotype was around 0.93 (95%CI [0.87; 0.99]). Nevertheless, as described in our main text, the SNP-heritability estimate for NICM is substantially lower (~6-7%) than for DCM (12-16%). While we acknowledge that there is case overlap between these sets, these results show that DCM and NICM have a strongly shared genetic basis, although the total contribution of genetics is higher in DCM.

To assess the heterogeneity of the genetic architectures for both traits, we used stratified LD fourth moments regression (sLD4M). sLD4M quantifies the polygenicity of traits from GWAS data, where polygenicity is defined as the *effective number of causal variants* (Me) (O'Connor et al. 2019). Traits with larger Me values have more evenly

distributed genetic effects across many variants in the genome; traits with smaller Me values have less evenly distributed effects across fewer variants in the genome.

For GWAS-DCM, we found that Me was equal to 1264 (95%CI [846; 1681]) with a similar estimate for biobank-only NI-DCM (1223, 95%CI [730; 1717]). Consistent with a more heterogeneous architecture, the Me estimate for NICM was numerically (but not significantly) higher at 1415 (95%CI [931; 1899]). Consistently, the mean *heritability per effectively causal variant* was numerically almost double in DCM (9.6e-05, 95%CI [7.2e-05; 0.000143]) as compared to NICM (4.9e-05, 95%CI [3.6e-05; 7.4e-05]), although again not significantly so.

These new analyses give an indication that the genetic architecture of DCM is less heterogeneous than NICM, despite a larger total genetic contribution in DCM. However, given currently large confidence intervals, we do not want to over-emphasize these results. We therefore opt to leave these data out of the manuscript at this stage.

How do the single cell results compare to those of Reichart et al Science 2022?

This is an interesting question raised by the Reviewer. Our single cell analyses consisted of two parts: One part focused on cell type enrichment, and the other on expression patterns and differential expression for candidate genes.

As suggested, we replicated our cell type enrichment analyses using the Reichart et al. dataset (Reichart et al. 2022). We processed the snRNAseq data using custom pipelines, and then performed the cell type enrichment analysis in a similar manner as described for the Chaffin et al. dataset. In both datasets, using cell type-specific gene programs, we only find robust evidence for enrichment of cardiomyocyte gene programs (**Figure R2** and **Extended Data Figure 5**). We have added these results to **Supplementary Table 17**. We also performed enrichment analyses for 'disease-dependent' gene programs, as discussed in our response to the previous comment. As mentioned, we find no robust evidence for enrichment of other cell types using disease-dependent gene programs (added to the new **Supplementary Table 17**).

Figure R2: Cell type enrichment results using cell type-specific gene programs from two snRNAseq datasets.

Figure R3: Cell type enrichment results using disease-dependent gene programs from two snRNAseq datasets.

To improve our analyses focused on cell type specific expression patterns and differential expression patterns, we now combine data from three different single cell datasets (Chaffin et al. 2022; Reichart et al. 2022; Koenig et al. 2022). In particular, to declare a gene significantly differentially expressed between DCM and non-failing hearts, we now require a consensus of at least 2 of 3 datasets. The data are showcased in **Figure R4** and new **Figure 3**.

Figure R4: Cell type-specific expression and differential expression of the top prioritized genes for DCM from three single-cell datasets of the left ventricle.

Manuscript change:

Page 5:

“Cell type enrichment analyses - using two published LV single nucleus RNA sequencing (snRNAseq) datasets^{22,23} - highlighted cell types of relevance to DCM. Only cardiomyocyte-specific genes were significantly and robustly enriched for DCM heritability across datasets ($P < 3e-7$ for enrichment coefficient; Supplementary Table 17; Extended Data Figure 5; Supplementary Figure 5).”

Page 6:

“To scrutinize the prioritized genes further, we queried published single cell data of the human LV from three datasets^{22,23,43} - including data from 61 non-failing donors and 81 DCM patients. We found that many of the prioritized genes showed high and/or preferential expression in cardiomyocytes (Figure 3; Supplementary Table 24). These genes underscore the role of the contractile apparatus in DCM pathogenesis⁴⁴, through known cardiac sarcomeric genes (eg. *TTN*, *OBSCN* and *ACTN2*), but also lesser-described structural genes including *SVIL* (encoding an actin-binding protein recently implicated in HCM²¹) and *PDLIM5* (a cytoskeletal linker⁴⁵). Other genes with cardiomyocyte-specific expression included *MITF* (a transcription factor implicated in cardiac hypertrophy in vitro⁴⁶) and *MLIP* (a lamin-

interacting protein associated with myocardial adaptation in mice⁴⁷). Several genes showed significant differential expression between DCM and non-failing hearts (Figure 3; Supplementary Table 25). Notably, within cardiomyocytes, such genes included MAP3K7 (a mitogen-activated protein implicated in cardiospondylofacial syndrome⁴⁸), ADAMTS7 (a thrombospondin-regulating metalloprotease⁴⁹), and both PRKCA and CAMK2D (involved in calcium-handling^{50,51}). Of note, several genes highlighted from both GWAS and single cell data are being investigated as targets for other conditions (Supplementary Table 26). Taken together, these results show how integration of GWAS and single cell data - paired with appropriate cell type priors - may identify plausible gene candidates for cardiomyopathy and LV function.”

Page 18:

“To identify causal cell types for GWAS-DCM and MTAG-DCM, we used stratified-LDSC, as described in Finucane et al., 2018 (ref.24). To this end, we utilized two published single-nucleus RNA-sequencing (snRNAseq) datasets, one from Chaffin et al., 2022 (ref.22) and another from Reichart et al., 2022 (ref.23). The Chaffin et al. dataset included LV expression data on 11 DCM hearts, 16 non-failing hearts and 15 hypertrophic cardiomyopathy hearts. The cardiomyopathy samples came from explanted hearts with end-stage disease. Chaffin et al. identified 17 major cell types, which were used to define cell type-specific gene programs for enrichment testing (see Supplementary Note for detailed methods). The Reichart et al. dataset included data on 61 end-stage cardiomyopathy hearts (52 with DCM) and 18 non-failing controls. Reichart et al. identified 9 major cell types in the LV, which were used to define cell type-specific gene programs for enrichment testing (see Supplementary Note for detailed methods). Finally, in addition to the ‘cell type-specific’ expression annotations described above, we also explored ‘disease-dependent’ cell type annotations. Disease-dependent programs were based on genes with significant differential expression between DCM samples and non-failing samples, irrespective of their cell type-specificity. The detailed methods for this analysis are described in the Supplementary Note. Of note, cell type enrichment analyses were not informed in any way by our GWAS/MTAG gene prioritization scheme.”

Page 19:

“We then aimed to identify cell type expression patterns and cellular functions for the prioritized genes from our GWAS and MTAG. To this end, we used available snRNAseq or scRNAseq data from three published datasets, including Chaffin et al.²², Reichart et al.²³, and Koenig et al.⁴³. Koenig et al. performed snRNAseq/scRNAseq on 18 LVs from DCM patients and 27 LVs from control donors.

Using the processed AnnData/Seurat objects from each study, we first restricted to control/non-failing samples from the LV, and then log-normalized the expression data with scale 10000 (if not already normalized). To harmonize cell type data across datasets, we then used the available cell type and/or cell state annotations to collapse or split cell types into ‘harmonized’ cell types (Supplementary Note). For genes with at least 0.5 points from our prioritization scheme in GWAS-DCM or MTAG-DCM, we then exported several expression measures from each dataset. These included i) the mean normalized expression within harmonized cell types and pseudo-bulk data and ii) the percentage of nuclei/cells with non-zero expression for each harmonized cell type and in pseudo-bulk. We then combined data by taking the weighted average of expression values (weighted by the number of nuclei/cells contributing in each dataset). For plotting purposes, we then focused on the list of 63 prioritized genes and computed the scaled relative normalized expression of a given gene in a given cell type, as compared to all other cell types.

We further aimed to identify genes differentially expressed between DCM and non-failing hearts. To this end, we utilized results from cell type-specific differential expression (DE) analysis for DCM versus non-failing hearts, as described in Chaffin et al.²² and Koenig et al.⁴³ For the published Chaffin et al. DE analysis, we consider results suggestive if reaching transcriptome-wide multiple-testing-adjusted two-

sided $P < 0.05$ using CellBender-adjusted counts, without failing the ‘background contamination’ flag. For the published Koenig et al. DE analysis, we considered results suggestive if reaching transcriptome-wide multiple-testing-adjusted two-sided $P < 0.05$. Finally, we used the Reichart et al. dataset²³, to perform a new DE analysis, comparing the 52 DCM LVs to 18 control LVs, using the same cell types that could be included for DE testing in their original publication (Supplementary Note). Again, a transcriptome-wide multiple-testing-adjusted two-sided $P < 0.05$ was considered suggestive. While we acknowledge that the cell types included in DE testing were not perfectly aligned across datasets, we approximately matched cell types to identify signals that were consistent across datasets (Supplementary Table 25). Finally, we declared significance for a gene, if at least two of three datasets showed a suggestive result with concordant direction of effect within comparable cell types.“

Supplementary Materials change:

Page 38:

“Cell type enrichment analysis using the Chaffin et al. snRNAseq dataset

Using the snRNA-seq data obtained from Chaffin et al., 2022 (ref.33), we performed several analyses focused on cell type enrichment. The dataset consisted of LV samples from 11 DCM patients, 16 non-failing controls and 15 HCM patients. In terms of analyses, we i) generated cell type-specific annotations for enrichment testing using stratified linkage disequilibrium score regression (s-LDSC)³⁴ and ii) generated ‘disease-dependent’ cell type annotations for enrichment testing using s-LDSC.

Cell type specific gene programs

Based on the Chaffin et al. dataset, we defined cell type-specific gene expression profiles by collapsing nuclei into 17 major cell types from the human left ventricle. We then identified differentially expressed genes in each cell type compared to all other cell types. To control for the inherent correlation of nuclei from the same individual, we created a pseudo-bulk expression profile after summing gene expression counts across all nuclei for each combination of individual and cell type. Individual and cell type combinations with fewer than 50 nuclei were omitted and lowly expressed genes were removed using the function `filterByExpr()` in `edgeR`³⁵. Gene expression was normalized with `DESeq2`³⁶ and differential expression testing was performed using `limma-voom`³⁷. Using a design matrix $\sim 0 + \text{cell_type} + \text{individual}$, we extracted an explicit contrast comparing expression in each cell type to all other cell types. For each cell type, we defined the cell type-specific profile as the top 10% most upregulated genes based on the t -statistic from the differential expression test.

s-LDSC analysis of cell type specific gene programs

We annotated SNPs within a 100 Kb window on either side of the transcribed region for each set of cell type specific genes, as in Finucane et al, 2018 (ref.38). Gene coordinates were based on the GRCh38 gene reference used in the snRNAseq data analysis. Using these annotations, we tested for cell type enrichment using s-LDSC, controlling for an annotation derived from all genes tested for differential expression and the baseline annotations from Finucane et al., 2015 (ref.34). As recommended, we report two-sided P -values from the tau ‘coefficient’ - which is conditional on all other annotations included in the model - and not the ‘enrichment’ statistic. As LD reference, we used the previously derived 1000 Genomes European ancestry LD reference provided with the software. To account for the 17 cell types tested for GWAS-DCM and MTAG-DCM, we applied a Bonferroni significance cutoff by setting significance at $0.05/17=0.0029$. P -values were one-sided.

s-LDSC analysis of disease-dependent gene programs

As described below for the Reichart dataset, we also performed an analysis of disease-dependent gene programs using the Chaffin et al. dataset (ref.33). We took the results from the differential expression analysis as described previously³³ (using CellBender-adjusted expression counts), and considered genes with $|\log FC| > 0.5$ and an FDR-adjusted $P < 0.05$ as 'disease-dependent' genes in the given cell type. We annotated SNPs within +/-100KB from each gene identified for each cell type and ran s-LDSC to identify GWAS heritability enrichment of these annotations, adjusting for baseline annotations from Finucane et al. 2015 (ref.34) and a set of annotations derived from all genes tested for differential expression in the given cell type. As recommended by Finucane et al., we report test statistics and corresponding P-values from the tau 'coefficient' - which is conditional on all other annotations in the model - and not the 'enrichment' statistic (which is not conditional on the other annotations).

Cell type enrichment and differential expression analyses in the Reichart et al. snRNAseq dataset

Using the snRNA-seq data obtained from Reichart et al., 2022 (ref.39), we performed several analyses focused on cell type enrichment and differential expression. The dataset consisted of samples from several anatomical locations (including several locations across the left and right ventricle) from 61 cardiomyopathy patients - of which 52 with DCM - and 18 non-failing controls. In terms of analyses, we i) generated cell type-specific annotations for enrichment testing using stratified linkage disequilibrium score regression (s-LDSC)³⁴, ii) generated differential expression data comparing left ventricles from DCM patients with non-failing control left ventricles, and iii) generated 'disease-dependent' cell type annotations for enrichment testing using s-LDSC.

Cell type specific gene programs

First, to test for enrichment of cell type specific gene programs in our GWAS/MTAG data, we generated a list of cell type specific genes. We removed nuclei labeled as 'native' or 'lowQC' prior to estimating cell type specific genes. We then performed 'pseudo-bulk' aggregation by summing gene counts across nuclei for each donor/tissue region combination, by cell type. We only retained a given donor/tissue region combination if they had at least 50 nuclei of that cell type. Lowly expressed genes identified with the filterByExpr() function in edgeR were removed. We normalized the pseudo-bulk expression with DESeq2 and fit the differential expression model $\sim 0 + \text{cell_type} + \text{donor_tissue}$ using limma-voom. Notably, we included a covariate for the donor/tissue region combination because each donor/tissue region will be represented across most cell types. We then extracted contrasts comparing gene expression in each focal cell type to all other cell types.

s-LDSC analysis of cell type specific gene programs

To generate annotations for s-LDSC, we sorted all genes tested for each cell type by t-statistic and selected the top 10% of genes to represent each cell type, as in Finucane et al, 2018 (ref.38). We annotated any SNP within +/-100KB of the genes for each cell type as 'cell type specific' SNPs. Using these annotations, we tested for cell type enrichment using s-LDSC, controlling for an annotation derived from all genes tested for differential expression and the baseline annotations from Finucane et al., 2015 (ref.34). As recommended by Finucane et al., we report test statistics and corresponding P-values from the tau 'coefficient' - which is conditional on all other annotations included in the model - and not the 'enrichment' statistic (which is not conditional on other annotations). To account for the 9 cell types tested for GWAS-DCM and MTAG-DCM, we applied a Bonferroni significance cutoff by setting significance at $0.05/9=0.0056$. P-values were one-sided.

Differential expression analysis of DCM versus controls

Second, we generated a list of differentially expressed genes between dilated cardiomyopathy (DCM) cases and normal controls by cell type. We first restricted our analysis to samples from the left ventricle (LV) and removed any nuclei flagged as low quality. Next, for a given cell type, we summed transcriptional counts across all nuclei from each donor of origin. Of note, we only generated a 'pseudo-bulk' profile for a donor if they had more than 20 nuclei of the given cell type. We then removed mitochondrial genes, ribosomal genes, and any gene that was found in <1% of nuclei from both DCM nuclei and control nuclei. We further removed lowly expressed genes using the function `filterByExpr()` from `edgeR`. We normalized the expression data using `DESeq2` normalization, and then tested for differential expression between DCM cases ($N_{\max}=52$) and non-failing controls ($N_{\max}=18$) using `limma-voom` with the model of $\sim 1 + \text{disease} + \text{sex}$. Multiple testing correction was performed using the Benjamini-Hochberg procedure.

s-LDSC analysis of disease-dependent gene programs

In contrast to the cell type specific gene programs defined by high cell type specificity of expression, we then also generated 'disease-dependent' gene programs for cell types. Disease-dependent gene programs consist of genes that are differentially expressed between the disease state and the healthy state, and therefore may consist partly of genes that are not expressed to a high degree in the given cell type or may not be cell type-specific. Such programs may capture disease-response mechanisms, rather than disease initiation mechanisms⁴⁰. To generate disease-dependent cell type annotations of s-LDSC, we used the results from the differential expression analysis described above, and considered genes with $|\log_{2}FC| > 0.5$ and an FDR-adjusted $P < 0.05$ as 'disease-dependent' genes in the given cell type. Of note, only 3 genes were identified in adipocytes with this procedure, and therefore we excluded adipocytes for the s-LDSC analysis. We annotated SNPs within $\pm 100\text{KB}$ from each gene identified for each cell type and ran s-LDSC to identify GWAS heritability enrichment of these annotations, adjusting for baseline annotations from Finucane et al. 2015 (ref.34) and a set of annotations derived from all genes tested for differential expression in the given cell type. As recommended by Finucane et al., we report test statistics and corresponding P-values from the tau 'coefficient' - which is conditional on all other annotations in the model - and not the 'enrichment' statistic (which is not conditional on the other annotations)."

Page 41:

"Harmonization of cell types across single cell datasets to construct LV expression patterns

We used three single cell datasets of heart to construct expression patterns for genes identified from our GWAS-DCM and MTAG-DCM. These datasets included Chaffin et al.³³, Reichart et al.³⁹, and Koenig et al.⁴¹. To harmonize cell type data across datasets, we used the available cell type and/or cell state annotations to collapse or split cell types into 'harmonized' cell types. In the Reichart dataset, nuclei with cell state 'PC1', 'PC2' or 'PC3' were collapsed into 'Pericytes'; nuclei with cell state 'SMC1.1', 'SMC1.2', or 'SMC2' were collapsed into 'VSMC'; nuclei with cell state 'EC7' were assigned 'Endocardial'; nuclei with cell state 'Meso' were assigned 'Epicardial'; nuclei with cell state 'EC8' were assigned 'Lymphatic endothelial'; nuclei with cell state 'EC1.0', 'EC2.0', 'EC5.0', or 'EC6.0' were assigned 'Cardiac endothelial'. In the Koenig dataset, cells/nuclei with cell type 'NK/T Cells' or 'B Cells' were collapsed into 'Lymphocyte'. In the Chaffin dataset, 'Cardiomyocyte_I', 'Cardiomyocyte_II', and 'Cardiomyocyte_III' were collapsed into 'Cardiomyocyte'; 'Endothelial_I', 'Endothelial_II', and 'Endothelial_III' were collapsed into 'Cardiac Endothelial'; 'Fibroblast_I', 'Fibroblast_II' and 'Activated_fibroblast' were collapsed into 'Fibroblast'; 'Pericyte_I' and 'Pericyte_II' were collapsed into 'Pericyte'; and 'Macrophage' and 'Proliferating_macrophage' were collapsed into 'Myeloid'.

An interaction analysis of rare variant status (possibly aggregated as genotype +, as currently done) and PRS would be highly interesting.

As suggested, we assessed the joint contribution of PRS and rare pathogenic variants to DCM risk. Because this analysis requires calling rare variants in both DCM cases and healthy controls, we turned to the AllofUs dataset for this analysis.

We acknowledge that rare variant status in AllofUs was defined using LOF variants (for genes where truncation is an established disease mechanism) and high-confidence ClinVar P/LP variants. For this reason, it is possible that some carriers are missed, although our variant curation is likely quite specific for P/LP. We used the same high-confidence genes as used previously; we designated this carrier definition as “ClinGen_PLP”.

We then performed logistic regression analyses among unrelated individuals, assessing the risk for DCM conferred by 3 tertiles of PRS in carriers and noncarriers of ClinGen_PLP. This analysis yielded a nice gradient of risk across tertiles in both carriers and noncarriers, suggesting that PRS and rare variants jointly contribute to DCM risk. The data are presented in **Figure R4** and the new **Extended Data Figure 8**.

Figure R4: Joint contribution of PRS and rare pathogenic variants to DCM risk in the All of Us dataset.

Manuscript change:

Page 9:

“and among carriers of pathogenic rare variants for DCM ($P=5.2 \times 10^{-7}$; Figure 5; Extended Data Figure 8).

”

To what extent could the finding of cardiomyocytes as main contributors be a consequence of the gene prioritization scheme? Specifically, the pops method is highly weighted and relies strongly on expression. As many important loci are cardiomyocyte specific this might lead to a self fulfilling prophecy.

This is an excellent point made by the Reviewer. If genes prioritized by external data are used for enrichment analyses, then these might be liable to some circular reasoning. We should note, however, that our cell type enrichment analysis was performed agnostic of our gene-prioritization approach. We used stratified LD score regression (sLDSC) for this analysis. As input, we used our GWAS summary statistics and cell type annotations - which were based on proximity to cell type-specific genes from the snRNAseq data. sLDSC then tested whether these regions were enriched for GWAS/MTAG heritability (without taking into account any gene prioritization schemes based on GWAS/MTAG).

At the same time, we acknowledge that our pathway enrichment analyses may have been affected by our gene prioritization scheme. In these analyses, prioritized genes were used as input, which might bias towards pathways that are strongly cardiac specific. While we stress that PoPS was agnostic to tissue a priori (it learnt tissue features on its own), we did decide to perform a secondary analysis for the pathway enrichment section. In this sensitivity analysis, we used only genes nominated by MAGMA. MAGMA is purely a distance-based gene prioritization tool that uses no external (tissue-specific) expression data. Reassuringly, for pathways that were potentially strongly cardiac-specific, the general pattern of enrichments persisted (eg, *sarcomere organization*, *myofibril assembly*, *actin cytoskeleton*, *ErbB signalling*, *cardiomyocyte signalling pathways converging on titin*, *left ventricular systolic dysfunction*, *dilated cardiomyopathy*, etc). These data show that our pathway enrichment results were not driven purely by our gene prioritization scheme. These data have been added to **Supplementary Table 23**.

Manuscript change:

Page 6:

“Accordingly, gene set enrichment analyses, using the 63 prioritized genes, identified several significant gene sets including “Cellular response to heat stress” (Supplementary Table 22-23, Supplementary Figure 7).”

Manuscript change:

Page 18:

“Since our prioritized genes may have been pre-selected towards genes with high cardiac expression (ie, through gene features learnt by PoPs), we performed a sensitivity analysis using genes nominated by MAGMA39 - a method based only on association signals near gene regions.”

Reviewer 2

Remarks to the Author:

The report by Ellinor, Daly, Aragam and Bezzina describes results of a large DCM GWAS meta-analysis. A large number of novel loci are described and the authors include a broad set of secondary analyses. The analyses are well-done and the paper is a pleasure to read. I do have some comments:

We are grateful to the Reviewer for their thoughtful consideration of our work.

1) Can the authors provide a pheWAS analysis of their PRS and lead variants? The specificity of the associations is not currently explored.

As suggested, we performed a pleiotropy analysis of our lead variants. To this end, we used the Cardiovascular Disease Knowledge Portal (CVDKP) to perform a look-up for several cardiovascular diseases and quantitative traits. The CVDKP assembles, harmonizes and meta-analyzes large-scale GWAS for important cardiovascular traits to produce well-powered and well-controlled large-scale GWAS data. The phenotypes included in this portal range from ECG traits, cardiac MRI traits, arrhythmia, heart failure, lipids, coronary disease, and more. Overall, most of the loci showed pleiotropic effects on ECG traits, blood pressure, heart rate, heart failure, atrial fibrillation, and cardiac functional and volumetric traits. The results from this look up are discussed in the main text, and have been added to **Supplementary Tables 34, 35 and 38**.

Since the above look-up is focused strongly on cardiovascular traits, we also performed a phenome-wide look-up using the UK Biobank PheWeb portal. This portal hosts GWAS data on thousands of disease codes, tested in the UK Biobank. We did not observe consistent patterns of association between our lead variants and extra-cardiac diseases, although - as expected - several lead variants were associated with atrial fibrillation, hypertension or heart failure. These results have been added to **Supplementary Tables 36 and 37**.

Manuscript change:

Page 4:

"GWAS-DCM signals showed strong pleiotropic effects on relevant cardiovascular traits, including cardiac MRI traits, electrocardiographic traits, blood pressure, heart failure, and arrhythmia (Supplementary Note)."

Supplementary Materials change:

Page 29:

"Assessment of pleiotropy for significant loci"

We aimed to identify pleiotropic effects for the lead variants identified in our GWAS-DCM and MTAG-DCM analyses (Supplementary Tables 34-38). First, we queried the Cardiovascular Disease Knowledge Portal (CVDKP; <https://cvd.hugeamp.org/>) to identify pleiotropic associations for relevant cardiovascular diseases and quantitative traits. At the suggestive significance level set by the portal, 33 of 38 GWAS-DCM loci showed potential pleiotropic associations with relevant traits (Supplementary Tables 34 and 38), which include cardiac MRI traits, ECG traits, HF, atrial fibrillation, and heart rate. In contrast, only two loci showed pleiotropic effects on coronary artery disease, of which one had a discordant effect between DCM and coronary disease (ADAMTS7). Similarly, of 65 MTAG-DCM loci, 40 loci showed potential pleiotropic associations with relevant traits (excluding MRI traits; Supplementary Tables 36 and 38); only three loci showed pleiotropic effects on coronary artery disease (again including the discordant ADAMTS7 locus).

The above look-up was based on the CVDKP, which is focused on cardiovascular traits. As such, this pleiotropy look-up was naturally biased towards potentially relevant traits, and would miss important pleiotropic associations outside of the cardiovascular system. We therefore performed a second look-up using a publicly-available phenome-wide disease analysis (PheWAS) from the UK Biobank (Supplementary Tables 35 and 37). Reassuringly, the vast majority of suggestive associations involved arrhythmia, conduction disease, hypertension, heart failure, and related cardiovascular diseases; there were only limited suggestive associations in other organ systems. These findings show that the phenotypic consequences of our DCM loci largely involve the cardiovascular system; furthermore, these results support the validity of DCM loci. “

2) How does the PRS compares to carrying a bona fide DCM Mendelian mutation in terms of DCM risk?

This is an interesting question raised by the Reviewer. In preliminary analyses performed within the UK Biobank, we found that individuals in the top centile of PRS have approximately 4-fold increased odds of DCM compared to all other individuals. In contrast, we found that carriers of pathogenic or likely pathogenic variants for DCM had an over 20-fold increased odds of DCM, as compared to noncarriers (in All of Us, the effect sizes were slightly lower at ~10-14-fold). These data suggest that PRS cannot yet identify individuals with the same risk as carriers of single monogenic mutations.

At the same time, as a quantitative variable, it is expected that PRS explains a larger proportion of disease variance in the general population. Indeed, we found that our PRS explains ~8-9% of population-wide variance on the liability-scale, while rare pathogenic and likely pathogenic variants explain ~2%. These results are similar to findings in hypertrophic cardiomyopathy (Biddinger et al. 2022). While interesting, these data are currently out of scope of our already dense study, and therefore we have opted to not pursue these analyses further.

Reviewer 3

Remarks to the Author:

The authors performed a large-scale genome-wide association study (GWAS) and multi-trait analysis (MTAG) for dilated cardiomyopathy (DCM). Using 9,365 DCM cases and 946,368 controls the study is about twice as large as previous GWAS. Expectedly, the number of loci showing genome-wide significance increased. Further analyses highlight the role of the contractile apparatus in the pathogenesis of DCM and mendelian randomization analyses showing that DCM liability is associated with an increased risk of systolic heart failure in context of other cardiovascular conditions.

The analyses were conducted with great care, the methodology is sound (as far as I can tell not being a bioinformatician) and the paper is written very well. The conclusions appear to be justified.

We appreciate the helpful guidance of the Reviewer.

Major comment

The combination of the DCM GWAS with the multi-trait analysis is not entirely transparent. Specifically, MRI-based measurements of end-systolic volume (LVESV) may reflect body size rather than pathological dilatation of the LV.

This is a good point, since LV volumetric traits may be affected by body size in a non-pathological way. We should note, however, that the LVESV GWAS data were indexed for body-surface-area, as is standard in clinical practice. While potentially not perfect, we posit that this approach should largely remove strong body-size effects.

To further assess the role of body size on risk of DCM, we have now added multi-variable Mendelian randomization (MVMR) analyses. In MVMR analyses, we jointly modeled several of the potentially causal risk factors. Reassuringly, we found that body weight, systolic blood pressure and atrial fibrillation all represented independent risk factors, while height was abolished upon adding body weight to the model ($b = -0.03111133$, $P = 0.6$). These analyses imply that any causal effect of height on DCM may be mediated through body weight. We have added these results to the new **Supplementary Table 28**.

Manuscript change:
Page 7:

“Weight, systolic blood pressure and AF remained as independent risk factors for DCM in multivariable MR analyses (Supplementary Table 28).“

This is of relevance, since the previous studies (with overlapping data sets) already provided evidence for 42 DCM loci (line 157). This number is similar to the 38 GWAS loci for DCM reported here. Thus, it appears that the major novelty comes from the 65 loci identified by MTAG, i.e. by GWAS for global circumferential strain (Ecc) and LVESV. However, GWAS on structural and functional evaluations of MRI data have been reported before and is unclear whether there is (substantial) overlap with previous reports.

We thank the Reviewer for this point. We would like to clarify some aspects of our results and methodology. Firstly, we acknowledge that the sentence on line 157 was ambiguous; the 42 lead variants referred to variants that were discovered in the current study and which were evaluated against previously reported GWAS of left ventricular parameters. The most recent comparable GWAS of a simple DCM outcome, reported in a preprint by Zheng et al., identified 26 genome-wide significant loci, whereas the current study identified 38 distinct loci, of which 27 have not been previously described.

With respect to related traits, the main focus of our work was to perform a GWAS for DCM. We leveraged GWAS data for cardiac functional/volumetric traits from a recent preprint (Tadros et al. 2023), with the goal of enhancing the DCM GWAS using correlated traits. To this end, we used an MTAG framework. MTAG takes >1 input GWAS and computes new summary statistics for all input GWAS, taking into account the shared genetics across the inputs.

Throughout our work, we discuss only the loci arising from the MTAG output for DCM ('MTAG-DCM'). We do not report or discuss the GWAS or MTAG output for the included endophenotypes. Theoretically, all reported loci should be interpreted as genetic loci for DCM. Of course, as with any method, MTAG has limitations and assumptions. To scrutinize the MTAG-DCM loci, we have added several new layers of analyses. For instance, as inspired by the Reviewer, we assessed whether MTAG loci show at least some level of significance in GWAS-DCM (see next comment below). Importantly, we also added replication in an independent set of cases (which were not aided by MTAG) which showed good replication given power (**Figure R1**).

As suggested, we have also added a look-up regarding the overlap of our DCM loci with genome-wide significant loci from previous LV trait and heart failure GWAS. These analyses are described in more detail below.

- The authors should clarify by which extent the MTAG loci provide at least a Bonferroni-corrected significant signals for DCM, e.g. <0.0007 . If some loci offer no signal for DCM, it is questionable whether they reflect LV dysfunction and can be meta-analysed with DCM in a meaningful way.

As suggested, we assessed whether MTAG lead variants showed at least some level of significance in GWAS-DCM. Focusing only on the strongest lead variant in each locus, we found that 60/65 loci reached the Bonferroni-corrected level of significance ($P<0.00077$) in GWAS-DCM, while 63 out of 65 loci reached at least a nominal level of significance ($P<0.05$). Of the loci showing no evidence of association (*IGFBP3*, *GATA4*), we note that *GATA4* was convincingly validated in our independent replication cohort. Taken together, our MTAG approach appears largely robust, which is likely a reflection of the stringent selection for input traits. Nevertheless, a limited number of loci (eg. *IGFBP3*) may represent false-positives.

- The authors should be more distinct on the number of novel loci, since GWAS for structural and functional evaluations of MRI data have been published before. I.e., loci reported for genome-wide association with functional and structural cardiac phenotypes (PMID: 32382064) or heart failure (e.g. PMID: 36376295) before should not be declared as novel here.

We thank the Reviewer for this point, and we appreciate their thoughts on novelty pertaining to related GWAS/traits. While we partially agree with the Reviewer on this point, we think it is a high bar to declare novelty solely for loci not associated with related traits. As we will explain, we think novelty for the specific phenotype of DCM is scientifically important to the field.

While the genetics of general heart failure (HF) and DCM are correlated, HF represents a highly heterogeneous phenotype for which many (extra-myocardial) mechanisms play a strong role (eg. stronger lifestyle effects, coronary artery health/disease, blood lipids, kidney disease, diabetes, and many more). We therefore feel that highlighting novel DCM loci - even if potentially identified for HF - provides valuable information. These loci may help tease out mechanisms for myocardial dysfunction, and also increase our understanding of HF loci more broadly.

Similarly, we concede that the common variant genetics of DCM and LV endophenotypes are highly correlated. However, the genetic correlations are not perfect ($\sim 0.6-0.75$, depending on trait) and several DCM loci do not reach meaningful significance for LV traits (eg, *PITX2*, *CAMK2D*). At the same time, it becomes a rather

blurry endeavor to declare novelty based on any published cardiac MRI trait. Where should we draw the line (volumetric and contractile; hypertrophy traits; cardiac fibrosis; AI-inferred LV shape; etc)?

All that being said, we think it is most clean to retain our ‘primary’ novelty classification based on previously published DCM GWAS studies. In agreement with the Reviewer, we have also added a ‘secondary’ level of novelty based on the studies provided by the Reviewer (using GWAS for HF, LVESVi, LVEDVi, LVEF, and SVi). We have added the look-ups to **Supplementary Tables 8 and 12**, and we refer to these data in the **Main Text**. We posit that this presentation provides a balanced view of novelty for our loci.

Manuscript change:

Page 3:

“At conventional genome-wide significance ($P < 5 \times 10^{-8}$) we uncovered 38 distinct loci, of which 27 were not previously described for DCM (Figure 2a, Supplementary Tables 6-7). Of novel loci, 12 were also not identified in two published GWAS for LV function9 and general HF18 (Supplementary Table 8).”

Page 3:

“MTAG-DCM identified 65 significant loci, of which 50 were not previously published for DCM (Supplementary Tables 10-11; Extended Data Figure 3). Of novel loci, 24 also did not overlap loci from published LV function9 and HF18 GWAS (Supplementary Table 12).”

The reported association between mean platelet thrombocyte volume and DCM is somewhat unexpected and not further discussed by the authors. It would be interesting to have some background information (or a statement that it may be a false positive association).

We agree with the Reviewer on this point. We should note that this potentially causal association – identified in our MR screen – did not survive our subsequent filtering/sensitivity pipeline (unlike the associations that we highlight in the main text). We have more clearly stated in a **Supplementary Note** that this link was not verified using all our sensitivity methods, and that it likely represents a false-positive.

Manuscript change:

“and 2 potential consequences of DCM liability (HF and mean platelet thrombocyte volume; Supplementary Table 27; Figure 4a; Supplementary Note).”

Supplementary Materials change:

Page 48:

“Causal consequences of DCM liability

In our Mendelian randomization (MR) screen, we identified two potentially causal consequences of DCM liability, namely heart failure (HF) and platelet volume. The potentially causal effect of DCM liability on

platelet volume was disputed by our sensitivity analyses. In particular, the link did not reach significance when using MR-Egger regression (Supplementary Table 27). For these reasons, we posit that this link likely represents a false-positive. Of note, the potentially causal link between DCM liability and HF did pass all sensitivity analyses and filters. In particular, CAUSE identified a strong causal effect of DCM liability on HF risk ($g=0.06$, 95%CI [0.04, 0.09]; Figure 5a; Supplementary Figure 11). This finding might reflect that a subset of HF cases have DCM, or that DCM genetics is causative of systolic HF more broadly, as investigated further in our PRS analyses. ”

References

- Biddinger, Kiran J., Sean J. Jurgens, Dimitri Maamari, Liam Gaziano, Seung Hoan Choi, Valerie N. Morrill, Jennifer L. Halford, et al. 2022. "Rare and Common Genetic Variation Underlying the Risk of Hypertrophic Cardiomyopathy in a National Biobank." *JAMA Cardiology* 7 (7): 715–22.
- Chaffin, Mark, Irinna Papangeli, Bridget Simonson, Amer-Denis Akkad, Matthew C. Hill, Alessandro Arduini, Stephen J. Fleming, et al. 2022. "Single-Nucleus Profiling of Human Dilated and Hypertrophic Cardiomyopathy." *Nature* 608 (7921): 174–80.
- Koenig, Andrew L., Irina Shchukina, Junedh Amrute, Prabhakar S. Andhey, Konstantin Zaitsev, Lulu Lai, Geetika Bajpai, et al. 2022. "Single-Cell Transcriptomics Reveals Cell-Type-Specific Diversification in Human Heart Failure." *Nature Cardiovascular Research* 1 (3): 263–80.
- O'Connor, Luke J., Armin P. Schoech, Farhad Hormozdiari, Steven Gazal, Nick Patterson, and Alkes L. Price. 2019. "Extreme Polygenicity of Complex Traits Is Explained by Negative Selection." *American Journal of Human Genetics* 105 (3): 456–76.
- Reichart, Daniel, Eric L. Lindberg, Henrike Maatz, Antonio M. A. Miranda, Anissa Viveiros, Nikolay Shvetsov, Anna Gärtner, et al. 2022. "Pathogenic Variants Damage Cell Composition and Single Cell Transcription in Cardiomyopathies." *Science* 377 (6606): eabo1984.
- Tadros, Rafik, Sean L. Zheng, Christopher Grace, Paloma Jordà, Catherine Francis, Sean J. Jurgens, Kate L. Thomson, et al. 2023. "Large Scale Genome-Wide Association Analyses Identify Novel Genetic Loci and Mechanisms in Hypertrophic Cardiomyopathy." *medRxiv : The Preprint Server for Health Sciences*, February. <https://doi.org/10.1101/2023.01.28.23285147>.
- Zheng, Sean L., Albert Henry, Douglas Cannie, Michael Lee, David Miller, Kathryn A. McGurk, Isabelle Bond, et al. 2023. "Genome-Wide Association Analysis Reveals Insights into the Molecular Etiology Underlying Dilated Cardiomyopathy." *medRxiv*. <https://doi.org/10.1101/2023.09.28.23295408>.

Decision Letter, first revision:

25th Jul 2024

Dear Dr. Bezzina,

Thank you for submitting your revised manuscript "Genome-wide association study reveals mechanisms underlying dilated cardiomyopathy and myocardial resilience" (NG-LE64126R). It has now been seen by the original referees and their comments are below. The reviewers find that the paper has improved in revision, and therefore we'll be happy in principle to publish it in Nature Genetics,

pending minor revisions to comply with our editorial and formatting guidelines.

Sincerely,
Wei

Wei Li, PhD
Senior Editor
Nature Genetics
www.nature.com/ng

Reviewer #1 (Remarks to the Author):

It was a real pleasure to read this thorough response, which convincingly addressed all of the points raised.

Reviewer #2 (Remarks to the Author):

N/A

Reviewer #3 (Remarks to the Author):

The authors performed a number of further analyses that addressed all my comments.

I have no further issues.

Final Decision Letter:

8th Oct 2024

Dear Dr. Bezzina,

I am delighted to say that your manuscript "Genome-wide association study reveals mechanisms underlying dilated cardiomyopathy and myocardial resilience" has been accepted for publication in an upcoming issue of Nature Genetics.

Over the next few weeks, your paper will be copyedited to ensure that it conforms to Nature Genetics

style. Once your paper is typeset, you will receive an email with a link to choose the appropriate publishing options for your paper and our Author Services team will be in touch regarding any additional information that may be required.

Your paper will be published online after we receive your corrections and will appear in print in the next available issue. You can find out your date of online publication by contacting the Nature Press Office (press@nature.com) after sending your e-proof corrections.

Please note that *Nature Genetics* is a Transformative Journal (TJ). Authors may publish their research with us through the traditional subscription access route or make their paper immediately open access through payment of an article-processing charge (APC). Authors will not be required to make a final decision about access to their article until it has been accepted. Find out more about Transformative Journals

Authors may need to take specific actions to achieve compliance with funder and institutional open access mandates. If your research is supported by a funder that requires immediate open access (e.g. according to Plan S principles) then you should select the gold OA route, and we will direct you to the compliant route where possible. For authors selecting the subscription

publication route, the journal's standard licensing terms will need to be accepted, including <https://www.nature.com/nature-portfolio/editorial-policies/self-archiving-and-license-to-publish>. Those licensing terms will supersede any other terms that the author or any third party may assert apply to any version of the manuscript.

If you have not already done so, we strongly recommend that you upload the step-by-step protocols used in this manuscript to protocols.io. protocols.io is an open online resource that allows researchers to share their detailed experimental know-how. All uploaded protocols are made freely available and are assigned DOIs for ease of citation. Protocols can be linked to any publications in which they are used and will be linked to from your article. You can also establish a dedicated workspace to collect all your lab Protocols. By uploading your Protocols to protocols.io, you are enabling researchers to more readily reproduce or adapt the methodology you use, as well as increasing the visibility of your protocols and papers. Upload your Protocols at <https://protocols.io>. Further information can be found at <https://www.protocols.io/help/publish-articles>.

Sincerely,
Wei

Wei Li, PhD
Senior Editor
Nature Genetics
www.nature.com/ng